# Diversity By Design: Leveraging Distribution Matching for Offline Model-Based Optimization

**Michael S. Yao**[1] **James C. Gee**[1] **Osbert Bastani**[1]

## Abstract

The goal of offline model-based optimization (MBO) is to propose new designs that maximize a reward function given only an offline dataset. However, an important desiderata is to also propose a *diverse* set of final candidates that capture many optimal and near-optimal design configurations. We propose **D**iversity **i**n **A**dversarial **M**odel-based **O**ptimization (**DynAMO**) as a novel method to introduce design diversity as an explicit objective into any MBO problem. Our key insight is to formulate diversity as a *distribution matching problem* where the distribution of generated designs captures the inherent diversity contained within the offline dataset. Extensive experiments spanning multiple scientific domains show that DynAMO can be used with common optimization methods to significantly improve the diversity of proposed designs while still discovering high-quality candidates.

## 1. Introduction

Discovering designs that optimize certain desirable properties is a ubiquitous task that spans a wide range of scientific and engineering domains. For example, we might seek to design a drug with the most potent therapeutic efficacy (Brown et al., 2019; Kong et al., 2023; Du et al., 2024); build a robot that is most capable of navigating complex environments (Ahn et al., 2020; Trabucco et al., 2021; Wang et al., 2023); or engineer a material with a certain desirable property (Stanev et al., 2018; Pogue et al., 2023; Gashmard et al., 2024; Ma et al., 2024). However, experimentally validating every proposed design can be expensive, time-intensive, or even impossible in many applications. These limitations can preclude the use of conventional 'online' optimization methods for such generative design tasks.

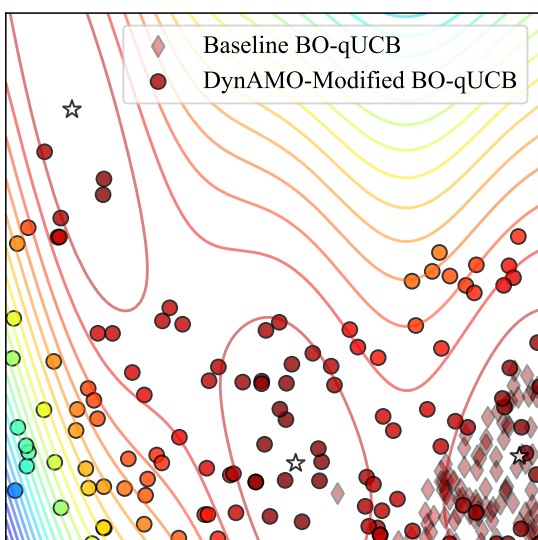

*Figure 1.* **Overview of Diversity in Adversarial Model-based Optimization (DynAMO).** Traditional model-based optimization (MBO) (Trabucco et al., 2021) techniques can generate high-scoring designs, although often at the expense of the *diversity* of proposed designs. Ideally, the final set of candidates should be of high quality while capturing multiple 'modes of goodness' within the design space. For example, although there are 3 unique global maxima (stars) in the 2D Branin (Branin, 1972) optimization problem, traditional Bayesian optimization (BO-qUCB) proposes designs clustered around only a singla optima (diamonds). In contrast, we show how DynAMO can be used to modify the MBO objective to discover diverse *and* high-quality designs (circles).

An alternative approach is to instead discover design candidates in the *offline* setting, where we assume that no newly proposed designs can be experimentally evaluated during the course of the optimization process. Instead, we only have access to a static dataset of previously observed designs and their corresponding reward values. The objective then is to propose a (small) set of candidate designs to ultimately evaluate experimentally, with the hope that using the information available in the offline dataset will yield desirable designs in the real-world.

Multiple prior works have proposed a variety of offline optimization algorithms (Yu et al., 2021; Trabucco et al., 2021; Fu & Levine, 2021; Chen et al., 2022; Mashkaria et al., 2023;

[1]University of Pennsylvania, Philadelphia, PA, USA. Correspondence to: Michael Yao <myao2199@seas.upenn.edu>.

*Proceedings of the 42nd International Conference on Machine Learning*, Vancouver, Canada. PMLR 267, 2025. Copyright 2025 by the author(s).

Krishnamoorthy et al., 2023; Nguyen et al., 2023; Kim et al., 2023). Using a static, offline dataset of previously evaluated designs and their corresponding oracle scores, an offline algorithm proposes a small set of final candidate designs that are empirically evaluated using the expensive 'oracle' function. Broadly, these algorithms can be divided into two categories: *model-based* and *model-free*, where 'model' refers to a predictive surrogate function trained on the offline dataset to approximate the hidden oracle function. We specifically consider **model-based optimization (MBO)** algorithms (Trabucco et al., 2021) here that explicitly optimize against the offline forward surrogate model to discover designs that maximize the final oracle reward.

A secondary, often overlooked metric in offline MBO is *candidate diversity* (Jain et al., 2022; Kim et al., 2023; Maus et al., 2023): it is often ideal to include a diverse array of designs in the final samples proposed by an optimization procedure (**Fig. 1**). Different designs may achieve promising oracle rewards in different ways, and many real-world optimization tasks seek to capture as many of these 'modes of goodness' as possible (Mullis et al., 2019; Jain et al., 2022). Furthermore, there may be secondary optimization objective(s) (e.g., manufacturing cost or drug toxicity) that are better explored and evaluated in a diverse sample set. In these settings, it may be more desirable to sample slightly suboptimal designs in addition to the most optimal design to achieve a greater diversity of proposed candidates.

To this end, we introduce **Diversity in Adversarial Model-based Optimization (DynAMO)** as a novel approach to explicitly control the trade-off between the reward-optimality and diversity of a proposed batch of designs in offline MBO. To motivate our contributions, we show how naïve optimization algorithms provably suffer from poor candidate diversity. To overcome this limitation, we propose a modified optimization objective in the offline setting that encourages discovery of designs that encapsulate the diversity of samples in the offline dataset—an approach inspired by recent advancements in imitation learning and offline reinforcement learning (Ho & Ermon, 2016; Kostrikov et al., 2020; Ke et al., 2021; Ma et al., 2022; Rafailov et al., 2023; Deka et al., 2023; Huang et al., 2024b). We then derive DynAMO as a provably optimal solution to our modified optimization objective. Finally, we empirically demonstrate how DynAMO can be used with a wide variety of different offline optimization methods to propose promising design candidates comparable to the state-of-the-art, while also achieving significantly better candidate diversity.

## 2. Background and Preliminaries

**Offline Model-Based Optimization.** In generative design problems, we seek to learn a generative policy $\pi^*$ over a space of policies $\Pi$ such that the distribution $q^{\pi^*}(x)$ :

$\mathcal{X} \to [0, 1]$ of designs generated by the policy maximizes an **oracle reward function** $r(x) : \mathcal{X} \to \mathbb{R}$

$$\pi^* = \arg\max_{\pi \in \Pi} \mathbb{E}_{x \sim q^\pi(x)}[r(x)] \quad (1)$$

over a design space $\mathcal{X}$. For example, $x$ might be a candidate drug, and the reward $r(x)$ the therapeutic efficacy of the drug. However, in many such problems the oracle reward function may be prohibitively expensive to evaluate; in the aforementioned example, we cannot administer arbitrary doses of potentially dangerous candidate molecules into patients to test their therapeutic efficacy. Similarly, experimentally evaluating designs in materials science discovery often necessitates many months of intensive laboratory work, and building candidate robots from scratch can be intractable. Instead, it is more common to have access to a static dataset of previously evaluated designs $\mathcal{D} = \{(x_i, y_i)\}_{i=1}^n$ where $y_i = r(x_i)$. Such settings where $\mathcal{D}$ is readily available but $r(x)$ is hidden are referred to as *offline optimization*.

To overcome the limitation that $r(x)$ cannot be queried during the optimization process, one approach is to learn a forward surrogate approximation $r_\theta(x)$ of the true reward function $r(x)$. Here, $r_\theta$ is parameterized by $\theta^*$ given by

$$\theta^* = \arg\min_{\theta \in \Theta} \mathbb{E}_{(x_i, y_i) \sim \mathcal{D}} ||y_i - r_\theta(x_i)||_2^2 \quad (2)$$

In practice, such a surrogate model might be a neural network, a physics simulator, or other domain-specific model. In our work, we do not require any particular surrogate model architecture or training paradigm, and consider neural network implementations of $r_\theta$ for generalizability across different domains. Rather than solving (1), we can now instead consider the related optimization problem

$$\pi^* = \arg\max_{\pi \in \Pi} \mathbb{E}_{x \sim q^\pi(x)}[r_\theta(x)] \quad (3)$$

with the hope that optimizing against $r_\theta(x)$ will learn a generative policy that also proposes optimal designs according to $r(x)$, too. Such an approach is often referred to as offline **model-based optimization** (MBO) (Trabucco et al., 2021). Traditionally, an important limitation of offline MBO is the distribution shift between the forward surrogate $r_\theta$ and the oracle reward $r$: that is, $r_\theta$ may incorrectly overestimate the reward associated with proposed designs that are out-of-distribution compared to $\mathcal{D}$, which can often be exploited by traditional optimization algorithms (Trabucco et al., 2021; Yu et al., 2021; Fu & Levine, 2021; Yao et al., 2024).

**Optimization Algorithms.** To solve (3) and similar problem formulations, a number of optimization algorithms have been reported in prior work. One of the most popular approaches is *first-order methods* such as gradient ascent, adaptive moment estimation (Adam) (Kingma & Ba, 2015), and derivative work (Duci et al., 2011; Loshchilov & Hutter,

2019; Zhu et al., 1997; Defazio et al., 2024). Broadly, these optimizers leverage the gradient $\vec{\nabla}_x r_\theta$ of the forward surrogate to iteratively update a candidate design. However, such techniques have been shown to struggle in optimizing against highly non-convex functions typical of real-world offline optimization problems (Trabucco et al., 2021; 2022).

*Evolutionary algorithms*, such as covariance matrix adaptation evolution strategy (CMA-ES) (Hansen, 2016; 2006) and cooperative synapse neuroevolution (CoSyNE) (Gomez et al., 2008), are an alternative approach to optimization. Inspired by biological evolution, such methods iteratively improve a population of candidate solutions using mechanisms like selection and mutation, and do not require gradient information from the forward model.

Separately, *Bayesian optimization* (BO) (Kushner, 1964) is another model-based optimization technique historically used to optimize reward functions that are non-convex, noisy, and/or lack a closed-form expression. Briefly, BO iteratively alternatives between (1) fitting a probabilistic surrogate model (e.g., a Gaussian process) to the acquired data and their scores according to $r_\theta$; and (2) acquiring new candidate designs according to an acquisition function, such as the expected improvement (EI) or upper confidence bound (UCB) (Ament et al., 2023; Wilson et al., 2018; Zhou et al., 2024a). While BO has traditionally been leveraged for optimization problems using expensive-to-evaluate black-box functions, recent work has shown that BO is also a powerful method for offline optimization tasks, too (Maus et al., 2022; Yao et al., 2024; Eriksson et al., 2019; Hvarfner et al., 2024; Eriksson & Jankowiak, 2021; Astudillo & Frazier, 2019). Prior work from Maus et al. (2023); Hernández-García et al. (2024); and others have investigated how to incorporate diversity in existing BO frameworks; however, such methods either (1) gate whether to sample candidate designs based on a diversity-based thresholding schema; or (2) have specifically been proposed for the BO optimization framework. In contrast, our method explicitly includes diversity via distribution matching as an optimization objective, and is readily compatible with standard optimization algorithms.

**Distribution Matching.** Distribution matching is a technique leveraged in recent work on imitation learning and offline reinforcement learning (RL) (Kostrikov et al., 2020; Ke et al., 2021). The approach considers an experimental setup where RL agents cannot interact with the environment and instead must learn from static, offline expert demonstrations sampled from an unknown state-action-reward distribution. The Kullback-Leibler (KL)- divergence (Matthews et al., 2016) is commonly used to train an agent to minimize the discrepancy between state-action visitations made by the RL agent and the offline expert. Given a sufficiently large and diverse dataset of expert demonstrations, we can also think of the KL divergence as encouraging the agent to

match the diversity of the non-zero support of $p(x)$. Distribution matching has been used in prior work to learn robotic control policies (Ho & Ermon, 2016; Wang et al., 2020; Kostrikov et al., 2020; Ke et al., 2021; Ma et al., 2022) and align language models (Rafailov et al., 2023; Huang et al., 2024b; Chakraborty et al., 2024); here, we demonstrate how distribution matching can also be leveraged in offline generative design (a non-RL application) by matching the distribution of designs learned by a generative policy with the distribution of designs from the offline dataset.

**Generative Adversarial Networks.** Generative adversarial networks (GANs) are a method popularized by Goodfellow et al. (2014); Arjovsky et al. (2017); and others to train a generative model. Such approaches train a generative policy using adversarial supervision provided by a *source critic* $c(x) : \mathcal{X} \to \mathbb{R}$. The source critic and generator are trained in a zero-sum 'game' as the discriminator learns to distinguish between generated and real designs, and the generator simultaneously learns to generate designs that are similar to real examples according to the source critic. One particular GAN architecture introduced by Arjovsky et al. (2017) is the *Wasserstein GAN*, which learns a source critic

$$c^*(x) = \underset{||c||_L \leq 1}{\arg\max} \left[ \mathbb{E}_{x' \sim p(x)} c(x') - \mathbb{E}_{x \sim q(x)} c(x) \right] \quad (4)$$

where $||c||_L$ is the Lipschitz constant of the source critic, $p(x)$ is a distribution over real designs (i.e., from an offline dataset $\mathcal{D}$), and $q(x)$ is a distribution over generated designs. Intuitively, we can think of $c^*(x)$ as assigning a real-number score of 'in-distribution-ness' to an input design $x$: large (resp., small) values of $c^*(x)$ mean that the source critic predicts the input design is in (resp., out of) distribution compared to the reference distribution $p(x)$ over real designs. Yao et al. (2024) previously showed how source critics as in (4) can be leveraged in offline generative design tasks to prevent out-of-distribution evaluation of the forward surrogate model $r_\theta(x)$; we leverage a similar approach in our work.

## 3. Distribution Matching for Generative Offline Optimization

### 3.1. Motivating Limitation of Naïve MBO

Prior work from Mullis et al. (2019); Jain et al. (2022); Kim et al. (2023) have shown that an important challenge in offline optimization as in (3) is that of **reward hacking**: learned generative policies can exploit a small region of the design space, resulting in a low diversity of proposed designs. For example, consider the following lemma:

**Lemma 3.1.** *(Diversity Collapse in Reward Optimization) Suppose that there exists a finite set of globally optimal designs $x_j^*$ such that $x_j^* := \arg\max_{x \in \mathcal{X}} r(x)$ and $r^* := r(x_j^*)$ is the optimal reward given a finite, non-uniform*

*reward function $r(x)$. Given any distribution $q^\pi$, we can decompose it into the form $q^\pi(x) = \sum_j w_j \delta(x - x_j^*) + \sum_j w_j' \mathbb{1}(x = x_j^*) + \tilde{q}(x)$, where $w_j \geq 0$ for all $j$, and $\tilde{q}(x) \geq 0$ and $\tilde{q}(x_j^*) = 0$ for all $j$. Then, $\tilde{q}(x)$ satisfies $\int dx\, \tilde{q}(x) = 0$.*

The proof for **Lemma 3.1** is included in **Appendix A**. Note that this result holds for both the oracle function $r(x)$ and the forward surrogate $r_\theta(x)$. Intuitively, this lemma states that an optimal policy that maximizes (1) (resp., (3) in the offline setting) can only have measurable nonzero support at the global optimizers in the design space $\mathcal{X}$. However, many real-world reward functions do not have a large number of globally optimal designs (Trabucco et al., 2021), leading to a low diversity of generated designs seen in practice (Kim et al., 2023). Furthermore, there is no guarantee that the set of optimal $x_j^*$ cover a large region of the design space; in practice, we might be interested in trading optimality of a subset of designs to achieve a greater diversity.

## 3.2. An Alternative MBO Problem Formulation

To reward generative policies in proposing diverse designs, we modify the original MBO objective in (3) according to

$$J(\pi) := \mathbb{E}_{x \sim q^\pi(x)}[r_\theta(x)] - \frac{\beta}{\tau} D_{\text{KL}}(q^\pi || p_{\mathcal{D}}^\tau) \qquad (5)$$

where $D_{\text{KL}}(\cdot||\cdot)$ is the Kullback–Leibler divergence (KL-divergence) and $\tau, \beta \in \mathbb{R}_+$ are hyperparameters. In subsequent steps, we abbreviate the expectation value over probability distributions $\mathbb{E}_{x \sim q^\pi(x)}[\cdot]$ as $\mathbb{E}_{q^\pi}[\cdot]$ for brevity.

**The temperature hyperparameter $\tau$.** Equation (5) implicitly introduces a hyperparameter $\tau \in \mathbb{R}_+$ to control the trade-off between diversity and optimality. Note that the KL-divergence in (5) is computed with respect to a distribution $p_{\mathcal{D}}^\tau(x)$ defined as the $\tau$-*weighted probability distribution*:

**Definition 3.2** ($\tau$-Weighted Probability Distribution)**.** Suppose that we are given a reward function $r(x) : \mathcal{X} \to \mathbb{R}$ over a space of possible designs $\mathcal{X}$, and access to a static, offline dataset $\mathcal{D}$ of real designs. We define the $\tau$-*weighted probability distribution* over $\mathcal{X}$ (for $\tau \geq 0$) as

$$p^\tau(x) := \frac{\exp(\tau r(x))}{Z^\tau} \qquad (6)$$

where the partition function $Z^\tau := \int_{\mathcal{X}} dx\, \exp(\tau r(x))$ is a normalizing constant. We use the dataset of prior observations $\mathcal{D} = \{(x_i, r(x_i))\}_{i=1}^n$ to empirically approximate $p^\tau(x)$, and refer to this approximation as $p_{\mathcal{D}}^\tau(x) \approx p^\tau(x)$. For $\tau \gg 1$, near-optimal designs that are associated with high reward scores are weighted more heavily in $p_{\mathcal{D}}^\tau$; conversely, $\tau = 0$ weights all designs equally to achieve the greatest diversity in designs. The penalized objective in (5) thereby encourages the learned policy to capture the diversity of designs in the $\tau$-weighted distribution $p_{\mathcal{D}}^\tau(x)$. We show empirical $\tau$-weighted distributions in **Appendix D.7**.

**The KL-divergence strength hyperparameter $\beta$.** Separately, the hyperparameter $\beta \geq 0$ controls the relative importance of the distribution matching objective. As $\beta \to \infty$, it becomes increasingly important for the generator to learn a distribution of designs that match $p_{\mathcal{D}}^\tau(x)$; setting $\beta = 0$ reduces $J(\pi)$ to the original MBO objective in (3).

## 3.3. Adversarial Source Critic as a Constraint

Separately, to address the problem of forward surrogate model overestimation of candidate design fitness according to $r_\theta(x)$, we constrain the optimization problem to ensure that expected source critic scores over $q^\pi(x)$ and $p_{\mathcal{D}}^\tau(x)$ differ by no more than a constant $W_0 \in \mathbb{R}_+$, similar to the approach to offline MBO used by Yao et al. (2024). That is,

$$\max_{\pi \in \Pi} \quad J(\pi) = \mathbb{E}_{q^\pi}[r_\theta(x)] - \frac{\beta}{\tau} D_{\text{KL}}(q^\pi || p_{\mathcal{D}}^\tau)$$
$$\text{s.t.} \quad \mathbb{E}_{p_{\mathcal{D}}^\tau}[c^*(x)] - \mathbb{E}_{q^\pi}[c^*(x)] \leq W_0 \qquad (7)$$

where the source critic $c^* : \mathcal{X} \to \mathbb{R}$ is a neural network as in (4) that maximizes $\mathbb{E}_{p_{\mathcal{D}}^\tau}[c^*(x)] - \mathbb{E}_{q^\pi}[c^*(x)]$ subject to the constraint $||c^*(x)||_L \leq 1$, where $||\cdot||_L$ is the Lipschitz norm. Intuitively, this constraint prevents the evaluation of the forward surrogate model $r_\theta(x)$ on wildly out-of-distribution inputs encountered in the offline setting.

We are now interested in finding a generative policy $\pi^*$ that solves this optimization problem in (7); in our work below, we demonstrate how this approach can yield a policy that generates high-scoring candidate designs that also better capture the diversity of possible designs in $\mathcal{X}$.

## 3.4. Constrained Optimization via Lagrangian Duality

Our problem in (7) is ostensibly challenging to solve: both the objective $J(\pi)$ and the constraint imposed by the source critic can be arbitrarily non-convex, making traditional constrained optimization techniques intractable in solving the optimization problem out-of-the-box. In this section, we derive an explicit solution to (7) to make the problem tractably solvable using any standard optimization algorithm.

Recall from Lagrangian duality that solving (7) is equivalent to the min-max problem

$$\min_{\pi \in \Pi} \max_{\lambda \in \mathbb{R}_+} \mathcal{L}(\pi; \lambda) \qquad (8)$$

where the Lagrangian $\mathcal{L}(\pi; \lambda) : \Pi \times \mathbb{R}_+ \to \mathbb{R}$ is given by

$$\mathcal{L}(\pi; \lambda) = -J(\pi) + \beta\lambda \left[ \mathbb{E}_{p_{\mathcal{D}}^\tau}[c^*(x)] - \mathbb{E}_{q^\pi}[c^*(x)] - W_0 \right] \qquad (9)$$

introducing $\lambda \in \mathbb{R}_+$ such that $\beta\lambda \in \mathbb{R}_+$ is the Lagrange multiplier associated with the constraint in (7). From weak duality, the *Lagrange dual problem* provides us with a tight

lower bound on the primal problem in (7):

$$\max_{\lambda \in \mathbb{R}_+} \min_{\pi \in \Pi} \mathcal{L}(\pi; \lambda) := \max_{\lambda \in \mathbb{R}_+} g(\lambda) \leq \min_{\pi \in \Pi} \max_{\lambda \in \mathbb{R}_+} \mathcal{L}(\pi; \lambda) \tag{10}$$

where $g(\lambda) := \min_{\pi \in \Pi} \mathcal{L}(\pi; \lambda)$ is the *Lagrange dual function*. In general, computing $g(\lambda)$ is challenging for an arbitrary offline optimization problem; in prior work, Trabucco et al. (2021) bypassed this dual problem entirely by treating $\lambda$ as a hyperparameter tuned by hand (albeit for a different constraint); and Yao et al. (2024) approximated the dual function under certain assumptions about the input space by performing a grid search over possible $\lambda$ values. In our approach, we look to rewrite the problem into an equivalent representation that admits a closed-form, computationally tractable expression for $g(\lambda)$.

**Lemma 3.3** (Entropy-Divergence Formulation). *Define* $J(\pi)$ *as in* (5). *An equivalent representation of* $J(\pi)$ *is*

$$J(\pi) \simeq -\mathcal{H}(q^\pi(x)) - (1 + \beta)D_{KL}(q^\pi(x)||p_{\mathcal{D}}^\tau(x)) \tag{11}$$

*where* $\mathcal{H}(\cdot)$ *is the Shannon entropy. Maximizing* (11) *is equivalent to maximizing* (5) *in the sense that both objectives admit the same optimal policy.*

The proof of this result is in **Appendix A**. To build intuition about how (11) is equivalent to (5), we can consider the behavior of the objective in the limit of $\tau \to +\infty$: the reference distribution $p_{\mathcal{D}}^\tau$ approaches the sum-of-$\delta$-distributions formulation in **Lemma 3.1**. In this setting, the entropy and KL-divergence terms are equivalent, and the optimal policy $\pi^*$ admits a distribution $q^\pi$ with nonzero support only at the globally optimal designs in $p_{\mathcal{D}}^\tau$. Alternatively in the limit that $\tau \to 0$ and $\beta \to +\infty$, $p_{\mathcal{D}}^\tau$ approaches a uniform distribution and both (5) and (11) simplify to a state-matching objective according to the KL-divergence loss term, without any explicit optimization against the surrogate model $r_\theta(x)$.

The utility of **Lemma 3.3** is to enable us to write an exact formulation for the Lagrangian dual function $g(\lambda)$:

**Lemma 3.4** (Explicit Dual Function of (7)). *Consider the primal problem*

$$\max_{\pi \in \Pi} \quad J(\pi) \simeq -\mathcal{H}(q^\pi) - (1 + \beta)D_{KL}(q^\pi||p_{\mathcal{D}}^\tau)$$
$$s.t. \quad \mathbb{E}_{p_{\mathcal{D}}^\tau}[c^*(x)] - \mathbb{E}_{q^\pi}[c^*(x)] \leq W_0 \tag{12}$$

*The Lagrangian dual function* $g(\lambda)$ *is bounded from below by the function* $g_\ell(\lambda)$ *given by*

$$g_\ell(\lambda) := \beta \left[ \lambda(\mathbb{E}_{p_{\mathcal{D}}^\tau}[c^*(x)] - W_0) - \mathbb{E}_{p_{\mathcal{D}}^\tau} e^{\lambda c^*(x) - 1} \right] \tag{13}$$

The proof of this result is included in **Appendix A**. **Lemma 3.4** admits an explicit concave function $g_\ell(\lambda)$ such that $g(\lambda) \geq g_\ell(\lambda)$ for all $\lambda \in \mathbb{R}_+$; because we are interested in maximizing the dual function in leveraging Lagrangian

duality as in (10), it follows that maximizing $g_\ell(\lambda)$ bounds the maxima over $g(\lambda)$ from below. In subsequent steps, we therefore optimize over this explicit function $g_\ell(\lambda)$.

The utility of **Lemma 3.4** is in solving for the optimal $\lambda$ that maximizes the dual function lower bound in (13). Prior work has explored approximating $\lambda$ via a grid search (Yao et al., 2024) or using iterative implicit solvers; these methods cannot provide any formal guarantee in arriving at a reasonable solution for $\lambda$. In contrast, maximizing against $g_\ell(\lambda)$ is easy because the function is *guaranteed* to be concave for any $\beta, \tau, W_0$ and source critic $c^*(x)$. We can therefore derive an *exact* solution for $\lambda$ using any convex optimization problem solver. We now have a method to write an explicit expression for the Lagrangian $\mathcal{L}(\pi; \lambda)$ by exactly specifying the optimal $\lambda$, and then leverage any out-of-the-box policy optimization method to solve (7) via solving the easier, ostensibly unconstrained problem in (8).

### 3.5. Overall Algorithm

To summarize, our work aims to solve two separate but related problems in offline MBO in (3): traditional model-based optimization approaches can yield candidate designs that are [1] of low diversity; and [2] not optimal due to exploiting out-of-distribution errors of the forward surrogate $r_\theta(x)$. We introduce a KL-divergence-based distribution matching objective—with input hyperparameters $\tau$ and $\beta$—to solve the diversity problem; and build off prior work (Yao et al., 2024) to constrain the search space using source critic feedback to solve the out-of-distribution evaluation problem. We then show that there exists a provable, explicit solution to our modified offline MBO problem (i.e., **Lemma 3.4** and (8)). In contrast with prior work imposing specific constraints on the forward model (Trabucco et al., 2021; Yu et al., 2021) or design space (Yao et al., 2024), or requiring the use of model-free optimization methods (Krishnamoorthy et al., 2023; Mashkaria et al., 2023), *our approach only modifies the MBO objective and is therefore both optimizer- and task- agnostic*. We refer to our method as **Di**versity **in** **A**dversarial **M**odel-based **O**ptimization (**DynAMO**).

## 4. Experimental Evaluation

**Datasets and Offline Optimization Tasks.** We evaluate DynAMO on a set of six real-world offline MBO tasks spanning multiple scientific domains and both discrete and continuous search spaces. Five of the tasks are from Design-Bench, a publicly available set of offline optimization benchmarking tasks from Trabucco et al. (2022): (1) **TFBind8** aims to maximize the transcription factor binding efficiency of a short DNA sequence (Barrera et al., 2016); (2) **UTR** the gene expression from a 5' UTR DNA sequence (Sample et al., 2019; Angermüeller et al., 2020); (3) **ChEMBL** the mean corpuscular hemoglobin concentration (MCHC) biological

---

**Algorithm 1 (DynAMO). D**iversity in **A**dversarial **M**odel-based **O**ptimization

---

**Inputs:**

$r_\theta : \mathcal{X} \to \mathbb{R}$ — pre-trained forward surrogate model
$c^* : \mathcal{X} \to \mathbb{R}$ — initialized source critic model
$\mathcal{D} = \{(x'_j, r(x'_j))\}_{j=1}^n$ — reference dataset
$\beta \geq 0$ — KL regularization strength
$\tau \geq 0$ — temperature
$b \geq 1$ — batch size
$a^b : \mathcal{X} \times \mathbb{R} \to \mathcal{X}^b$ — optimizer algorithm
$\eta_{\text{critic}} > 0$ — source critic learning rate
$\eta_\lambda > 0$ — $\lambda$ dual step size
$k \geq 1$ — oracle evaluation budget

Initialize sampled candidates $\mathcal{D}_{\text{gen}} = \varnothing \subset \mathcal{X} \times \mathbb{R}$
**while** $a^b$ has not converged **do**

// Solve for the globally optimal $\lambda$ using (13)
$\lambda \leftarrow \lambda_0 \quad (\lambda_0 = 1.0$ in our experiments$)$
**while** $\lambda$ has not converged **do**
$\qquad \lambda \leftarrow \lambda + \eta_\lambda \frac{\partial g_\ell(\lambda)}{\partial \lambda}$
**end while**

// Given previously sampled candidates $\mathcal{D}_{\text{gen}}$ as input,
// sample new candidates using the optimizer
$\{x_i^{\text{new}}\}_{i=1}^b \leftarrow a^b(\mathcal{D}_{\text{gen}})$

// Re-train the source critic parameters $\theta_c$
**while** $\delta W$ has not converged **do**
$\qquad \delta W \leftarrow \vec{\nabla}_{\theta_c} \left[ \mathbb{E}_{x' \sim \mathcal{D}}[c^*(x')] - \mathbb{E}_{x \sim \{x_i^{\text{new}}\}_{i=1}^b}[c^*(x)] \right]$
$\qquad \theta_c \leftarrow \min(\max(\theta_c + \eta_{\text{critic}} \cdot \delta W, -0.01), 0.01)$
**end while**

// Evaluate and cache the candidates according to (9)
$\mathcal{D}_{\text{gen}} \leftarrow \mathcal{D}_{\text{gen}} \cup \{(x_i^{\text{new}}, -\mathcal{L}(x_i^{\text{new}}; \lambda))\}_{i=1}^b$
**end while**
**return** top-$k$ candidates from $\mathcal{D}_{\text{gen}}$ according to their penalized MBO objective values

---

response of a molecule using an offline dataset collected from the ChEMBL assay CHEMBL3885882 (Gaulton et al., 2011); (4) **Superconductor** the critical temperature of a superconductor material specified by its chemical formula design (Hamidieh, 2018); and (5) **D'Kitty** the morphological structure of a quadrupedal robot (Ahn et al., 2020). Tasks (1) - (3) (i.e., TFBind8, UTR, and ChEMBL) are discrete optimization tasks, where tasks (4) and (5) (i.e., Superconductor and D'Kitty) are continuous optimization tasks. We also evaluate our method on the discrete (6) **Molecule** task described in Brown et al. (2019); Chen et al. (2021); Flam-Shepherd et al. (2022); Yao et al. (2024), where the goal is to design a maximally hydrophobic molecule. Additional implementation details are detailed in **Appendix B**.

**Experiment Implementation.** All our optimization tasks include an offline, static dataset $\mathcal{D} = \{(x_i, r(x_i))\}_{i=1}^n$ of previously observed designs and their corresponding objective values. We first use $\mathcal{D}$ to train a task-specific forward surrogate model $r_\theta$ with parameters $\theta^*$ according to (2). We parameterize each forward surrogate model $r_\theta(x)$ as a fully connected neural network with two hidden layers of size 2048 and LeakyReLU activations, trained using an Adam optimizer with a learning rate of $\eta = 0.0003$ for 100 epochs.

Importantly, optimization problems over *discrete* search spaces are generally NP-hard and often involve heuristic-based solutions (Papalexopoulos et al., 2022; Xiong, 2022). Instead, we use the standard approach of learning a variational autoencoder (VAE) (Kingma & Welling, 2014) to encode and decode discrete designs to and from a continuous latent space, and optimize over the continuous VAE latent space instead—see **Appendix B** for additional details.

DynAMO also involves training and implementing a source critic model $c^*(x)$ as in (4); we implement $c^*$ as a fully connected neural network with two hidden layers each with size 512. We implement the constraint on the model's Lipschitz norm by clamping the weights of the model such that the $\ell_\infty$-norm of the parameters is no greater than 0.01 after each optimization step, consistent with Arjovsky et al. (2017). We train the critic using gradient descent with a learning rate of $\eta = 0.01$ according to (4). Separately to solve for the globally optimal $\lambda$ using **Lemma 3.4**, we perform gradient ascent on $\lambda$ until the algorithm converges. Finally, we fix the KL-divergence weighting $\beta = 1.0$, temperature hyperparameter $\tau = 1.0$, and constraint bound $W_0 = 0$ for all experiments to avoid overfitting DynAMO to any particular task or optimizer. All experiments were run for 10 random seeds on a single internal cluster with 8 NVIDIA RTX A6000 GPUs. Of note, all DynAMO experiments were run using only a single GPU.

**Baseline Methods.** Our proposed work, DynAMO, specifically looks to modify an offline MBO optimization problem as in (3) where we assume access to a forward surrogate model $r_\theta(x)$ to rank proposed design candidates and offer potential information about the design space. We compare DynAMO against other objective modifying MBO approaches: (1) Conservative Objective Models (**COMs**; Trabucco et al. (2021)) penalizes the objective at a 'look-ahead' gradient-ascent iterate to prevent falsely promising gradient ascent steps; (2) Robust Model Adaptation (**RoMA**; Yu et al. (2021)) modifies the objective $r_\theta(x)$ to enforce a local smoothness prior; (3) Retrieval-enhanced Offline Model-Based Optimization (**ROMO**; Chen et al. (2023c)) retrieves relevant samples from the offline dataset for more trustworthy gradient updates; and (4) Generative Adversarial Model-Based Optimization (**GAMBO**; Yao et al. (2024)) introduces a framework for initially leveraging source critic feedback to regularize an MBO objective. We evaluate each of these MBO objective transformation methods alongside

*Table 1.* **Quality and Diversity of Designs Under MBO Objective Transforms.** We evaluate DynAMO against other MBO objective-modifying methods using six different backbone optimizers. Each cell consists of '**Best@128 (Best)**/**Pairwise Diversity (PD)**' Rank and Optimality Gap scores separated by a forward slash. **Bolded** (resp., Underlined) entries indicate the best (resp., second best) performing algorithm for a given optimizer (i.e., within each column). See **Supplementary Table A1** for detailed results broken down by MBO task.

| **Best/PD** | **Rank ↓** | | | | | | **Optimality Gap ↑** | | | | | |
|---|---|---|---|---|---|---|---|---|---|---|---|---|
| | Grad. | Adam | CMA-ES | CoSyNE | BO-qEI | BO-qUCB | Grad. | Adam | CMA-ES | CoSyNE | BO-qEI | BO-qUCB |
| Baseline | 5.0/5.5 | 4.5/6.0 | 3.7/3.8 | 5.3/4.5 | 5.8/5.2 | 3.7/3.0 | 6.8/-53.2 | 0.5/-52.4 | 14.4/9.5 | -0.6/-52.1 | 18.7/47.1 | 19.4/43.5 |
| COMs⁻ | 7.3/6.5 | 6.0/5.3 | 5.7/5.2 | 5.7/5.7 | 4.3/4.0 | 4.5/4.5 | -3.0/-53.5 | -3.0/-52.4 | 7.6/-9.7 | 1.1/-51.6 | 19.2/51.4 | 19.0/45.5 |
| COMs⁺ | **2.5**/2.8 | **3.2**/3.0 | 7.3/7.8 | 5.7/3.3 | 6.0/5.7 | 5.2/5.7 | 12.3/-6.9 | 8.1/-12.3 | 7.1/-38.2 | 3.5/-40.4 | 17.9/40.3 | 18.6/51.3 |
| RoMA⁻ | 6.7/5.7 | 4.5/6.3 | 3.7/3.5 | 5.3/4.5 | 2.7/3.3 | 3.8/3.0 | -1.2/-53.3 | 0.5/-52.4 | 14.4/9.5 | -0.6/-52.2 | **21.0**/48.4 | 19.2/43.6 |
| RoMA⁺ | 3.8/5.8 | **2.8**/5.2 | 5.0/4.8 | 2.8/5.0 | 5.2/6.3 | 4.7/6.5 | 9.2/-46.8 | **14.5**/-45.8 | 14.1/8.0 | 6.2/-52.0 | 18.3/32.9 | 18.5/39.9 |
| ROMO | 4.2/2.8 | 4.8/2.8 | 4.2/4.3 | 4.2/5.2 | 5.0/6.0 | 4.7/6.2 | 10.9/-12.7 | 6.4/-20.5 | 15.7/-3.1 | 3.1/-50.8 | 19.2/34.9 | 19.9/33.2 |
| GAMBO | 3.2/5.3 | 5.3/5.8 | **2.2**/4.3 | 3.7/6.3 | **2.2**/4.0 | 4.7/5.0 | 10.5/-52.1 | 8.6/-51.9 | 16.7/16.8 | 5.0/-53.6 | 20.8/30.0 | 20.2/30.3 |
| **DynAMO** | 2.8/**1.2** | **2.8**/**1.2** | 3.3/**1.8** | **2.3**/**1.2** | 3.0/**1.3** | 3.5/**1.8** | **14.2**/**27.8** | **14.5**/**35.7** | **17.5**/**55.2** | **12.3**/**-20.7** | 20.7/**74.2** | 20.5/**59.4** |

**DynAMO** and naïve, unmodified **Baseline** MBO using representative first-order methods (1) **Grad.** (Gradient Ascent) and (2) **Adam** (Adaptive Moment Estimation (Kingma & Ba, 2015)); evaluationary algorithms (3) **CMA-ES** (Covariance Matrix Adaptation Evolution Strategy (Hansen, 2016)) and (4) **CoSyNE** (Cooperative Synapse Neuroevolution (Gomez et al., 2008)); and Bayesian optimization with (5) Expected Improvement (**BO-qEI**) and (6) Upper Confidence Bound (**BO-qUCB**) acquisition functions.

Notably, the baseline methods COMs and RoMA impose specific constraints on the training process for the forward surrogate model $r_\theta(x)$, and/or also assume that the forward model can be updated during the sampling process (Yu et al., 2021; Trabucco et al., 2021). These constraints are not generally satisfied for any arbitrary offline MBO problem; for example, $r_\theta$ may be a non-differentiable black-box simulator with fixed parameters. In contrast, both our method (DynAMO) and baseline methods GAMBO and ROMO are compatible with this more general experimental setting; to ensure a fair experimental comparison, we evaluate both RoMA and COMs using a baseline forward surrogate (i.e., RoMA⁻, COMs⁻) and using a specialized forward surrogate model trained and updated according to the methods described by the respective authors (i.e., RoMA⁺, COMs⁺).

**Evaluating the Diversity of Candidate Designs.** To empirically evaluate the diversity of a final set of $k = 128$ candidate designs $\{x_i^F\}_{i=1}^k$ proposed by an offline MBO experiment, we report the **Pairwise Diversity** (PD) of a batch of $k$ candidate designs, defined by Jain et al. (2022); Kim et al. (2023); and Maus et al. (2023) as

$$\text{PD}(\{x_i^F\}_{i=1}^k) := \mathbb{E}_{x_i^F}\left[\mathbb{E}_{x_j^F \neq x_i^F}\left[d(x_i^F, x_j^F)\right]\right] \quad (14)$$

where $d(\cdot, \cdot)$ is the normalized Levenshtein edit distance (Haldar & Mukhopadhyay, 2011) (resp., Euclidean distance) for discrete (resp., continuous) tasks. We detail alternative definitions of diversity in **Appendix D.2**.

**Evaluating the Quality of Candidate Designs.** To en-

sure that diversity does not come at the expense of finding optimal design candidates, we report the **Best@$k$** oracle score obtained by evaluating $k = 128$ candidate designs $\{x_i^F\}_{i=1}^k$ proposed in an experiment. Consistent with prior work (Trabucco et al., 2021; Yao et al., 2024), we define

$$\text{Best@}k(\{x_i^F\}_{i=1}^k) := \max_{1 \leq i \leq k} r(x_i^F) \quad (15)$$

Crucially, the Best@$k$ metric is computed with respect to the oracle function $r(x)$ that was hidden during optimization; we only use $r(x)$ in (15) to report the true reward associated with each candidate design.

Finally, we rank each method for a given optimizer and task and report the method's **Rank** averaged over the six tasks according to the Best@128 (15) and PD (14) metrics. We also report the **Optimality Gap** (Opt. Gap) (averaged over the six tasks), defined as as difference between the score achieved by an MBO optimization method and the score in the offline dataset, for both the Best@128 and PD metrics.

## 5. Results

**Main Results.** DynAMO consistently proposes the most diverse set of designs and achieves an Optimality Gap as high as **74.2** (DynAMO-BO-qUCB) and an average Rank as low as **1.2** compared to baseline methods (**Table 1**). We find that DynAMO offers the largest improvements in diversity for first-order methods, although also improves upon the evolutionary algorithms and Bayesian optimization methods evaluated. This makes sense, as both Grad. and Adam are only local optimizers that often end up exploring a much smaller region of the design space (without using DynAMO) compared to gradient-free methods. For example, DynAMO-Grad. (resp., DynAMO-CMA-ES; resp., DynAMO-BO-qEI) achieves a Pairwise Diversity Optimality Gap of 35.7 (resp., 55.2; resp., 74.2); in contrast, no other baseline method achieves a diversity score greater than -6.9 (resp., 16.8; resp., 51.4) within the same optimizer class.

These results do not come at the cost of the quality of designs; for example, for all 3 optimizers where DynAMO scores an average Rank of 1.2 (i.e., Grad., Adam, and CoSyNE backbone optimizers), DynAMO is also within the **top 2 methods** in proposing *high-quality* designs according to both Rank and Optimality Gap. In fact, DynAMO proposes the best designs for 5 out of the 6 backbone optimizers according to the Best@128 Optimality Gap. These results suggest that DynAMO can be used to improve both the quality *and* diversity of designs in a variety of experimental settings for both discrete and continuous search spaces.

**Ablation Studies.** DynAMO consists of two important but separate algorithmic components: (1) a KL-divergence-based distribution matching objective; and (2) a constraint dependent on an adversarial source critic. We show both components are important for DynAMO to generate both diverse *and* high-quality designs (**Appendix E.2**). DynAMO also takes as input two important hyperparameters—$\beta$ and $\tau$—as introduced in (5). We empirically ablate the values of these hyperparameters in **Appendix E.3**. Additional results and discussion are included in **Appendix E**.

## 6. Related Work

**Model-free offline optimization.** In our work, we specifically look at *model-based optimization* methods that explicitly optimize against a forward surrogate model $r_\theta(x)$ that acts as a proxy for the hidden oracle function $r(x)$. However, related work have also proposed offline optimization methods that do not require access to a model $r_\theta(x)$ and instead impose constraints on the backbone optimization method—we refer to such work as *model-free* offline optimization. For example, Mashkaria et al. (2023) frame generative design tasks as a 'next-sample' prediction problem and learn a transformer to roll out sample predictions; and Krishnamoorthy et al. (2023); Yun et al. (2024) learn a diffusion model to sample candidate designs conditioned on reward values. Because DynAMO operates on MBO forward surrogate models $r_\theta(x)$, we cannot leverage DynAMO with these model-free methods. However, we compare them against DynAMO in **Appendix D.5**.

**Active learning in optimization.** In our work, we specifically consider the experimental setup of *one-shot, batched oracle evaluation*: that is, the final candidate designs that are scored by the oracle function at the end of optimization are *not* used to subsequently update the prior over the design space to better inform subsequent optimization steps. In contrast, a separate body of recent work has investigated generative design in the setting of *active learning* where there can be multiple rounds of offline optimization to inform subsequent online acquisitions (Hernández-García et al., 2024; Li et al., 2022b;a; Wu et al., 2023; Palizhati et al., 2022). For example, Li et al. (2024) show how active learning can be

formulated as a multi-fidelity optimization problem.

**Reinforcement learning.** Prior work has explored how to formulate offline generative design tasks as reinforcement learning (RL) problems. Trabucco et al. (2022) used REINFORCE-style methods similar to Williams (1992) to learn a myopic sampling policy, although do not use RL for offline generative design. Angermueller et al. (2020); Korshunova et al. (2022); and Jang et al. (2022) leverage RL for offline optimization in the active learning setting described above, which is outside the scope of our work.

## 7. Discussion and Conclusion

We introduce **DynAMO**, a novel task- and optimizer- agnostic approach to MBO that improves the diversity of proposed designs in offline optimization tasks. By framing diversity as a distribution-matching problem, we show how DynAMO can enable generative policies to sample both high-quality and diverse sets of designs. Our experiments reveal that DynAMO significantly improves the diversity of proposed designs while also discovering high-quality candidates.

**Limitations and Future Work.** There are also important limitations of our method. Firstly, we note that while DynAMO can significantly improve the diversity of proposed designs in offline MBO while preserving Best@128 performance, our method is not as competitive with existing baselines according to the *median* score obtained by the 128 designs (**Supp. Fig. A2**, **Appendix B**). The suboptimal performance of DynAMO according to this Median@128 metric is unsurprising given that our primary motivation of DynAMO is to obtain a diverse sample of designs while simultaneously ensuring that a nonzero subset of them are (near-) optimal. Furthermore, we empirically observe that no method is state-of-the-art on both Best@128 and Median@128 metrics. While it would be ideal for DynAMO (or any method) to be state-of-the-art for all quality and diversity metrics, we argue that obtaining a good Best@128 score is more important than a good Median@128 score, as the the principle real-world goal of offline MBO is to find *a* design that maximizes the oracle function.

Secondly, we also limit our study of DynAMO to offline MBO tasks that are well-described and studied in prior work. In principle, real-world optimization problems may be complicated by noisy and/or sparse objective functions, ultra-high dimensional search spaces, small offline datasets, and other practical limitations. We leave a more rigorous interrogation of how such offline MBO methods perform in such settings for future work.

Finally, our work focuses on evaluating DynAMO and baseline methods in a one-shot, batched oracle evaluation setting—future work might explore how to extend our method to the active learning setting. Separately, recent domain-

specific foundation models (Lin et al., 2023; Ohana et al., 2024; Nguyen et al., 2024; Zeni et al., 2025) may also give rise to more sophisticated and accurate forward surrogate models $r_\theta(x)$ that can be leveraged with DynAMO and other MBO methods in future work.

## Acknowledgments

We thank Shuo Li at the University of Pennsylvania for their helpful comments and feedback on an earlier draft of this work, and the anonymous referees whose peer review helped significantly improve the quality of our manuscript. MSY is supported by NIH Award F30 MD020264. JCG is supported by NIH Award R01 EB031722. OB is supported by NSF Award CCF-1917852.

## Impact Statement

Offline generative methods such as DynAMO have significant potential to help design more effective drugs; engineer new materials with desirable properties; and solve other scientific tasks. However, like any real-world algorithm, these methods can also be misused to create potentially harmful or dangerous designs. Careful oversight by domain experts and researchers is essential to ensure our contributions are used for social good.

## Code and Data Availability

All datasets used in our experiments are publicly available; offline datasets associated with Design-Bench tasks are made available by Trabucco et al. (2022). The offline dataset for the **Molecule** task is made available by Brown et al. (2019). Our custom code implementation for our experiments is made publicly available at github.com/michael-s-yao/DynAMO.

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

# Appendix

## Table of Contents

## A. Proofs

**Lemma 3.1.** (Diversity Collapse in Reward Optimization) Suppose that there exists a finite set of globally optimal designs $x_j^*$ such that $x_j^* := \arg\max_{x \in \mathcal{X}} r(x)$ and $r^* := r(x_j^*)$ is the optimal reward given a finite, non-uniform reward function $r(x)$. Given any distribution $q^\pi$, we can decompose it into the form $q^\pi(x) = \sum_j w_j \delta(x - x_j^*) + \sum_j w_j' \mathbb{1}(x = x_j^*) + \tilde{q}(x)$, where $w_j \geq 0$ for all $j$, and $\tilde{q}(x) \geq 0$ and $\tilde{q}(x_j^*) = 0$ for all $j$. Then, $\tilde{q}(x)$ satisfies $\int dx\, \tilde{q}(x) = 0$.

*Proof.* First, note that if $\int dx\, \tilde{q}(x) = 0$, we have

$$\mathbb{E}_{x \sim q^\pi(x)}[r(x)] = \int dx \sum_j w_j \delta(x - x_j^*) r(x) = \sum_j w_j \int dx\, \delta(x - x_j^*) r(x) = \sum_j w_j r^* = r^* \sum_j w_j = r^*, \quad (16)$$

which is optimal. Next, we prove that if $\int dx\, \tilde{q}(x) > 0$, then $\mathbb{E}_{x \sim q^\pi}[r(x)] < r^*$. To this end, we define

$$\mathcal{X}_1 := \{x \in \mathcal{X} \mid 1 < r^* - r(x)\}, \quad \mathcal{X}_n := \left\{ x \in \mathcal{X} \,\middle|\, \frac{1}{n} < r^* - r(x) \leq \frac{1}{n-1} \right\} \subseteq \mathcal{X} \quad \forall n \geq 2 \quad (17)$$

for each $n \in \mathbb{N}$. Note that all $\mathcal{X}_n$ are disjoint by construction; also by construction, we have $\mathcal{X} \setminus \{x_j^*\} = \bigcup_{n=1}^\infty \mathcal{X}_n$. Furthermore, note that since $\tilde{q}(x) = 0$ for $x = x_j^*$ for some $j$, we have $0 < \int dx\, \tilde{q}(x) = \sum_{n=1}^\infty \int_{\mathcal{X}_n} dx\, \tilde{q}(x)$, so it must be that $\int_{\mathcal{X}_m} dx\, \tilde{q}(x) > 0$ for some $m$. As a consequence, we have

$$\int_{\mathcal{X}_m} dx\, \tilde{q}(x)(r^* - r(x)) \geq \frac{1}{m} \int_{\mathcal{X}_m} dx\, \tilde{q}(x) > 0.$$

Thus, the expected reward would be

$$
\begin{aligned}
\mathbb{E}_{x \sim q^\pi}[r(x)] &= \int dx\, q^\pi(x) r(x) \\
&= \int dx\, \left( \sum_j w_j \delta(x - x_j^*) + \sum_j w_j' \mathbb{1}(x = x_j^*) \right) r(x) + \sum_{n=1}^{\infty} \int_{\mathcal{X}_n} dx\, \tilde{q}(x) r(x) \\
&< \int dx\, \left( \sum_j w_j \delta(x - x_j^*) + \sum_j w_j' \mathbb{1}(x = x_j^*) \right) r^* + \sum_{n=1}^{\infty} \int_{\mathcal{X}_n} dx\, \tilde{q}(x) r^* \\
&= \int dx\, q^\pi(x) r^* \\
&= r^*
\end{aligned}
\tag{18}
$$

so $q^\pi$ is suboptimal. The claim follows. $\qquad \square$

**Lemma 3.3.** (Entropy-Divergence Formulation) Define $J(\pi)$ as in (5). An equivalent representation of $J(\pi)$ is

$$
J(\pi) = -\mathcal{H}(q^\pi(x)) - (1 + \beta) D_{\mathrm{KL}}(q^\pi(x) \| p_{\mathcal{D}}^\tau(x))
\tag{19}
$$

where $\mathcal{H}(\cdot)$ is the Shannon entropy and $D_{\mathrm{KL}}(\cdot \| \cdot)$ is the KL divergence.

*Proof.* Firstly, note that

$$
\begin{aligned}
J(\pi) &= \mathbb{E}_{q^\pi}[r_\theta(x)] - \frac{\beta}{\tau} D_{\mathrm{KL}}(q^\pi \| p_{\mathcal{D}}^\tau) \\
&\simeq \tau \cdot \mathbb{E}_{q^\pi}\left[ \log e^{r_\theta(x)} \right] - \beta D_{\mathrm{KL}}(q^\pi \| p_{\mathcal{D}}^\tau) \\
&= \mathbb{E}_{q^\pi}\left[ \log e^{\tau r_\theta(x)} \right] - \beta D_{\mathrm{KL}}(q^\pi \| p_{\mathcal{D}}^\tau)
\end{aligned}
\tag{20}
$$

where $\simeq$ denotes an equivalent representation of the objective (i.e., scaling $J(\pi)$ by $\tau > 0$ does not change the optimal policy $\pi^*$). Further rewriting,

$$
J(\pi) = \mathbb{E}_{q^\pi}\left[ \log \frac{e^{\tau r_\theta(x)}}{Z^\tau} \right] + \mathbb{E}_{q^\pi} \log Z^\tau - \beta D_{\mathrm{KL}}(q^\pi \| p_{\mathcal{D}}^\tau) \simeq \mathbb{E}_{q^\pi}\left[ \log \frac{e^{\tau r_\theta(x)}}{Z^\tau} \right] - \beta D_{\mathrm{KL}}(q^\pi \| p_{\mathcal{D}}^\tau)
\tag{21}
$$

where we omit the constant $\mathbb{E}_{q^\pi} \log \mathcal{Z}_\theta^\tau$ because the expectation value argument is independent of the policy $\pi$. The remaining expectation value can be re-expressed via *importance weighting*:

$$
J(\pi) = \mathbb{E}_{p_{\mathcal{D}}^\tau}\left[ \frac{q^\pi}{p_{\mathcal{D}}^\tau} \log \frac{e^{\tau r_\theta(x)}}{Z^\tau} \right] - \beta D_{\mathrm{KL}}(q^\pi \| p_{\mathcal{D}}^\tau)
\tag{22}
$$

Assuming that the surrogate $r_\theta(x)$ is well-trained on the offline dataset $\mathcal{D}$ (i.e., $r(x) \approx r_\theta(x)\ \forall x \in \mathcal{D}$), we have

$$
J(\pi) \approx \mathbb{E}_{p_{\mathcal{D}}^\tau}\left[ \frac{q^\pi}{p_{\mathcal{D}}^\tau} \log \frac{e^{\tau r(x)}}{Z^\tau} \right] - \beta D_{\mathrm{KL}}(q^\pi \| p_{\mathcal{D}}^\tau) = \mathbb{E}_{p_{\mathcal{D}}^\tau}\left[ \frac{q^\pi}{p_{\mathcal{D}}^\tau} \log p_{\mathcal{D}}^\tau \right] - \beta D_{\mathrm{KL}}(q^\pi \| p_{\mathcal{D}}^\tau)
\tag{23}
$$

from **Definition 3.2**. Further rewriting, we have

$$
\begin{aligned}
J(\pi) &= \mathbb{E}_{p_{\mathcal{D}}^\tau}\left[ \frac{q^\pi}{p_{\mathcal{D}}^\tau} \log \left( p_{\mathcal{D}}^\tau \cdot \frac{q^\pi}{q^\pi} \right) \right] - \beta D_{\mathrm{KL}}(q^\pi \| p_{\mathcal{D}}^\tau) \\
&= \mathbb{E}_{p_{\mathcal{D}}^\tau}\left[ \frac{q^\pi}{p_{\mathcal{D}}^\tau} \log \frac{p_{\mathcal{D}}^\tau}{q^\pi} \right] + \mathbb{E}_{p_{\mathcal{D}}^\tau}\left[ \frac{q^\pi}{p_{\mathcal{D}}^\tau} \log q^\pi \right] - \beta D_{\mathrm{KL}}(q^\pi \| p_{\mathcal{D}}^\tau) \\
&= -\mathbb{E}_{p_{\mathcal{D}}^\tau}\left[ \frac{q^\pi}{p_{\mathcal{D}}^\tau} \log \frac{q^\pi}{p_{\mathcal{D}}^\tau} \right] - \mathbb{E}_{q^\pi}[-\log q^\pi] - \beta D_{\mathrm{KL}}(q^\pi \| p_{\mathcal{D}}^\tau)
\end{aligned}
\tag{24}
$$

From the definition of KL-divergence,

$$
\begin{aligned}
J(\pi) &= -(1+\beta)\mathbb{E}_{p_{\mathcal{D}}^{\tau}}\left[\frac{q^{\pi}}{p_{\mathcal{D}}^{\tau}}\log\frac{q^{\pi}}{p_{\mathcal{D}}^{\tau}}\right] - \mathbb{E}_{q^{\pi}}\left[-\log q^{\pi}\right] \\
&= -(1+\beta)\mathbb{E}_{p_{\mathcal{D}}^{\tau}}\left[f_{\mathrm{KL}}\left(\frac{q^{\pi}}{p_{\mathcal{D}}^{\tau}}\right)\right] - \mathbb{E}_{q^{\pi}}\left[f_{\ell}\left(q^{\pi}\right)\right] \\
&= -(1+\beta)D_{\mathrm{KL}}(q^{\pi}\|p_{\mathcal{D}}^{\tau}) - \mathcal{H}(q^{\pi})
\end{aligned}
\tag{25}
$$

up to a constant, where $f_{\mathrm{KL}}(x) := x\log x$ and $f_{\ell}(x) := -\log x$ are convex functions, $\mathcal{H}(\cdot)$ is the Shannon entropy, and $D_{\mathrm{KL}}(\cdot\|\cdot)$ is the KL divergence. $\qquad\square$

*Remark* A.1 (Equivalence of **Lemma 3.3** and Canonical State-Matching). Continuing from (25), one might notice that $J(\pi)$ can be equivalently rewritten as

$$
\begin{aligned}
J(\pi) &= -(1+\beta)\mathbb{E}_{p_{\mathcal{D}}^{\tau}}\left[\frac{q^{\pi}}{p_{\mathcal{D}}^{\tau}}\log\frac{q^{\pi}}{p_{\mathcal{D}}^{\tau}}\right] - \mathbb{E}_{q^{\pi}}\left[-\log q^{\pi}\right] \\
&= -(1+\beta)\mathbb{E}_{p_{\mathcal{D}}^{\tau}}\left[\frac{q^{\pi}}{p_{\mathcal{D}}^{\tau}}\log\frac{q^{\pi}}{p_{\mathcal{D}}^{\tau}}\right] - \mathbb{E}_{p_{\mathcal{D}}^{\tau}}\left[-\frac{q^{\pi}}{p_{\mathcal{D}}^{\tau}}\log q^{\pi}\right] \\
&= -(1+\beta)\mathbb{E}_{p_{\mathcal{D}}^{\tau}}\left[\frac{q^{\pi}}{p_{\mathcal{D}}^{\tau}}\log\frac{q^{\pi}}{p_{\mathcal{D}}^{\tau}}\right] - (1+\beta)\mathbb{E}_{p_{\mathcal{D}}^{\tau}}\left[\frac{q^{\pi}}{p_{\mathcal{D}}^{\tau}}\log(q^{\pi})^{-1/(1+\beta)}\right] \\
&= -(1+\beta)\mathbb{E}_{p_{\mathcal{D}}^{\tau}}\left[\frac{q^{\pi}}{p_{\mathcal{D}}^{\tau}}\log\left(\frac{q^{\pi}}{p_{\mathcal{D}}^{\tau}}\cdot\frac{1}{(q^{\pi})^{1/(1+\beta)}}\right)\right] \\
&= -(1+\beta)\mathbb{E}_{q^{\pi}}\left[\log\frac{(q^{\pi})^{\beta/(1+\beta)}}{p_{\mathcal{D}}^{\tau}}\right]
\end{aligned}
\tag{26}
$$

Assume that there exists a probability distribution $\hat{p}_{\mathcal{D}}^{\tau}(x)$ such that $\hat{p}_{\mathcal{D}}^{\tau}(x) \propto (p_{\mathcal{D}}^{\tau}(x))^{(1+\beta)/\beta}$. Then

$$
J(\pi) \simeq -(1+\beta)\mathbb{E}_{q^{\pi}}\left[\log\left(\frac{q^{\pi}}{\hat{p}_{\mathcal{D}}^{\tau}(x)}\right)^{\beta/(1+\beta)}\right] = -\beta\mathbb{E}_{\hat{p}_{\mathcal{D}}^{\tau}}\left[\frac{q^{\pi}}{\hat{p}_{\mathcal{D}}^{\tau}}\log\frac{q^{\pi}}{\hat{p}_{\mathcal{D}}^{\tau}(x)}\right] = -\beta D_{\mathrm{KL}}(q^{\pi}\|\hat{p}_{\mathcal{D}}^{\tau})
\tag{27}
$$

In other words, the optimization objective considered in (5) and in **Lemma 3.3** is equivalent to a pure state-matching objective $-\beta D_{\mathrm{KL}}(q^{\pi}\|\hat{p}_{\mathcal{D}}^{\tau})$ predicated on the existence of a 'rescaled' probability distribution $\hat{p}_{\mathcal{D}}^{\tau}(x)$ as defined above.

**Lemma 3.4.** (Explicit Dual Function of (7)) Consider the primal problem

$$
\begin{aligned}
\max_{\pi\in\Pi}\quad & J(\pi) \simeq -\mathcal{H}(q^{\pi}) - (1+\beta)D_{\mathrm{KL}}(q^{\pi}\|p_{\mathcal{D}}^{\tau}) \\
\text{s.t.}\quad & \mathbb{E}_{p_{\mathcal{D}}^{\tau}}[c^{*}(x)] - \mathbb{E}_{q^{\pi}}[c^{*}(x)] \leq W_{0}
\end{aligned}
\tag{28}
$$

The Lagrangian dual function $g(\lambda)$ is bounded from below by the function $g_{\ell}(\lambda)$ given by

$$
g_{\ell}(\lambda) := \beta\left[\lambda(\mathbb{E}_{p_{\mathcal{D}}^{\tau}}c^{*}(x) - W_{0}) - \mathbb{E}_{p_{\mathcal{D}}^{\tau}}e^{\lambda c^{*}(x)-1}\right]
\tag{29}
$$

*Proof.* Define $f_{\mathrm{KL}}(u) := u\log u$. From (10), the dual function $g(\lambda) : \mathbb{R}_{+} \to \mathbb{R}$ is given by

$$
\begin{aligned}
g(\lambda) &:= \min_{\pi\in\Pi}\left[(1+\beta)\mathbb{E}_{p_{\mathcal{D}}^{\tau}}f_{\mathrm{KL}}\left(\frac{q^{\pi}}{p_{\mathcal{D}}^{\tau}}\right) - \mathbb{E}_{q^{\pi}}\log(q^{\pi}) + \beta\lambda\left(\mathbb{E}_{p_{\mathcal{D}}^{\tau}}c^{*}(x) - \mathbb{E}_{q^{\pi}}c^{*}(x) - W_{0}\right)\right] \\
&= \min_{\pi\in\Pi}\left[(1+\beta)\mathbb{E}_{p_{\mathcal{D}}^{\tau}}f_{\mathrm{KL}}\left(\frac{q^{\pi}}{p_{\mathcal{D}}^{\tau}}\right) - \left(\mathbb{E}_{p_{\mathcal{D}}^{\tau}}f_{\mathrm{KL}}\left(\frac{q^{\pi}}{p_{\mathcal{D}}^{\tau}}\right) + \mathbb{E}_{q^{\pi}}\log p_{\mathcal{D}}^{\tau}\right) + \beta\lambda\left(\mathbb{E}_{p_{\mathcal{D}}^{\tau}}c^{*}(x) - \mathbb{E}_{q^{\pi}}c^{*}(x) - W_{0}\right)\right] \\
&= \min_{\pi\in\Pi}\left[\beta\mathbb{E}_{p_{\mathcal{D}}^{\tau}}f_{\mathrm{KL}}\left(\frac{q^{\pi}}{p_{\mathcal{D}}^{\tau}}\right) - \mathbb{E}_{q^{\pi}}\log p_{\mathcal{D}}^{\tau} + \beta\lambda\left(\mathbb{E}_{p_{\mathcal{D}}^{\tau}}c^{*}(x) - \mathbb{E}_{q^{\pi}}c^{*}(x) - W_{0}\right)\right]
\end{aligned}
\tag{30}
$$

where we define $\beta\lambda \in \mathbb{R}_{+}$ as the Lagrangian multiplier associated with the constraint in (7). Rearranging terms,

$$
g(\lambda) = \min_{\pi\in\Pi}\left[\beta\mathbb{E}_{p_{\mathcal{D}}^{\tau}}\left[-\left(\lambda c^{*}\cdot\frac{q^{\pi}}{p_{\mathcal{D}}^{\tau}}\right) + f_{\mathrm{KL}}\left(\frac{q^{\pi}}{p_{\mathcal{D}}^{\tau}}\right)\right] - \mathbb{E}_{q^{\pi}}\log p_{\mathcal{D}}^{\tau} + \beta\lambda\mathbb{E}_{p_{\mathcal{D}}^{\tau}}c^{*}(x) - \beta\lambda W_{0}\right]
\tag{31}
$$

Because the sum of function minima is a lower bound on the minima of the sum of the functions themselves, we have

$$
\begin{aligned}
g(\lambda) &\geq \beta \mathbb{E}_{p_{\mathcal{D}}^{\tau}} \min_{\pi \in \Pi} \left[ -\left( \lambda c^* \cdot \frac{q^{\pi}}{p_{\mathcal{D}}^{\tau}} \right) + f_{\mathrm{KL}} \left( \frac{q^{\pi}}{p_{\mathcal{D}}} \right) \right] - \max_{\pi \in \Pi} \left[ \mathbb{E}_{q^{\pi}} \log p_{\mathcal{D}}^{\tau} \right] + \min_{\pi \in \Pi} \left[ \beta \lambda \mathbb{E}_{p_{\mathcal{D}}^{\tau}} c^*(x) - \beta \lambda W_0 \right] \\
&\sim \beta \mathbb{E}_{p_{\mathcal{D}}^{\tau}} \min_{\pi \in \Pi} \left[ -\left( \lambda c^* \cdot \frac{q^{\pi}}{p_{\mathcal{D}}^{\tau}} \right) + f_{\mathrm{KL}} \left( \frac{q^{\pi}}{p_{\mathcal{D}}} \right) \right] + \beta \lambda \mathbb{E}_{p_{\mathcal{D}}^{\tau}} c^*(x) - \beta \lambda W_0
\end{aligned}
\tag{32}
$$

ignoring the term $\max_{\pi \in \Pi} \left[ \mathbb{E}_{q^{\pi}} \log p_{\mathcal{D}}^{\tau} \right]$ that is constant with respect to $\lambda$. In general, simplifying (32) is challenging if not intractable. Instead, we note that minimizing over the set of admissible policies $\Pi$ achieves an optimum that is lower bounded by minimizing over the superset

$$
\begin{aligned}
g(\lambda) &\geq \beta \mathbb{E}_{p_{\mathcal{D}}^{\tau}} \min_{z \in \mathbb{R}_+} \left[ -(\lambda c^*(x) \cdot z) + f_{\mathrm{KL}}(z) \right] + \beta \lambda \mathbb{E}_{p_{\mathcal{D}}^{\tau}} c^*(x) - \beta \lambda W_0 \\
&= \beta \left[ -\mathbb{E}_{p_{\mathcal{D}}^{\tau}} f_{\mathrm{KL}}^{\star}(\lambda c^*(x)) + \lambda (\mathbb{E}_{p_{\mathcal{D}}^{\tau}} c^*(x) - W_0) \right]
\end{aligned}
\tag{33}
$$

where $f^{\star}(\cdot)$ is the Fenchel conjugate of a convex function $f(\cdot)$. The Fenchel conjugate of $f_{\mathrm{KL}}(u) = u \log u$ is $f_{\mathrm{KL}}^{\star}(v) = e^{v-1}$ following Borwein & Lewis (2006), and so

$$
g(\lambda) \geq \beta \left[ -\mathbb{E}_{p_{\mathcal{D}}^{\tau}} e^{\lambda c^*(x)-1} + \lambda (\mathbb{E}_{p_{\mathcal{D}}^{\tau}} c^*(x) - W_0) \right]
\tag{34}
$$

Define the right hand side of this inequality as the function $g_{\ell}(\lambda)$ and the result is immediate. $\square$

## B. Additional Implementation Details

**Oracle Functions for Optimization Tasks.** The task-specific oracle reward functions $r(x)$ are developed by domain experts and assumed to exactly return the noiseless reward of all possible input designs in the search space $\mathcal{X}$. The oracle functions associated wit tasks from the Design-Bench MBO evaluation suite are detailed by the original Design-Bench authors in Trabucco et al. (2022); briefly, the **TFBind8** (i.e., `TFBind8-Exact-v0` in Design-Bench) task uses the oracle function from Barrera et al. (2016); the **UTR** (`UTR-ResNet-v0`) task uses the oracle function from Angermüeller et al. (2020); the **ChEMBL** (`ChEMBL_MCHC_CHEMBL3885882_MorganFingerprint-RandomForest-v0`) task uses the oracle function from Trabucco et al. (2022); the **Superconductor** (`Superconductor-RandomForest-v0`) task uses the oracle function from Hamidieh (2018); and the **D'Kitty** (`DKittyMorphology-Exact-v0`) task uses a MuJoCo (Todorov et al., 2012) simulation environment and learned control policy from Trabucco et al. (2022) to evaluate input designs. The **Molecule** task uses the oracle function from Wildman & Crippen (1999).

**Data Preprocessing.** For all experiments, we follow Mashkaria et al. (2023) and normalize the objective values both in the offline dataset $\mathcal{D}$ and in those reported in **Section 5** according to:

$$
y = \frac{\hat{y} - y_{\min}}{y_{\max} - y_{\min}}
\tag{35}
$$

where $\hat{y} = r(x)$ is the original unnormalized oracle value for an input design $x$, and $y_{\max}$ (resp., $y_{\min}$) is the maximum (resp., minimum) value in the full offline dataset. A reported value of $y > 1$ means that an offline optimization experiment proposed a candidate design better than the best design in the offline dataset. Note that in many of the MBO tasks, the publicly available offline dataset $\mathcal{D}$ is only a subset of the designs in the full offline dataset; it is therefore possible (and frequently the case) that $\max_{y \in \mathcal{D}} y < 1$ in our MBO tasks.

As introduced in the main text, we learn a VAE (Kingma & Welling, 2014) model to encode and decode designs for discrete optimization tasks to and from a continuous latent space, and perform our optimization experiments over the continuous VAE latent space. Following prior work (Maus et al., 2022; Tripp et al., 2020; Yao et al., 2024), we co-train a Transformer-based VAE autoencoder (consisting of an encoder $e_{\varphi} : \hat{\mathcal{X}} \to \mathcal{X}$ parameterized by $\varphi^*$ and decoder $d_{\phi} : \mathcal{X} : \mathcal{X} \to \hat{\mathcal{X}}$ parameterized by $\gamma^*$) with the surrogate model $r_{\theta} : \mathcal{X} \to \mathbb{R}$ (parameterized by $\theta^*$) according to

$$
\theta^*, \varphi^*, \phi^* = \underset{(\theta, \varphi, \phi) \in \Theta \times \Gamma \times \Phi}{\arg \min} \mathbb{E}_{(x, r(x)) \sim \mathcal{D}} \left[ -\log d_{\phi}(x | e_{\varphi}(x)) + \beta D_{\mathrm{KL}}(\mathcal{N}(0, I) || e_{\varphi}(x)) + \alpha || r_{\theta}(e_{\varphi}(x)) - r(x) ||_2^2 \right]
\tag{36}
$$

where $\mathcal{N}(0, I)$ is the standard multivariate normal prior and $\alpha = 1$, $\beta = 10^{-4}$ are constant hyperparameters. We can then perform optimization against $r_{\theta}$ trained on the 256-dimensional continuous latent space of the VAE, and then decode the

candidate designs using $d_\phi(\cdot)$ to derive the corresponding discrete design following prior work from Maus et al. (2022); Gómez-Bombarelli et al. (2018). We again use an Adam optimizer with a learning rate of $\eta = 3 \times 10^{-4}$ for both the VAE and the forward surrogate. In this way, the search space for our discrete tasks becomes the $\mathcal{X} \subseteq \mathbb{R}^d$ for $d = 256$, the surrogate model is simply $r_\theta : \mathcal{X} \to \mathbb{R}$, and the reward function $r : \mathcal{X} \to \mathbb{R}$ is now

$$r(x) := \mathbb{E}_{\hat{x} \sim d_\phi(\hat{x}|x)}[\hat{r}(\hat{x})] \tag{37}$$

where $\hat{r} : \hat{\mathcal{X}} \to \mathbb{R}$ is the original expert oracle reward function over the discretized input space $\hat{\mathcal{X}}$, and $r(x)$ is the corresponding oracle reward function that accepts our continuous inputs from $\mathcal{X}$ as input. Note that for the MBO tasks over continuous search spaces (i.e., the **Superconductor** and **D'Kitty** tasks), we treat $\mathcal{X} = \hat{\mathcal{X}}$ and fix both the encoder $e_\varphi$ and decoder $d_\phi$ to be the identity functions, as no transformation to a separate continuous search space is necessary.

**Optimization Experiments.** All baseline methods run evaluated using their official open-source implementations made publicly available by the respective authors. In DynAMO, we initialize all optimizers using the first $b$ elements from a $d$-dimensional scrambled Sobol sequence (Sobol, 1967) using the official PyTorch quasi-random generator `SobolEngine` implementation, where $b$ is the sampling batch size and $d$ is the dimensionality of the search space. Note that the Sobol sequence only returns points with dimensions between 0 and 1; for each task, we therefore un-normalize the sampled Sobol points $\tilde{x}_0$ according to $x_0 = x_{\min} + (\tilde{x}_0 \cdot (x_{\max} - x_{\min}))$, where $x_{\max}, x_{\min}$ are the maximum and minimum bounds on the search space for our experiments, respectively. We fix $x_{\min} = -4.0$ and $x_{\max} = +4.0$ for all $d$ dimensions across all tasks.

In all experiments reported in **Table 1**, each optimizer continues to sample from the search space in batched acquisitions of $b$ samples—we set $b = 64$ for all our experiments unless otherwise stated. After each acquisition, we score the sampled designs using the (penalized) forward surrogate model (i.e., the Lagrangian in (9) for DynAMO). If the maximum prediction from the recently sampled batch is not at least as optimal as the maximum prediction of the previously sampled designs, then we define the acquisition step as a *failure*; a sequence of 10 consecutive failures triggers a *restart* in the optimization process where the optimizer starts from the scratch beginning with sampling with the Sobol sequence to initialize the optimizer as described above. After 3 restarts are triggered, we consider the optimization process terminated, and all designs across all restarts are aggregated to choose the top $k = 128$ final candidate designs to be evaluated using the oracle reward function.

**Excluded Baselines.** We exclude Boosting offline Optimizers with Surrogate Sensitivity (BOSS) from Dao et al. (2024) and Normalized maximum likelihood Estimation for Model-based Optimization (NEMO) from Fu & Levine (2021) from our experiments because they do not have open-source implementations.

**Excluded Optimization Tasks.** In our experiments, we primarily evaluate DynAMO and baseline methods on optimization tasks from Design Bench, a suite of offline MBO tasks introduced by Trabucco et al. (2022). The following tasks from the original authors were excluded from our experiments: (1) **Ant** Morphology, excluded due to reproducibility issues as per GitHub Issues Link and OpenReview Discussion; (2) **Hopper** Controller, excluded due to errors in the original open-source implementation per GitHub Issues Link and prior work (Tan et al., 2024; Mashkaria et al., 2023); (3) **NAS** (Neural Architecture Search) on CIFAR10, excluded due to its prohibitively expensive computational cost for evaluating the oracle function as noted in prior work (Tan et al., 2024; Yu et al., 2021; Fu & Levine, 2021; Nguyen et al., 2023); and (4) **TFBind10**, excluded due to its domain and experimental similarity with the **TFBind8** task already included in our evaluation suite. We augment our evaluation suite with the **UTR** task from Trabucco et al. (2022) and the **Molecule** task from Yao et al. (2024); Brown et al. (2019) to provide a comprehensive experimental evaluation of DynAMO and baseline methods across a wide variety of scientific domains.

## C. Additional Background and Preliminaries

### C.1. $f$-Divergence and Fenchel Conjugates

**Definition C.1** ($f$-Divergence). Suppose we are given two probability distributions $P(x), Q(x)$ defined over a common support $\mathcal{X}$. For any continuous, convex function $f : \mathbb{R}_+ \to \mathbb{R}$ that is finite over $\mathbb{R}_{++}$, we define the *f-divergence* between $P(x), Q(x)$ as

$$D_f(Q(x)||P(x)) := \mathbb{E}_{x \sim P(x)}\left[f\left(\frac{Q(x)}{P(x)}\right)\right] \tag{38}$$

We refer to $f$ as the *generator* of $D_f(\cdot||\cdot)$. Two commonly used $f$-divergences are the Kullback-Leibler (KL)-Divergence (defined by the generator $f_{KL}(u) = u \log u$) and the $\chi^2$-Divergence (defined by the generator $f_{\chi^2}(u) = (u - 1)^2/2$).

**Definition C.2** (Fenchel Conjugate). The *Fenchel conjugate* (i.e., Legendre-Fenchel transform) of a function $f : \mathcal{U} \to \mathbb{R}$ is defined as

$$f^{\star}(v) := -\inf\ \{-\langle u, v \rangle + f(u) \mid u \in \mathcal{U}\} \tag{39}$$

where $\langle u, v \rangle$ is the inner product, and $f^{\star} : \mathcal{V} \to \mathbb{R}$ is the Fenchel conjugate defined over the *dual space* $\mathcal{V}$ of $\mathcal{U}$. Importantly, the Fenchel conjugate function is guaranteed to always be convex Borwein & Lewis (2006) regardless of the (non-)convexity of the original function $f$. This allows us to make important convergence guarantees in **Appendix D.4** in solving the Lagrangian dual problem in **Algorithm 1**. Fenchel conjugates are commonly used in optimization problems to rewrite difficult primal problems into more tractable dual formulations Ma et al. (2022); Borwein & Lewis (2006); Agrawal & Horel (2021)—we leverage a similar technique in our work in **Algorithm 1**.

**Lemma C.3** (Fenchel Conjugate of the KL-Divergence Generator Function). *Recall that the generator function of the KL-divergence is $f_{KL}(u) := u \log u$ for $u \in \mathbb{R}_{++}$. The Fenchel conjugate of this generator is $f^{\star}_{KL}(v) = e^{v-1}$.*

*Proof.* The proof follows immediately from the definition of the Fenchel conjugate in (39).

$$f^{\star}_{\text{KL}}(v) := \sup\ \{uv - u \log u \mid u \in \mathbb{R}_{++}\} \tag{40}$$

We differentiate the argument on the right hand side with respect to $u$ to find the supremum given a particular $v \in \mathcal{V}$:

$$\frac{\partial}{\partial u} [uv - u \log u] \Big|_{u=u^*} = v - \log u^* - 1 = 0 \to u^* = e^{v-1} \tag{41}$$

It is easy to verify that $u^*$ is a maxima. Plugging this result into (40),

$$f^{\star}_{\text{KL}}(v) = u^* v - u^* \log u^* = v e^{v-1} - (v-1) e^{v-1} = e^{v-1} \tag{42}$$

$\square$

**Lemma C.4** (Fenchel Conjugate of the $\chi^2$-Divergence Generator Function). *Recall that the generator function of the $\chi^2$-divergence is $f_{\chi^2}(u) := \frac{1}{2}(u-1)^2$ for $u \in \mathbb{R}_{++}$. The Fenchel conjugate of this generator is $f^{\star}_{\chi^2}(v) = \frac{v^2}{2} + v$.*

*Proof.* The proof follows immediately from the definition of the Fenchel conjugate in (39).

$$f^{\star}_{\chi^2}(v) := \sup\ \left\{uv - \frac{1}{2}(u-1)^2 \mid u \in \mathbb{R}_{++}\right\} \tag{43}$$

We differentiate the argument on the right hand side with respect to $u$ to find the supremum given a particular $v \in \mathcal{V}$:

$$\frac{\partial}{\partial u} \left[uv - \frac{1}{2}(u-1)^2\right] \Big|_{u=u^*} = v - u^* + 1 = 0 \to u^* = v + 1 \tag{44}$$

It is easy to verify that $u^*$ is a maxima. Plugging this result into (43),

$$f^{\star}_{\chi^2}(v) = u^* v - \frac{1}{2}(u^* - 1)^2 = (v+1)v - \frac{1}{2}((v+1)-1)^2 = v^2 + v - \frac{1}{2}v^2 = \frac{v^2}{2} + v \tag{45}$$

$\square$

Additional details and related technical discussion are offered by Borwein & Lewis (2006); Nachum & Dai (2020); Ma et al. (2022); Amos (2023); and Terjék & González-Sánchez (2022).

## C.2. Constrained Optimization via Lagrangian Duality

In our problem formulation in (7) for DynAMO, we reformulate any naïve MBO problem as a separate *constrained optimization* problem, which is generally of the form

$$\begin{aligned} \text{minimize}_{x \in \mathcal{X}} \quad & f(x) \\ \text{subject to} \quad & f_i(x) \le 0 \quad \forall i \in \{1, \ldots, m\} \end{aligned} \tag{46}$$

given a set of $m$ constraints. In general, satisfying any arbitrary set of (potentially nonlinear) constraints is challenging if not intractable, and it is often desirable instead to solve a related *unconstrained* optimization problem. One common mechanism to perform such a problem transformation is to define the *Lagrangian* of (46) as

$$\mathcal{L}(x; \vec{\lambda}) = f(x) + \langle \vec{\lambda}, \begin{bmatrix} f_1(x) & f_2(x) & \cdots & f_m(x) \end{bmatrix} \rangle \tag{47}$$

where $\vec{\lambda} \in \mathbb{R}_+^m$ and $\mathcal{L} : \mathcal{X} \times \mathbb{R}_+^m \to \mathbb{R}$ is a real-valued function. It can be shown (Boyd & Vandenberghe, 2004) that the constrained optimization problem in (46) is equivalent to the unconstrained problem

$$\text{minimize}_{x \in \mathcal{X}} \quad \text{maximize}_{\vec{\lambda} \in \mathbb{R}_+^m} \mathcal{L}(x; \vec{\lambda}) \tag{48}$$

in terms of the Lagrangian, where $\succeq$ represents an element-wise inequality. The *dual problem* of (48) is constructed by reversing the order of the minimization and maximization problems:

$$\text{maximize}_{\vec{\lambda} \in \mathbb{R}_+^m} \quad \text{minimize}_{x \in \mathcal{X}} \mathcal{L}(x; \vec{\lambda}) = \text{maximize}_{\vec{\lambda} \in \mathbb{R}_+^m} g(\vec{\lambda}) \tag{49}$$

where we implicitly define the *dual function* $g(\vec{\lambda}) := \min_{x \in \mathcal{X}} \mathcal{L}(x; \vec{\lambda})$. In general, it is guaranteed that the optimal solution to the dual problem in (49) is a lower bound for the optimal solution for the original problem in (46) from weak duality; if $f(x)$ and $f_i(x)$ are convex and bounded from below such that Slater's condition applies, then strong duality guarantees that the optimal solutions to the dual and original problems are equal.

As an additional remark, we note that in our problem formulation in (7), Slater's condition is not satisfied as both the surrogate reward function $r_\theta(x)$ and adversarial source critic $c^*(x)$ may be arbitarily non-convex. However, we find empirically that the guarantee of weak duality is sufficient to make the approach of Lagrangian duality both tractable and effective in solving (7) to give us DynAMO.

Furthermore, we also find that solving the dual optimization problem in (49) requires us to first solve for the dual function $g(\vec{\lambda})$—this is a challenging task in general, and prior work has attempted to either approximate $g(\vec{\lambda})$ under specific assumptions on the search space $\mathcal{X}$ (Yao et al., 2024) or forego solving for $g(\vec{\lambda})$ entirely by instead treating $\vec{\lambda}$ as a hyperparameter to be manually tuned or set heuristically (Trabucco et al., 2021; Yu et al., 2021; Chen et al., 2023c). In our work, we show how penalizing the optimization objective via a KL-divergence term as in (7) is sufficient to yield an *exact* solution for the dual function $g(\vec{\lambda})$ (see **Lemma 3.4**). This is advantageous because it can be shown (Boyd & Vandenberghe, 2004) that $g(\vec{\lambda})$ is convex; assuming that the gradient of $g(\vec{\lambda})$ has a bounded Lipschitz constant, we can therefore arrive in an $\varepsilon$-neighborhood around the optimal $\vec{\lambda}^*$ within $\mathcal{O}(1/\varepsilon)$ time (Bubeck, 2015; Grimmer et al., 2023; Zhang et al., 2019). We leverage this convergence guarantee to solve for $\vec{\lambda}^*$ naïvely via gradient ascent in DynAMO (see **Algorithm 1**).

### C.3. Wasserstein Distance and Optimal Transport

In our motivating problem formulation in (7), we introduce a constrained optimization problem where the constraint is a function of a source critic $c^* : \mathcal{X} \to \mathbb{R}$ with a bounded Lipschitz norm. In this section, we show how this choice in adversarial source critic is connected to classical theory in optimal transport.

In general, the *p-Wasserstein distance* $W_p(P(x)||Q(x))$ is a distance function between pairs of probability distributions $P(x)$ and $Q(x)$. Given a metric space $(M, d)$, we define the $p$-Wasserstein distance as

$$W_p(P(x)||Q(x)) = \inf_{\gamma \in \Gamma(P,Q)} \left( \mathbb{E}_{(x,x') \sim \gamma} d(x, x')^p \right)^{1/p} \tag{50}$$

for $p \geq 1$, and where $\Gamma(P, Q)$ is the set of all couplings between $P(x), Q(x)$. Intuitively, one can think of the $p$-Wasserstein distance as representing the cost associated with the optimal (i.e., cost-minimizing) strategy of 'transporting' the 'mass' of one probability distribution to another.

For any two arbitrary multidimensional distributions, exactly computing the Wasserstein distance between them is computationally expensive (Pele & Werman, 2009; Watanabe & Isobe, 2025; Cuturi, 2013) and in many cases intractable in practice. Instead, a common technique used by Arjovsky et al. (2017) and others is to leverage the Kantorovich-Rubinstein duality theorem (Kantorovich & Rubinstein, 1958) to exactly rewrite (50) (specifically for the $p = 1$ Wasserstein distance) as

$$W_1(P(x)||Q(x)) = \sup_{||c||_L \leq 1} \left[ \mathbb{E}_{x \sim P(x)} c(x) - \mathbb{E}_{x \sim Q(x)} c(x) \right] \tag{51}$$

where $||c||_L$ is the Lipschitz norm of a *source critic* function $c : \mathcal{X} \to \mathbb{R}$. Intuitively, we can think of the function $c(x)$ as assigning of value of 'in-distribution-ness' relative to $P(x)$: a larger value of $c(x)$ (informally) means that the source critic predicts the input $x$ to be more likely to have been drawn from $P(x)$ as opposed to $Q(x)$. In DynAMO, we follow Arjovsky et al. (2017); Yao et al. (2024); and others to constrain the MBO optimization problem by requiring that the estimated Wasserstein distance between the distribution of generated designs and $\tau$-weighted distribution of designs from the offline dataset is less than a constant $W_0$ according to (51)—see **Algorithm 1** for additional details.

## D. Additional Results

### D.1. Additional Design Quality Results

We supplement the results shown in **Table 1** with the raw Best@128 oracle quality scores reported for each of the 6 tasks in our evaluation suite in **Supplementary Table A1**.

In **Section 4**, we define the **Best@$k$** score to evaluate the quality of observed designs according to a hidden oracle function used for evaluation of candidate fitness. Achieving a high Best@$B = k$ score ensures that a desirable design is found. Consistent with prior work on batched optimization methods (Trabucco et al., 2021; Mashkaria et al., 2023; Krishnamoorthy et al., 2023), we are also interested in the **Median@$k$** score defined as

$$\text{Median@}k(\{x_i^F\}_{i=1}^k) := \text{median}_{1 \leq i \leq k}\, r(x_i^F) \tag{52}$$

to evaluate whether a *batch* of candidate designs (as opposed to any singular design) is generally of high quality according to the oracle $r(x)$. We report the Median@$k$ score for $k = 128$ in **Supplementary Table A2**; in general, we find that DynAMO does not perform as well as other objective-modifying baseline methods according to this metric. However, we note that in many offline optimization applications, we are often not as interested in how the median design performs, but rather if we are able to discover optimal and near-optimal designs. For this reason, we chose to focus on the Best@128 oracle scores in **Table 1** to evaluate the quality of designs proposed by an optimizer in our main results. Nonetheless, future work may explore how to better tune DynAMO (e.g., the $\tau$ and $\beta$ hyperparameters in **Algorithm 1**) to achieve more desirable Median@128 scores.

### D.2. Additional Design Diversity Results

We supplement the results shown in **Table 1** with the raw Pairwise Diversity scores reported for each of the 6 tasks in our evaluation suite in **Supplementary Table A1**.

In **Section 4**, we describe the **Pairwise Diversity** metric previously used in prior work (Kim et al., 2023; Jain et al., 2022; Maus et al., 2023) to measure the diversity of samples obtained from a given offline optimization method. We can think of Pairwise Diversity as measuring the *between-candidate* diversity of candidates proposed by a generative algorithm. However, this is far from the only relevant definition of diversity; other possible metrics might measure the following:

1. *Candidate-Dataset Diversity*: How ***novel*** is a proposed candidate compared to the real designs previously observed in the offline dataset?

2. *Aggregate Diversity*: How well does the batch of candidate designs collectively cover the possible search space?

To evaluate (1), we follow prior work by Kim et al. (2023) and Jain et al. (2022) and evaluate the **Minimum Novelty (MN)** for a batch of $k$ final proposed candidates with respect to the offline dataset $\mathcal{D}$, defined as

$$\text{MN}(\{x_i^F\}_{i=1}^k; \mathcal{D}) := \mathbb{E}_{x_i^F} \left[ \min_{x \in \mathcal{D}} d(x_i^F, x) \right] \tag{53}$$

where $\mathcal{D}$ is the task-specific dataset of offline sample designs and $x_i^F$ is the $i$th candidate design proposed by an optimization experiment. Following (14), we define the distance function $d(\cdot, \cdot)$ as the normalized Levenshtein edit distance (Haldar & Mukhopadhyay, 2011) (resp., Euclidean distance) for discrete (resp., continuous) tasks.

For (2), we report the $L_1$ **Coverage (L1C)** of the candidate designs, defined as

$$\text{L1C}(\{x_i^F\}_{i=1}^B) := \frac{1}{\dim(x)} \sum_{k=1}^{\dim(x)} \max_{i \neq j} |x_{ik}^F - x_{jk}^F| \tag{54}$$

*Table A1.* **Quality and Diversity of Designs Under MBO Objective Transforms (Full).** We evaluate DynAMO against other MBO objective-modifying methods using six different backbone optimizers. Each cell consists of '**Best@128**/Pairwise Diversity' oracle scores separated by a forward slash. Both metrics are reported mean$^{(95\% \text{ confidence interval})}$ across 10 random seeds, where higher is better. **Dataset** $\mathcal{D}$ reports the maximum oracle score and mean pairwise diversity in the offline dataset. **Bolded** entries indicate overlapping 95% confidence intervals with the best performing algorithm (according to the mean) per optimizer. **Bolded** (resp., Underlined) Rank and Optimality Gap (Opt. Gap) metrics indicate the best (resp., second best) for a given backbone optimizer.

| Grad. | TFBind8 | UTR | ChEMBL | Molecule | Superconductor | D'Kitty | Rank ↓ | Opt. Gap ↑ |
|---|---|---|---|---|---|---|---|---|
| Dataset $\mathcal{D}$ | 43.9/65.9 | 59.4/57.3 | 60.5/60.0 | 88.9/36.7 | 40.0/66.0 | 88.4/85.7 | —/— | —/— |
| Baseline | $90.0^{(4.3)}$/$12.5^{(8.0)}$ | $\mathbf{80.9^{(12.1)}}$/$7.8^{(8.8)}$ | $60.2^{(8.9)}$/$7.9^{(7.8)}$ | $88.8^{(4.0)}$/$24.1^{(13.3)}$ | $36.0^{(6.8)}$/$0.0^{(0.0)}$ | $65.6^{(14.5)}$/$0.0^{(0.0)}$ | 5.0/5.5 | 6.8/-53.2 |
| COMs⁻ | $60.4^{(9.8)}$/$10.4^{(8.7)}$ | $60.2^{(12.4)}$/$7.5^{(9.2)}$ | $60.2^{(8.8)}$/$7.9^{(7.5)}$ | $88.4^{(4.0)}$/$24.8^{(10.0)}$ | $22.5^{(3.2)}$/$0.0^{(0.0)}$ | $71.2^{(10.7)}$/$0.0^{(0.0)}$ | 7.3/6.5 | -3.0/-53.5 |
| COMs⁺ | $93.1^{(3.4)}$/$\mathbf{66.6^{(1.0)}}$ | $67.0^{(0.9)}$/$57.4^{(0.2)}$ | $\mathbf{64.6^{(1.0)}}$/$\mathbf{81.6^{(4.9)}}$ | $\mathbf{97.1^{(1.6)}}$/$3.8^{(0.9)}$ | $41.2^{(4.8)}$/$99.5^{(2.6)}$ | $91.8^{(0.9)}$/$21.1^{(23.5)}$ | 2.5/2.8 | 12.3/-6.9 |
| RoMA⁻ | $62.0^{(10.7)}$/$12.3^{(8.3)}$ | $60.9^{(12.1)}$/$7.9^{(8.9)}$ | $60.2^{(8.8)}$/$7.7^{(7.6)}$ | $88.8^{(4.0)}$/$24.2^{(13.3)}$ | $36.0^{(6.8)}$/$0.0^{(0.0)}$ | $65.6^{(14.5)}$/$0.0^{(0.0)}$ | 6.7/5.7 | -1.2/-53.3 |
| RoMA⁺ | $66.5^{(0.0)}$/$20.3^{(0.7)}$ | $77.8^{(0.0)}$/$3.8^{(0.0)}$ | $63.3^{(0.0)}$/$6.2^{(0.0)}$ | $84.5^{(0.0)}$/$1.8^{(0.0)}$ | $\mathbf{49.0^{(1.6)}}$/$54.1^{(1.4)}$ | $\mathbf{95.2^{(1.2)}}$/$4.9^{(0.0)}$ | 3.8/5.8 | 9.2/-46.8 |
| ROMO | $\mathbf{98.1^{(0.7)}}$/$62.1^{(0.8)}$ | $66.8^{(1.0)}$/$57.1^{(0.1)}$ | $63.0^{(0.8)}$/$53.9^{(0.6)}$ | $91.8^{(0.9)}$/$48.7^{(0.1)}$ | $38.7^{(2.5)}$/$51.7^{(3.2)}$ | $87.8^{(0.9)}$/$22.1^{(5.5)}$ | 4.2/2.8 | 10.9/-12.7 |
| GAMBO | $73.1^{(12.8)}$/$17.3^{(12.8)}$ | $\mathbf{77.1^{(9.6)}}$/$11.2^{(10.3)}$ | $64.4^{(1.5)}$/$6.9^{(7.7)}$ | $\mathbf{92.8^{(8.0)}}$/$22.1^{(10.5)}$ | $\mathbf{46.0^{(6.8)}}$/$0.0^{(0.0)}$ | $90.6^{(14.5)}$/$1.5^{(3.2)}$ | 3.2/5.3 | 10.5/-52.1 |
| DynAMO | $90.3^{(4.7)}$/$\mathbf{66.9^{(6.9)}}$ | $\mathbf{86.2^{(0.0)}}$/$\mathbf{68.2^{(1.8)}}$ | $64.4^{(2.5)}$/$\mathbf{77.2^{(2.2)}}$ | $91.2^{(0.0)}$/$\mathbf{93.0^{(1.2)}}$ | $44.2^{(7.8)}$/$\mathbf{129^{(5.5)}}$ | $89.8^{(3.2)}$/$\mathbf{104^{(5.6)}}$ | **2.8**/**1.2** | **14.2**/**27.8** |

| Adam | TFBind8 | UTR | ChEMBL | Molecule | Superconductor | D'Kitty | Rank ↓ | Opt. Gap ↑ |
|---|---|---|---|---|---|---|---|---|
| Baseline | $62.9^{(13.0)}$/$12.0^{(12.3)}$ | $69.7^{(10.5)}$/$11.0^{(12.1)}$ | $\mathbf{62.9^{(1.9)}}$/$4.8^{(3.8)}$ | $92.3^{(8.9)}$/$16.8^{(12.4)}$ | $37.8^{(6.3)}$/$6.4^{(14.5)}$ | $58.4^{(18.5)}$/$6.2^{(14.0)}$ | 4.5/6.0 | 0.5/-52.4 |
| COMs⁻ | $62.9^{(13.0)}$/$13.6^{(12.2)}$ | $65.1^{(11.0)}$/$11.0^{(10.6)}$ | $\mathbf{62.9^{(1.9)}}$/$5.0^{(3.8)}$ | $92.4^{(1.0)}$/$21.2^{(18.2)}$ | $22.5^{(3.2)}$/$0.0^{(0.0)}$ | $57.3^{(19.5)}$/$6.3^{(14.3)}$ | 6.0/5.3 | -3.0/-52.4 |
| COMs⁺ | $\mathbf{95.6^{(2.6)}}$/$44.2^{(1.5)}$ | $67.1^{(0.6)}$/$57.4^{(0.2)}$ | $64.6^{(0.9)}$/$\mathbf{81.5^{(5.7)}}$ | $95.3^{(1.9)}$/$3.7^{(1.3)}$ | $39.6^{(5.8)}$/$79.3^{(3.3)}$ | $67.1^{(9.5)}$/$31.8^{(34.5)}$ | 3.2/3.0 | 8.1/-12.3 |
| RoMA⁻ | $62.9^{(13.0)}$/$12.3^{(12.4)}$ | $69.7^{(10.5)}$/$10.9^{(12.0)}$ | $\mathbf{62.9^{(1.9)}}$/$4.7^{(3.8)}$ | $84.7^{(0.0)}$/$16.8^{(12.4)}$ | $37.8^{(6.3)}$/$6.4^{(14.5)}$ | $58.4^{(1.9)}$/$6.2^{(14.0)}$ | 4.5/6.3 | 0.5/-52.4 |
| RoMA⁺ | $96.5^{(0.0)}$/$21.3^{(0.3)}$ | $77.8^{(0.0)}$/$3.8^{(0.0)}$ | $63.3^{(0.0)}$/$5.9^{(0.2)}$ | $92.3^{(8.9)}$/$1.8^{(0.0)}$ | $\mathbf{49.8^{(1.4)}}$/$49.4^{(6.1)}$ | $95.7^{(1.6)}$/$14.8^{(0.6)}$ | 2.8/5.2 | **14.5**/-45.8 |
| ROMO | $95.6^{(0.0)}$/$\mathbf{55.7^{(0.3)}}$ | $67.0^{(0.2)}$/$56.3^{(0.1)}$ | $63.3^{(0.0)}$/$53.5^{(0.1)}$ | $90.4^{(0.0)}$/$50.7^{(0.0)}$ | $31.8^{(3.1)}$/$25.5^{(20.3)}$ | $71.0^{(0.6)}$/$7.2^{(3.9)}$ | 4.8/2.8 | 6.4/-20.5 |
| GAMBO | $94.0^{(2.2)}$/$15.1^{(1.2)}$ | $60.0^{(12.6)}$/$10.3^{(11.5)}$ | $60.9^{(8.7)}$/$12.1^{(11.3)}$ | $\mathbf{91.4^{(6.3)}}$/$19.6^{(15.2)}$ | $37.8^{(6.3)}$/$0.3^{(0.8)}$ | $\mathbf{88.4^{(13.8)}}$/$2.6^{(3.9)}$ | 5.3/5.8 | 8.6/-51.9 |
| DynAMO | $95.2^{(1.7)}$/$\mathbf{54.8^{(8.9)}}$ | $\mathbf{86.2^{(0.0)}}$/$\mathbf{72.3^{(3.4)}}$ | $65.2^{(1.1)}$/$\mathbf{84.8^{(9.2)}}$ | $91.2^{(0.0)}$/$\mathbf{89.9^{(5.3)}}$ | $45.5^{(5.7)}$/$\mathbf{158^{(37.3)}}$ | $84.9^{(12.0)}$/$\mathbf{126^{(5.7)}}$ | **2.8**/**1.2** | **14.5**/**35.7** |

| CMA-ES | TFBind8 | UTR | ChEMBL | Molecule | Superconductor | D'Kitty | Rank ↓ | Opt. Gap ↑ |
|---|---|---|---|---|---|---|---|---|
| Baseline | $87.6^{(8.3)}$/$47.2^{(11.2)}$ | $86.2^{(0.0)}$/$44.6^{(15.9)}$ | $66.1^{(1.0)}$/$93.5^{(2.0)}$ | $106^{(5.9)}$/$66.2^{(9.4)}$ | $49.0^{(1.0)}$/$12.8^{(0.6)}$ | $72.2^{(0.1)}$/$164^{(10.6)}$ | 3.7/3.8 | 14.4/9.5 |
| COMs⁻ | $75.6^{(10.2)}$/$46.0^{(17.6)}$ | $85.7^{(1.3)}$/$56.2^{(15.8)}$ | $64.8^{(1.0)}$/$63.1^{(23.0)}$ | $119^{(3.3)}$/$58.8^{(24.2)}$ | $18.8^{(7.9)}$/$22.0^{(7.9)}$ | $62.9^{(2.1)}$/$67.2^{(8.0)}$ | 5.7/5.2 | 7.6/-9.7 |
| COMs⁺ | $68.0^{(6.0)}$/$24.8^{(11.3)}$ | $77.2^{(9.7)}$/$35.4^{(16.5)}$ | $63.6^{(0.5)}$/$36.7^{(9.0)}$ | $116^{(5.6)}$/$45.8^{(16.1)}$ | $36.8^{(3.5)}$/$0.0^{(0.0)}$ | $62.2^{(15.5)}$/$0.0^{(0.0)}$ | 7.3/7.8 | 7.1/-38.2 |
| RoMA⁻ | $\mathbf{87.6^{(8.3)}}$/$46.7^{(11.1)}$ | $\mathbf{86.2^{(0.0)}}$/$44.8^{(15.8)}$ | $\mathbf{66.1^{(1.0)}}$/$\mathbf{93.5^{(2.1)}}$ | $106^{(5.9)}$/$66.2^{(9.4)}$ | $\mathbf{49.0^{(1.0)}}$/$12.8^{(0.6)}$ | $72.2^{(0.1)}$/$164^{(10.6)}$ | 3.7/3.5 | 14.4/9.5 |
| RoMA⁺ | $85.9^{(7.0)}$/$53.1^{(15.0)}$ | $79.8^{(3.7)}$/$31.9^{(15.2)}$ | $64.6^{(1.1)}$/$60.5^{(14.9)}$ | $118^{(6.6)}$/$63.7^{(21.6)}$ | $44.6^{(3.2)}$/$92.8^{(18.9)}$ | $71.2^{(0.1)}$/$112^{(86.4)}$ | 5.0/4.8 | 14.1/8.0 |
| ROMO | $\mathbf{88.3^{(6.0)}}$/$57.5^{(11.6)}$ | $\mathbf{86.2^{(0.0)}}$/$40.2^{(13.1)}$ | $64.5^{(0.9)}$/$66.5^{(13.5)}$ | $113^{(6.0)}$/$70.2^{(11.5)}$ | $45.7^{(1.3)}$/$97.7^{(15.4)}$ | $77.3^{(3.2)}$/$20.9^{(40.9)}$ | 4.2/4.3 | 15.7/-3.1 |
| GAMBO | $90.4^{(4.4)}$/$39.6^{(15.5)}$ | $\mathbf{86.2^{(0.0)}}$/$53.4^{(8.4)}$ | $\mathbf{66.2^{(1.6)}}$/$84.8^{(4.8)}$ | $\mathbf{121^{(0.0)}}$/$61.3^{(14.6)}$ | $45.2^{(3.5)}$/$\mathbf{173^{(19.4)}}$ | $72.2^{(0.1)}$/$59.9^{(19.6)}$ | **2.2**/4.3 | 16.7/16.8 |
| DynAMO | $89.8^{(3.6)}$/$\mathbf{73.6^{(0.6)}}$ | $85.7^{(5.8)}$/$\mathbf{73.1^{(3.1)}}$ | $63.9^{(0.9)}$/$72.0^{(3.1)}$ | $117^{(6.7)}$/$\mathbf{94.0^{(0.5)}}$ | $\mathbf{50.6^{(4.8)}}$/$97.8^{(13.2)}$ | $78.5^{(5.5)}$/$\mathbf{292^{(83.5)}}$ | 3.3/**1.8** | **17.5**/**55.2** |

| CoSyNE | TFBind8 | UTR | ChEMBL | Molecule | Superconductor | D'Kitty | Rank ↓ | Opt. Gap ↑ |
|---|---|---|---|---|---|---|---|---|
| Baseline | $61.7^{(10.0)}$/$5.6^{(5.0)}$ | $57.3^{(9.6)}$/$\mathbf{12.7^{(9.8)}}$ | $63.6^{(0.4)}$/$28.2^{(11.3)}$ | $94.8^{(10.1)}$/$\mathbf{12.2^{(7.3)}}$ | $\mathbf{37.0^{(4.1)}}$/$0.0^{(0.0)}$ | $62.7^{(1.3)}$/$0.0^{(0.0)}$ | 5.3/4.5 | -0.6/-52.1 |
| COMs⁻ | $70.1^{(12.8)}$/$\mathbf{16.5^{(8.5)}}$ | $\mathbf{73.2^{(8.9)}}$/$8.7^{(10.2)}$ | $63.5^{(0.4)}$/$\mathbf{19.3^{(14.1)}}$ | $88.4^{(11.7)}$/$\mathbf{17.4^{(4.1)}}$ | $28.6^{(5.5)}$/$0.0^{(0.0)}$ | $63.8^{(1.6)}$/$0.0^{(0.0)}$ | 5.7/5.7 | 1.1/-51.6 |
| COMs⁺ | $66.9^{(5.1)}$/$\mathbf{15.4^{(4.9)}}$ | $56.2^{(9.5)}$/$\mathbf{26.5^{(6.8)}}$ | $63.3^{(0.0)}$/$18.5^{(7.0)}$ | $\mathbf{117^{(3.6)}}$/$\mathbf{17.8^{(8.1)}}$ | $28.0^{(7.8)}$/$34.3^{(13.5)}$ | $70.6^{(1.1)}$/$16.6^{(9.0)}$ | 5.7/3.3 | 3.5/-40.4 |
| RoMA⁻ | $61.7^{(10.0)}$/$5.9^{(5.2)}$ | $57.3^{(9.6)}$/$\mathbf{12.7^{(9.9)}}$ | $63.6^{(0.4)}$/$27.7^{(10.9)}$ | $94.8^{(10.1)}$/$\mathbf{12.2^{(7.3)}}$ | $\mathbf{37.0^{(4.1)}}$/$0.0^{(0.0)}$ | $62.7^{(1.3)}$/$0.0^{(0.0)}$ | 5.3/4.5 | -0.6/-52.2 |
| RoMA⁺ | $70.4^{(7.2)}$/$\mathbf{17.7^{(5.5)}}$ | $\mathbf{77.8^{(4.4)}}$/$11.9^{(4.0)}$ | $64.4^{(2.5)}$/$19.8^{(3.9)}$ | $\mathbf{117^{(6.5)}}$/$10.2^{(4.4)}$ | $38.0^{(8.1)}$/$0.0^{(0.0)}$ | $50.7^{(1.5)}$/$0.0^{(0.0)}$ | 2.8/5.0 | 6.2/-52.0 |
| ROMO | $79.7^{(12.7)}$/$15.7^{(10.1)}$ | $62.0^{(9.8)}$/$5.8^{(4.6)}$ | $64.1^{(0.6)}$/$27.6^{(12.4)}$ | $90.8^{(3.9)}$/$\mathbf{17.3^{(9.8)}}$ | $30.6^{(4.5)}$/$0.0^{(0.0)}$ | $72.1^{(1.1)}$/$0.1^{(0.2)}$ | 4.2/5.2 | 3.1/-50.8 |
| GAMBO | $79.8^{(10.6)}$/$5.2^{(5.7)}$ | $68.0^{(12.5)}$/$9.1^{(9.0)}$ | $64.2^{(0.9)}$/$28.4^{(15.7)}$ | $\mathbf{99.4^{(15.0)}}$/$7.1^{(8.0)}$ | $\mathbf{37.0^{(4.1)}}$/$0.0^{(0.0)}$ | $62.7^{(1.3)}$/$0.0^{(0.0)}$ | 3.7/6.3 | 5.0/-53.6 |
| DynAMO | $91.3^{(4.4)}$/$\mathbf{18.1^{(13.0)}}$ | $77.2^{(11.6)}$/$\mathbf{20.3^{(2.3)}}$ | $63.9^{(0.9)}$/$\mathbf{35.0^{(17.9)}}$ | $114^{(7.0)}$/$\mathbf{22.8^{(11.9)}}$ | $40.6^{(8.6)}$/$\mathbf{74.4^{(46.3)}}$ | $67.5^{(1.4)}$/$\mathbf{77.0^{(35.9)}}$ | **2.3**/**1.2** | **12.3**/-20.7 |

| BO-qEI | TFBind8 | UTR | ChEMBL | Molecule | Superconductor | D'Kitty | Rank ↓ | Opt. Gap ↑ |
|---|---|---|---|---|---|---|---|---|
| Baseline | $87.3^{(5.8)}$/$73.7^{(0.6)}$ | $\mathbf{86.2^{(0.0)}}$/$73.8^{(0.5)}$ | $65.4^{(1.0)}$/$99.3^{(0.1)}$ | $116^{(3.1)}$/$\mathbf{93.0^{(0.5)}}$ | $\mathbf{53.1^{(3.3)}}$/$190^{(0.8)}$ | $84.4^{(0.9)}$/$124^{(7.4)}$ | 5.8/5.2 | 18.7/47.1 |
| COMs⁻ | $\mathbf{93.2^{(2.7)}}$/$73.7^{(0.7)}$ | $\mathbf{86.2^{(0.0)}}$/$74.3^{(0.5)}$ | $\mathbf{66.4^{(0.4)}}$/$99.3^{(0.1)}$ | $\mathbf{121^{(0.0)}}$/$\mathbf{93.2^{(0.4)}}$ | $43.2^{(5.1)}$/$192^{(10.2)}$ | $\mathbf{86.2^{(0.9)}}$/$147^{(8.9)}$ | 4.3/4.0 | 19.2/51.4 |
| COMs⁺ | $84.5^{(5.5)}$/$68.8^{(0.8)}$ | $85.6^{(0.8)}$/$71.8^{(0.3)}$ | $65.1^{(0.8)}$/$96.6^{(0.9)}$ | $\mathbf{121^{(0.0)}}$/$90.8^{(0.8)}$ | $47.3^{(4.1)}$/$206^{(1.2)}$ | $84.9^{(1.1)}$/$79.0^{(2.8)}$ | 6.0/5.7 | 17.9/40.3 |
| RoMA⁻ | $\mathbf{95.2^{(2.2)}}$/$74.1^{(0.4)}$ | $\mathbf{86.3^{(0.1)}}$/$74.1^{(0.3)}$ | $65.4^{(1.1)}$/$99.3^{(0.1)}$ | $\mathbf{121^{(0.0)}}$/$\mathbf{93.5^{(0.6)}}$ | $\mathbf{53.1^{(3.3)}}$/$190^{(0.8)}$ | $85.8^{(0.7)}$/$131^{(15.9)}$ | 2.7/3.3 | **21.0**/48.4 |
| RoMA⁺ | $82.9^{(5.2)}$/$67.5^{(2.0)}$ | $84.1^{(1.0)}$/$64.2^{(1.0)}$ | $\mathbf{66.6^{(0.9)}}$/$98.6^{(0.2)}$ | $\mathbf{121^{(0.1)}}$/$78.0^{(3.8)}$ | $50.9^{(2.1)}$/$196^{(0.5)}$ | $84.8^{(1.3)}$/$115^{(15.3)}$ | 5.2/6.3 | 18.3/32.9 |
| ROMO | $\mathbf{93.8^{(1.6)}}$/$73.8^{(0.6)}$ | $\mathbf{86.3^{(0.1)}}$/$68.7^{(1.6)}$ | $63.9^{(0.8)}$/$94.8^{(1.6)}$ | $118^{(5.5)}$/$92.5^{(1.0)}$ | $48.5^{(3.7)}$/$196^{(2.7)}$ | $85.5^{(1.7)}$/$55.2^{(36.3)}$ | 5.0/6.0 | 19.2/34.9 |
| GAMBO | $\mathbf{94.1^{(1.9)}}$/$74.0^{(0.1)}$ | $\mathbf{86.3^{(0.2)}}$/$74.3^{(0.4)}$ | $\mathbf{66.8^{(0.7)}}$/$99.3^{(0.1)}$ | $\mathbf{121^{(0.0)}}$/$\mathbf{93.3^{(0.4)}}$ | $50.8^{(3.3)}$/$193^{(1.2)}$ | $86.7^{(1.1)}$/$17.7^{(3.5)}$ | **2.2**/4.0 | 20.8/30.0 |
| DynAMO | $91.9^{(4.4)}$/$\mathbf{74.8^{(0.2)}}$ | $\mathbf{86.2^{(0.0)}}$/$\mathbf{74.6^{(0.3)}}$ | $\mathbf{67.0^{(1.3)}}$/$\mathbf{99.4^{(0.1)}}$ | $\mathbf{121^{(0.0)}}$/$\mathbf{93.5^{(0.4)}}$ | $\mathbf{53.5^{(5.0)}}$/$198^{(1.9)}$ | $85.5^{(1.1)}$/$\mathbf{277^{(59.7)}}$ | 3.0/**1.3** | 20.7/**74.2** |

| BO-qUCB | TFBind8 | UTR | ChEMBL | Molecule | Superconductor | D'Kitty | Rank ↓ | Opt. Gap ↑ |
|---|---|---|---|---|---|---|---|---|
| Baseline | $88.1^{(5.3)}$/$\mathbf{73.9^{(0.5)}}$ | $\mathbf{86.2^{(0.1)}}$/$\mathbf{74.3^{(0.4)}}$ | $\mathbf{66.4^{(0.7)}}$/$\mathbf{99.4^{(0.1)}}$ | $\mathbf{121^{(1.3)}}$/$\mathbf{93.6^{(0.5)}}$ | $\mathbf{51.3^{(3.6)}}$/$198^{(10.3)}$ | $84.5^{(0.8)}$/$94.1^{(3.9)}$ | 3.7/3.0 | 19.4/43.5 |
| COMs⁻ | $\mathbf{88.5^{(6.4)}}$/$73.4^{(0.6)}$ | $\mathbf{86.2^{(0.0)}}$/$74.2^{(0.7)}$ | $66.0^{(1.1)}$/$99.2^{(0.1)}$ | $\mathbf{121^{(0.0)}}$/$93.3^{(0.4)}$ | $47.7^{(3.5)}$/$198^{(1.6)}$ | $85.4^{(1.8)}$/$107^{(5.5)}$ | 4.5/4.5 | 19.0/45.5 |
| COMs⁺ | $\mathbf{89.1^{(7.1)}}$/$69.0^{(0.8)}$ | $85.9^{(0.4)}$/$72.3^{(0.5)}$ | $65.6^{(1.1)}$/$97.1^{(0.9)}$ | $\mathbf{122^{(0.4)}}$/$91.2^{(0.6)}$ | $45.7^{(3.7)}$/$261^{(50.0)}$ | $84.7^{(1.6)}$/$89.2^{(14.0)}$ | 5.2/5.7 | 18.6/51.3 |
| RoMA⁻ | $86.9^{(5.0)}$/$\mathbf{73.9^{(0.4)}}$ | $\mathbf{86.2^{(0.1)}}$/$\mathbf{74.4^{(0.4)}}$ | $\mathbf{66.4^{(0.7)}}$/$\mathbf{99.4^{(0.0)}}$ | $120^{(1.3)}$/$\mathbf{93.7^{(0.5)}}$ | $\mathbf{51.3^{(3.6)}}$/$198^{(10.3)}$ | $84.5^{(0.8)}$/$94.1^{(3.9)}$ | 3.8/3.0 | 19.2/43.6 |
| RoMA⁺ | $84.6^{(5.9)}$/$68.2^{(2.2)}$ | $84.3^{(1.1)}$/$63.3^{(2.5)}$ | $\mathbf{66.9^{(1.0)}}$/$98.3^{(0.3)}$ | $\mathbf{121^{(0.2)}}$/$78.3^{(4.5)}$ | $52.1^{(3.2)}$/$194^{(0.3)}$ | $82.9^{(1.2)}$/$109^{(8.3)}$ | 4.7/6.5 | 18.5/39.9 |
| ROMO | $\mathbf{95.2^{(2.5)}}$/$\mathbf{74.0^{(0.5)}}$ | $\mathbf{86.2^{(0.0)}}$/$67.2^{(2.0)}$ | $64.7^{(1.0)}$/$94.9^{(1.3)}$ | $118^{(2.1)}$/$92.4^{(1.0)}$ | $50.2^{(4.7)}$/$197^{(1.3)}$ | $85.5^{(1.1)}$/$45.2^{(5.6)}$ | 4.7/6.2 | 19.9/33.2 |
| GAMBO | $\mathbf{95.4^{(1.6)}}$/$\mathbf{74.0^{(0.5)}}$ | $\mathbf{86.2^{(0.0)}}$/$\mathbf{74.3^{(0.3)}}$ | $66.3^{(1.1)}$/$\mathbf{99.3^{(0.1)}}$ | $\mathbf{121^{(1.3)}}$/$\mathbf{93.4^{(0.4)}}$ | $50.2^{(2.8)}$/$190^{(9.3)}$ | $83.6^{(1.0)}$/$22.0^{(2.1)}$ | 4.7/5.0 | 20.2/30.3 |
| DynAMO | $95.1^{(1.9)}$/$\mathbf{74.3^{(0.5)}}$ | $\mathbf{86.2^{(0.0)}}$/$\mathbf{74.4^{(0.6)}}$ | $\mathbf{66.7^{(1.5)}}$/$\mathbf{99.3^{(0.1)}}$ | $\mathbf{121^{(0.0)}}$/$\mathbf{93.5^{(0.6)}}$ | $48.1^{(4.0)}$/$211^{(22.8)}$ | $86.9^{(4.5)}$/$175^{(44.7)}$ | **3.5**/**1.8** | 20.5/**59.4** |

*Table A2.* **Additional Model-Based Optimization Quality Results.** Each cell is the **Median@128** oracle score (i.e., the median oracle score achieved by 128 sampled design candidates), reported as mean$^{(95\% \text{ confidence interval})}$ across 10 random seeds, where higher is better. **Bolded** entries indicate overlapping 95% confidence intervals with the best performing algorithm (according to the mean) per optimizer. **Bolded** (resp., Underlined) Rank and Optimality Gap metrics indicate the best (resp., second best) for a given backbone optimizer.

| **Grad.** | **TFBind8** | **UTR** | **ChEMBL** | **Molecule** | **Superconductor** | **D'Kitty** | **Rank ↓** | **Opt. Gap ↑** |
|---|---|---|---|---|---|---|---|---|
| Dataset $\mathcal{D}$ | 33.7 | 42.8 | 50.9 | 87.6 | 6.7 | 77.8 | —/— | —/— |
| Baseline | $58.1^{(6.1)}$ | $58.6^{(13.1)}$ | $59.3^{(8.6)}$ | $85.3^{(7.7)}$ | $36.0^{(6.7)}$ | $65.1^{(14.4)}$ | 4.3 | 10.5 |
| COMs⁻ | $53.0^{(8.5)}$ | $58.3^{(13.2)}$ | $59.1^{(8.6)}$ | $84.0^{(7.2)}$ | $22.5^{(3.2)}$ | $71.0^{(10.7)}$ | 5.8 | 8.1 |
| COMs⁺ | $43.9^{(0.0)}$ | $59.0^{(0.5)}$ | $63.3^{(0.0)}$ | $93.2^{(7.7)}$ | $21.3^{(5.6)}$ | $89.9^{(1.0)}$ | 4.0 | 11.8 |
| RoMA⁻ | $51.1^{(6.1)}$ | $58.6^{(13.1)}$ | $59.1^{(8.6)}$ | $85.3^{(7.7)}$ | $36.0^{(6.7)}$ | $65.1^{(14.4)}$ | 5.0 | 9.3 |
| RoMA⁺ | $48.2^{(4.3)}$ | $77.4^{(0.0)}$ | $63.3^{(0.0)}$ | $84.5^{(0.0)}$ | $38.2^{(0.8)}$ | $88.5^{(0.1)}$ | 3.2 | 16.8 |
| ROMO | $58.7^{(3.3)}$ | $37.7^{(0.3)}$ | $27.4^{(1.2)}$ | $61.8^{(2.6)}$ | $27.0^{(0.6)}$ | $46.0^{(11.7)}$ | 6.5 | -6.8 |
| GAMBO | $63.8^{(13.7)}$ | $75.3^{(9.9)}$ | $60.1^{(3.3)}$ | $91.6^{(11.2)}$ | $46.0^{(6.7)}$ | $90.1^{(14.4)}$ | 1.8 | 21.2 |
| DynAMO | $47.0^{(2.8)}$ | $69.8^{(6.0)}$ | $61.9^{(2.2)}$ | $85.9^{(0.4)}$ | $23.4^{(8.5)}$ | $68.7^{(12.1)}$ | 4.5 | 9.5 |

| **Adam** | **TFBind8** | **UTR** | **ChEMBL** | **Molecule** | **Superconductor** | **D'Kitty** | **Rank ↓** | **Opt. Gap ↑** |
|---|---|---|---|---|---|---|---|---|
| Baseline | $54.7^{(8.8)}$ | $60.4^{(12.7)}$ | $59.2^{(8.6)}$ | $87.9^{(10.0)}$ | $37.4^{(6.2)}$ | $56.8^{(19.8)}$ | 3.3 | 9.5 |
| COMs⁻ | $54.8^{(8.8)}$ | $59.5^{(12.7)}$ | $59.2^{(8.6)}$ | $90.8^{(10.4)}$ | $22.5^{(3.2)}$ | $57.1^{(19.6)}$ | 4.0 | 7.4 |
| COMs⁺ | $48.0^{(1.7)}$ | $59.1^{(0.5)}$ | $63.3^{(0.0)}$ | $89.3^{(10.4)}$ | $23.3^{(3.7)}$ | $56.4^{(13.3)}$ | 4.8 | 6.6 |
| RoMA⁻ | $54.7^{(8.8)}$ | $60.4^{(12.7)}$ | $59.2^{(8.6)}$ | $87.9^{(10.0)}$ | $37.4^{(6.2)}$ | $56.8^{(19.8)}$ | 3.3 | 9.5 |
| RoMA⁺ | $50.1^{(4.3)}$ | $77.4^{(0.0)}$ | $63.3^{(0.0)}$ | $84.7^{(0.0)}$ | $34.9^{(1.8)}$ | $63.7^{(6.2)}$ | 3.3 | 12.4 |
| ROMO | $54.0^{(0.0)}$ | $36.8^{(0.1)}$ | $63.3^{(0.0)}$ | $50.5^{(0.3)}$ | $26.1^{(0.5)}$ | $30.9^{(0.0)}$ | 5.7 | -6.3 |
| GAMBO | $49.5^{(8.9)}$ | $55.7^{(12.7)}$ | $57.7^{(9.1)}$ | $84.3^{(9.6)}$ | $37.4^{(6.2)}$ | $87.8^{(4.3)}$ | 5.0 | 12.1 |
| DynAMO | $47.7^{(3.0)}$ | $69.0^{(5.2)}$ | $62.4^{(1.9)}$ | $86.4^{(0.6)}$ | $23.0^{(6.0)}$ | $65.6^{(14.1)}$ | 4.7 | 9.1 |

| **CMA-ES** | **TFBind8** | **UTR** | **ChEMBL** | **Molecule** | **Superconductor** | **D'Kitty** | **Rank ↓** | **Opt. Gap ↑** |
|---|---|---|---|---|---|---|---|---|
| Baseline | $50.7^{(2.7)}$ | $71.7^{(10.4)}$ | $63.3^{(0.0)}$ | $83.9^{(1.0)}$ | $37.9^{(0.7)}$ | $59.3^{(10.9)}$ | 3.2 | 11.2 |
| COMs⁻ | $45.0^{(2.2)}$ | $68.0^{(8.6)}$ | $60.5^{(3.0)}$ | $89.9^{(10.8)}$ | $18.8^{(7.9)}$ | $59.8^{(9.9)}$ | 5.3 | 7.1 |
| COMs⁺ | $44.3^{(3.6)}$ | $62.0^{(8.8)}$ | $59.7^{(4.3)}$ | $91.6^{(9.3)}$ | $29.0^{(5.9)}$ | $61.2^{(15.0)}$ | 5.0 | 8.0 |
| RoMA⁻ | $50.7^{(2.7)}$ | $71.7^{(10.4)}$ | $63.3^{(0.0)}$ | $83.9^{(1.0)}$ | $37.9^{(0.7)}$ | $59.3^{(10.9)}$ | 3.2 | 11.2 |
| RoMA⁺ | $47.4^{(4.2)}$ | $58.0^{(7.0)}$ | $59.9^{(4.4)}$ | $91.6^{(10.0)}$ | $31.6^{(5.0)}$ | $60.4^{(7.7)}$ | 4.5 | 8.2 |
| ROMO | $48.9^{(3.1)}$ | $74.0^{(9.2)}$ | $60.0^{(3.4)}$ | $84.5^{(1.7)}$ | $22.8^{(1.6)}$ | $61.6^{(15.3)}$ | 3.5 | 8.7 |
| GAMBO | $44.2^{(0.8)}$ | $72.7^{(3.8)}$ | $62.7^{(1.1)}$ | $86.1^{(0.5)}$ | $21.4^{(2.0)}$ | $54.9^{(9.6)}$ | 5.5 | 7.1 |
| DynAMO | $45.3^{(2.4)}$ | $65.8^{(8.9)}$ | $59.3^{(3.8)}$ | $99.0^{(12.1)}$ | $22.5^{(5.1)}$ | $60.6^{(15.0)}$ | 4.8 | 8.8 |

| **CoSyNE** | **TFBind8** | **UTR** | **ChEMBL** | **Molecule** | **Superconductor** | **D'Kitty** | **Rank ↓** | **Opt. Gap ↑** |
|---|---|---|---|---|---|---|---|---|
| Baseline | $55.3^{(8.0)}$ | $53.6^{(10.2)}$ | $60.8^{(3.2)}$ | $87.4^{(16.6)}$ | $36.6^{(4.4)}$ | $59.3^{(14.5)}$ | 4.3 | 8.9 |
| COMs⁻ | $51.7^{(10.6)}$ | $70.9^{(9.2)}$ | $62.8^{(0.7)}$ | $83.1^{(8.3)}$ | $28.3^{(5.4)}$ | $58.9^{(17.3)}$ | 5.3 | 9.3 |
| COMs⁺ | $53.9^{(2.4)}$ | $41.1^{(1.1)}$ | $63.3^{(0.0)}$ | $107^{(3.8)}$ | $23.4^{(7.9)}$ | $61.2^{(15.6)}$ | 4.2 | 8.4 |
| RoMA⁻ | $55.3^{(8.0)}$ | $53.6^{(10.2)}$ | $60.8^{(3.2)}$ | $87.4^{(16.6)}$ | $36.6^{(4.4)}$ | $59.3^{(14.5)}$ | 4.3 | 8.9 |
| RoMA⁺ | $60.2^{(7.1)}$ | $67.7^{(8.7)}$ | $60.2^{(4.5)}$ | $103^{(11.2)}$ | $37.8^{(8.1)}$ | $48.9^{(13.9)}$ | 3.5 | 13.1 |
| ROMO | $69.1^{(12.9)}$ | $58.5^{(11.3)}$ | $62.1^{(1.4)}$ | $88.4^{(5.2)}$ | $29.9^{(4.5)}$ | $70.7^{(10.7)}$ | 3.2 | 13.2 |
| GAMBO | $59.5^{(12.0)}$ | $63.5^{(11.2)}$ | $55.4^{(9.6)}$ | $84.2^{(17.2)}$ | $36.6^{(4.4)}$ | $59.3^{(14.5)}$ | 4.5 | 9.8 |
| DynAMO | $53.8^{(11.0)}$ | $63.4^{(11.5)}$ | $59.3^{(3.8)}$ | $99.0^{(12.1)}$ | $20.5^{(5.8)}$ | $60.6^{(15.0)}$ | 5.3 | 9.5 |

| **BO-qEI** | **TFBind8** | **UTR** | **ChEMBL** | **Molecule** | **Superconductor** | **D'Kitty** | **Rank ↓** | **Opt. Gap ↑** |
|---|---|---|---|---|---|---|---|---|
| Baseline | $48.5^{(1.5)}$ | $59.9^{(2.0)}$ | $63.3^{(0.0)}$ | $86.7^{(0.6)}$ | $28.7^{(1.8)}$ | $72.4^{(1.8)}$ | 4.3 | 10.0 |
| COMs⁻ | $50.9^{(1.9)}$ | $59.6^{(1.6)}$ | $63.3^{(0.0)}$ | $86.6^{(0.6)}$ | $19.4^{(1.1)}$ | $78.1^{(1.1)}$ | 4.7 | 9.7 |
| COMs⁺ | $43.6^{(0.0)}$ | $66.0^{(1.6)}$ | $63.3^{(0.0)}$ | $87.5^{(0.6)}$ | $20.6^{(0.9)}$ | $66.3^{(2.2)}$ | 4.5 | 8.0 |
| RoMA⁻ | $50.0^{(1.6)}$ | $60.0^{(2.1)}$ | $63.3^{(0.0)}$ | $86.4^{(0.5)}$ | $28.7^{(1.8)}$ | $78.5^{(1.2)}$ | 3.5 | 11.2 |
| RoMA⁺ | $52.5^{(0.0)}$ | $61.0^{(1.1)}$ | $63.3^{(0.0)}$ | $93.3^{(5.7)}$ | $26.4^{(1.1)}$ | $74.3^{(1.3)}$ | 2.7 | 11.9 |
| ROMO | $49.9^{(2.2)}$ | $59.0^{(1.4)}$ | $63.3^{(0.0)}$ | $86.8^{(0.6)}$ | $24.6^{(0.6)}$ | $73.8^{(2.0)}$ | 4.7 | 9.6 |
| GAMBO | $46.4^{(1.8)}$ | $63.4^{(3.3)}$ | $63.3^{(0.0)}$ | $86.3^{(0.5)}$ | $28.9^{(1.1)}$ | $79.1^{(0.7)}$ | 3.5 | 11.3 |
| DynAMO | $51.5^{(0.9)}$ | $65.6^{(3.1)}$ | $63.3^{(0.0)}$ | $86.7^{(0.6)}$ | $23.5^{(2.4)}$ | $77.0^{(0.7)}$ | 3.3 | 11.3 |

| **BO-qUCB** | **TFBind8** | **UTR** | **ChEMBL** | **Molecule** | **Superconductor** | **D'Kitty** | **Rank ↓** | **Opt. Gap ↑** |
|---|---|---|---|---|---|---|---|---|
| Baseline | $50.3^{(1.8)}$ | $62.1^{(3.4)}$ | $63.3^{(0.0)}$ | $86.6^{(0.6)}$ | $31.7^{(1.2)}$ | $74.4^{(0.6)}$ | 2.7 | 11.5 |
| COMs⁻ | $51.1^{(1.0)}$ | $61.0^{(2.9)}$ | $63.3^{(0.0)}$ | $86.3^{(0.7)}$ | $19.8^{(1.3)}$ | $74.2^{(1.3)}$ | 4.5 | 9.4 |
| COMs⁺ | $43.6^{(0.0)}$ | $65.6^{(1.4)}$ | $63.3^{(0.0)}$ | $87.5^{(0.9)}$ | $20.1^{(1.0)}$ | $54.2^{(11.9)}$ | 4.5 | 5.8 |
| RoMA⁻ | $50.0^{(1.7)}$ | $62.1^{(3.4)}$ | $63.3^{(0.0)}$ | $86.7^{(0.6)}$ | $31.7^{(1.2)}$ | $74.4^{(0.6)}$ | 2.7 | 11.4 |
| RoMA⁺ | $52.5^{(0.0)}$ | $60.8^{(0.9)}$ | $63.3^{(0.0)}$ | $91.4^{(5.6)}$ | $29.5^{(1.4)}$ | $65.8^{(9.3)}$ | 3.2 | 10.6 |
| ROMO | $49.8^{(2.0)}$ | $58.2^{(0.2)}$ | $63.3^{(0.0)}$ | $86.8^{(0.5)}$ | $24.3^{(0.7)}$ | $75.0^{(1.7)}$ | 3.8 | 9.6 |
| GAMBO | $47.9^{(1.9)}$ | $59.8^{(1.2)}$ | $63.3^{(0.0)}$ | $86.0^{(0.6)}$ | $33.1^{(2.9)}$ | $73.8^{(1.2)}$ | 4.8 | 10.7 |
| DynAMO | $48.8^{(1.8)}$ | $65.9^{(3.7)}$ | $63.3^{(0.0)}$ | $86.5^{(0.5)}$ | $22.7^{(2.0)}$ | $50.4^{(14.6)}$ | 4.7 | 6.3 |

where $\dim(x)$ is the number of design dimensions and $x_{ik}^F$ is the $k$th dimension of design $x_i^F$. Note that the L1C metric is only defined for designs sampled from a continuous search space; to compute the L1C metric for discrete optimization tasks, we use task-specific foundation models to embed discrete designs into a continuous latent space. For DNA design tasks (i.e., **TFBind8** and **UTR**), we use the DNABERT-2 foundation model with 117M parameters (`zhihan1996/DNABERT-2-117M`) from Zhou et al. (2024b) to embed candidate DNA sequences into a continuous latent space. Similarly for molecule design tasks (i.e., **ChEMBL** and **Molecule**), we use the ChemBERT model (`jonghyunlee/ChemBERT_ChEMBL_pretrained`) from Zhang et al. (2022) to embed candidate molecules into a continuous latent space.

We report MN and L1C metric scores in **Supplementary Table A3**. We find that compared with other MBO objective-modifying methods, DynAMO achieves the best Rank and Optimality Gap for 3 of the 6 optimizers evaluated (Grad., Adam, and BO-qEI). For the remaining 3 optimizers evaluated, DynAMO is within the top 2 evaluated methods in terms of both average Rank and Optimality Gap for the L1C ($L_1$ coverage) metric. Altogether, our results support that DynAMO is competitive according to the MN and L1C diversity metrics in addition to the Pairwise Diversity metric reported in **Table 1**.

**What is the *best* notion of diversity?** In our work, we focus on the Pairwise Diversity metric in our main results (**Table 1**)—however, this does not mean that this metric is the best for all applications. Rather, our focus on the Pairwise Diversity metric is determined by our problem motivation. Compared with the minimum novelty and $L_1$ coverage diversity metrics, the definition of pairwise diversity best captures the notion of diversity that we are interested in—that is, capturing many possible 'modes of goodness' in optimizing the oracle reward function. We note that these modes of goodness may not necessarily be significantly 'novel' according to our task-specific distance metric, and so we treated the Minimum Novelty metric as only a secondary diversity objective for evaluating DynAMO. (Indeed, because DynAMO encourages a generative policy to match a distribution of designs constructed from the offline dataset, DynAMO may *not* increase the minimum novelty of designs compared to those proposed by the comparable baseline optimizer.) Similarly, we find that the $L_1$ Coverage metric is more sensitive to outlier designs when compared to the Pairwise Diversity, and therefore also treat it as a secondary diversity evaluation metric for our experiments in **Supplementary Table A3**. Future work might explore other methods that focus on improving not only the Pairwise Diversity metric, but also other diversity metric(s), too.

### D.3. Imposing Alternative $f$-Divergence Diversity Objectives via Mixed-Divergence Regularization

The MBO problem formulation proposed in (7) introduces a weighted KL-divergence regularization of the original MBO optimization objective. However, alternative distribution matching objectives have been used in prior work (Agarwal et al., 2024; Gong et al., 2021; Ma et al., 2022), and one might hypothesize that we can similarly generalize (7) as

$$\begin{aligned} \max_{\pi \in \Pi} \quad & J_f(\pi) = \mathbb{E}_{q^\pi}[r_\theta(x)] - \frac{\beta}{\tau} D_f(q^\pi || p_{\mathcal{D}}^\tau) \\ \text{s.t.} \quad & \mathbb{E}_{x \sim p_{\mathcal{D}}^\tau(x)}[c^*(x)] - \mathbb{E}_{x \sim q^\pi(x)}[c^*(x)] \leq W_0 \end{aligned} \tag{55}$$

to any arbitrary $f$-divergence metric $D_f(\cdot||\cdot)$ that measures the difference between two probability distributions $Q, P$ over a space $\Omega$ defined by $D_f(Q||P) := \int_\Omega dP\, f\left(\frac{dQ}{dP}\right)$ for a convex univariate *generator* function $f$. For example in our main text, we specialize to the KL-divergence where $f_{\text{KL}}(u) := u \log u$ traditionally used in the imitation learning literature.

However, we found that such a naïve approach does ***not*** generalize well to alternative $f$-divergences: recall that a core contribution of our work was the ability to reformulate the optimization objective as a weighted sum over distribution entropy and divergence (i.e., **Lemma 3.3**) in order to admit an explicit, closed form solution for the dual function in **Lemma 3.4**. Such an approach is intractable using standard algebraic techniques. This is not ideal, as a number of prior works have proposed that alternative divergences—such as the $\chi^2$-divergence defined by the generator $f_{\chi^2}(u) = (u-1)^2/2$—can better penalize out-of-distribution surrogate behavior and better quantify model uncertainty when compared to the KL-divergence (Tsybakov, 2008; Nishiyama & Sason, 2020; Ma et al., 2022; Wang et al., 2024).

In this section, we show how to overcome this limitation and demonstrate how our theoretical and empirical results generalize to alternative $f$-divergence objectives for enforcing distribution matching in the sampling policy. Firstly, we look to recent work by Huang et al. (2024a) and others describing 'mixed $f$-regularization' defined by a mixed generator function $f_\gamma(u) := \gamma f(u) + u \log u$ for some weighting scalar $\gamma \in [1, +\infty)$, which admits a 'mixed $f$-divergence' given by

$$D_f(Q||P; \gamma) := \gamma D_f(Q||P) + D_{\text{KL}}(Q||P) \tag{56}$$

*Table A3.* **Additional Model-Based Optimization Diversity Results.** Each cell is a value `mn`/`llc`; where `mn` is the **Minimum Novelty** and `llc` the $L_1$ **Coverage**. Both metrics are reported mean$^{\text{(95\% confidence interval)}}$ across 10 random seeds, where higher is better. **Bolded** entries indicate overlapping 95% confidence intervals with the best performing algorithm (according to the mean) per optimizer. **Bolded** (resp., Underlined) Rank and Optimality Gap (Opt. Gap) metrics indicate the best (resp., second best) for a given backbone optimizer.

| Grad. | TFBind8 | UTR | ChEMBL | Molecule | Superconductor | D'Kitty | Rank ↓ | Opt. Gap ↑ |
|---|---|---|---|---|---|---|---|---|
| Dataset $\mathcal{D}$ | 0.0/0.42 | 0.0/0.31 | 0.0/1.42 | 0.0/0.68 | 0.0/6.26 | 0.0/0.58 | —/— | —/— |
| Baseline | **21.2**$^{(3.0)}$/0.16$^{(0.1)}$ | **51.7**$^{(2.9)}$/0.20$^{(0.13)}$ | **97.4**$^{(3.9)}$/0.21$^{(0.10)}$ | 79.5$^{(19.7)}$/0.42$^{(0.18)}$ | 95.0$^{(0.7)}$/0.00$^{(0.00)}$ | 102$^{(6.1)}$/0.00$^{(0.00)}$ | 3.3/6.3 | 74.5/-1.44 |
| COMs⁻ | **22.1**$^{(2.7)}$/0.14$^{(0.11)}$ | **51.4**$^{(3.2)}$/0.20$^{(0.12)}$ | **97.4**$^{(3.9)}$/0.24$^{(0.11)}$ | **89.0**$^{(7.7)}$/0.40$^{(0.10)}$ | 94.1$^{(0.7)}$/0.00$^{(0.00)}$ | 100$^{(4.8)}$/0.00$^{(0.00)}$ | 3.3/7.7 | 75.7/-1.45 |
| COMs⁺ | 10.9$^{(0.3)}$/**0.49**$^{(0.02)}$ | 31.7$^{(0.8)}$/0.31$^{(0.00)}$ | 52.4$^{(11.0)}$/**1.11**$^{(0.16)}$ | 13.7$^{(1.1)}$/0.61$^{(0.09)}$ | **99.6**$^{(0.3)}$/0.37$^{(0.11)}$ | 100$^{(0.0)}$/0.80$^{(0.76)}$ | 6.2/2.7 | 51.4/-1.00 |
| RoMA⁻ | **21.2**$^{(3.1)}$/0.16$^{(0.1)}$ | **51.7**$^{(2.9)}$/0.21$^{(0.12)}$ | **97.4**$^{(3.9)}$/0.26$^{(0.10)}$ | 79.5$^{(19.7)}$/0.40$^{(0.19)}$ | 95.0$^{(0.7)}$/0.00$^{(0.00)}$ | 102$^{(6.1)}$/0.00$^{(0.00)}$ | 2.8/5.8 | 74.5/-1.44 |
| RoMA⁺ | 18.1$^{(1.4)}$/0.27$^{(0.02)}$ | 40.1$^{(0.2)}$/0.44$^{(0.01)}$ | 18.7$^{(0.1)}$/0.41$^{(0.01)}$ | 95.3$^{(0.0)}$/0.41$^{(0.02)}$ | 7.1$^{(0.8)}$/1.28$^{(0.01)}$ | 0.2$^{(0.0)}$/0.45$^{(0.00)}$ | 5.8/3.5 | 29.9/-1.07 |
| ROMO | 16.1$^{(0.5)}$/0.33$^{(0.02)}$ | 32.9$^{(0.1)}$/0.30$^{(0.00)}$ | 5.0$^{(0.7)}$/**1.31**$^{(0.02)}$ | 23.1$^{(0.0)}$/0.61$^{(0.02)}$ | 78.5$^{(0.5)}$/0.34$^{(0.16)}$ | **153**$^{(0.4)}$/**6.13**$^{(2.80)}$ | 6.0/2.7 | 51.5/-0.11 |
| GAMBO | 14.0$^{(2.0)}$/0.17$^{(0.10)}$ | 46.7$^{(2.7)}$/0.24$^{(0.13)}$ | 96.8$^{(3.9)}$/0.25$^{(0.16)}$ | 76.8$^{(19.7)}$/0.37$^{(0.11)}$ | 83.8$^{(6.8)}$/0.00$^{(0.00)}$ | 31.5$^{(3.6)}$/0.09$^{(0.14)}$ | 6.0/5.7 | 58.3/-1.42 |
| DynAMO | 21.1$^{(1.1)}$/**0.36**$^{(0.04)}$ | **52.2**$^{(1.3)}$/**0.52**$^{(0.06)}$ | **98.6**$^{(1.5)}$/**1.46**$^{(0.38)}$ | 85.8$^{(1.0)}$/**2.49**$^{(0.06)}$ | 95.0$^{(0.4)}$/**6.47**$^{(1.24)}$ | 107$^{(6.7)}$/**5.85**$^{(1.35)}$ | **2.2/1.3** | **76.7/1.25** |

| Adam | TFBind8 | UTR | ChEMBL | Molecule | Superconductor | D'Kitty | Rank ↓ | Opt. Gap ↑ |
|---|---|---|---|---|---|---|---|---|
| Baseline | **23.7**$^{(2.8)}$/0.11$^{(0.06)}$ | **51.1**$^{(3.5)}$/0.22$^{(0.09)}$ | **95.5**$^{(5.3)}$/0.23$^{(0.15)}$ | 79.3$^{(21.2)}$/0.48$^{(0.31)}$ | 94.8$^{(0.7)}$/0.27$^{(0.55)}$ | **103**$^{(6.3)}$/0.24$^{(0.49)}$ | 3.5/5.8 | 74.5/-1.35 |
| COMs⁻ | **23.8**$^{(2.8)}$/0.13$^{(0.06)}$ | **51.6**$^{(3.0)}$/0.20$^{(0.11)}$ | **95.5**$^{(5.3)}$/0.24$^{(0.16)}$ | 78.6$^{(20.7)}$/0.49$^{(0.33)}$ | 94.1$^{(0.7)}$/0.00$^{(0.00)}$ | **102**$^{(6.0)}$/0.03$^{(0.00)}$ | 3.3/6.8 | 74.2/-1.43 |
| COMs⁺ | 13.0$^{(0.4)}$/**0.44**$^{(0.02)}$ | 31.7$^{(0.8)}$/0.31$^{(0.00)}$ | 53.0$^{(12.8)}$/**1.10**$^{(0.20)}$ | 15.0$^{(1.6)}$/0.50$^{(0.12)}$ | **99.7**$^{(0.2)}$/0.58$^{(0.28)}$ | 99.9$^{(0.1)}$/0.88$^{(0.95)}$ | 6.0/2.7 | 52.0/-0.98 |
| RoMA⁻ | **23.6**$^{(2.8)}$/0.12$^{(0.06)}$ | **51.1**$^{(3.5)}$/0.21$^{(0.09)}$ | **95.5**$^{(5.3)}$/0.21$^{(0.16)}$ | 79.3$^{(21.2)}$/0.48$^{(0.32)}$ | 94.8$^{(0.7)}$/0.27$^{(0.55)}$ | **103**$^{(6.3)}$/0.04$^{(0.03)}$ | 3.3/6.8 | 74.5/-1.39 |
| RoMA⁺ | 18.3$^{(0.5)}$/0.28$^{(0.00)}$ | 40.1$^{(0.2)}$/0.46$^{(0.01)}$ | 18.9$^{(0.2)}$/0.41$^{(0.02)}$ | 95.3$^{(0.0)}$/0.42$^{(0.01)}$ | 47.6$^{(2.4)}$/1.87$^{(0.06)}$ | 5.1$^{(0.2)}$/0.78$^{(0.01)}$ | 6.0/3.7 | 37.6/-0.91 |
| ROMO | 13.6$^{(0.2)}$/0.28$^{(0.01)}$ | 32.9$^{(0.1)}$/0.30$^{(0.00)}$ | 21.9$^{(0.0)}$/**1.05**$^{(0.02)}$ | 23.4$^{(0.0)}$/0.74$^{(0.00)}$ | 98.1$^{(0.1)}$/1.34$^{(0.03)}$ | 99.8$^{(0.1)}$/0.08$^{(0.00)}$ | 6.0/3.7 | 48.3/-0.98 |
| GAMBO | **23.7**$^{(3.1)}$/0.14$^{(0.06)}$ | **51.3**$^{(3.4)}$/0.22$^{(0.10)}$ | **95.0**$^{(5.1)}$/0.35$^{(0.24)}$ | 80.0$^{(20.6)}$/0.50$^{(0.34)}$ | 84.8$^{(6.4)}$/0.26$^{(0.53)}$ | 27.3$^{(3.4)}$/0.09$^{(0.12)}$ | 4.3/5.2 | 60.4/-1.35 |
| DynAMO | 14.7$^{(1.9)}$/0.33$^{(0.05)}$ | 46.2$^{(0.5)}$/**0.55**$^{(0.03)}$ | **98.7**$^{(1.2)}$/**1.44**$^{(0.39)}$ | 85.9$^{(1.8)}$/**2.40**$^{(0.16)}$ | 94.9$^{(0.4)}$/**7.06**$^{(0.73)}$ | 108$^{(7.2)}$/**6.91**$^{(0.71)}$ | **3.0/1.2** | **74.7/1.50** |

| CMA-ES | TFBind8 | UTR | ChEMBL | Molecule | Superconductor | D'Kitty | Rank ↓ | Opt. Gap ↑ |
|---|---|---|---|---|---|---|---|---|
| Baseline | 16.5$^{(2.1)}$/**0.33**$^{(0.05)}$ | 47.8$^{(1.0)}$/0.48$^{(0.04)}$ | 96.5$^{(0.7)}$/2.18$^{(0.04)}$ | 73.0$^{(18.0)}$/1.82$^{(0.12)}$ | **100**$^{(0.0)}$/3.26$^{(1.42)}$ | 100$^{(0.0)}$/**3.77**$^{(1.36)}$ | 3.8/4.2 | 72.3/0.36 |
| COMs⁻ | 13.6$^{(0.7)}$/**0.34**$^{(0.05)}$ | 46.9$^{(0.9)}$/0.52$^{(0.06)}$ | **97.3**$^{(2.6)}$/1.81$^{(0.48)}$ | **84.7**$^{(6.9)}$/2.10$^{(0.32)}$ | **100**$^{(0.0)}$/0.31$^{(0.18)}$ | 100$^{(0.0)}$/0.43$^{(0.18)}$ | 4.7/4.7 | 73.7/-0.69 |
| COMs⁺ | 11.1$^{(2.1)}$/0.32$^{(0.04)}$ | 44.0$^{(2.8)}$/0.43$^{(0.06)}$ | 98.5$^{(1.1)}$/1.20$^{(0.27)}$ | 77.4$^{(19.7)}$/1.85$^{(0.16)}$ | 86.1$^{(3.5)}$/0.06$^{(0.01)}$ | 39.0$^{(7.9)}$/0.02$^{(0.00)}$ | 6.3/7.0 | 59.3/-0.96 |
| RoMA⁻ | 16.5$^{(2.2)}$/0.32$^{(0.04)}$ | 47.9$^{(0.9)}$/0.49$^{(0.03)}$ | 96.6$^{(0.7)}$/**2.16**$^{(0.08)}$ | 73.0$^{(18.0)}$/1.82$^{(0.12)}$ | **100**$^{(0.0)}$/**4.15**$^{(1.93)}$ | 100$^{(0.0)}$/**3.77**$^{(1.36)}$ | 3.5/4.0 | 72.3/0.51 |
| RoMA⁺ | 13.4$^{(0.7)}$/**0.36**$^{(0.04)}$ | 47.7$^{(1.7)}$/0.42$^{(0.04)}$ | **99.8**$^{(0.2)}$/1.35$^{(0.39)}$ | 80.0$^{(6.2)}$/1.80$^{(0.45)}$ | **100**$^{(0.0)}$/0.30$^{(0.15)}$ | 100$^{(0.0)}$/**3.97**$^{(2.11)}$ | **3.2**/5.7 | 73.5/-0.24 |
| ROMO | 16.2$^{(2.3)}$/**0.38**$^{(0.02)}$ | 47.0$^{(1.7)}$/0.49$^{(0.06)}$ | 97.7$^{(1.7)}$/2.01$^{(0.24)}$ | 85.3$^{(5.9)}$/2.13$^{(0.15)}$ | **100**$^{(0.0)}$/3.14$^{(1.85)}$ | 51.9$^{(17.6)}$/0.50$^{(0.33)}$ | 3.5/4.0 | 66.4/-0.17 |
| GAMBO | 24.3$^{(0.9)}$/0.31$^{(0.04)}$ | **53.3**$^{(1.4)}$/0.51$^{(0.01)}$ | 95.0$^{(1.5)}$/**2.17**$^{(0.06)}$ | 72.5$^{(23.6)}$/1.83$^{(0.15)}$ | 85.6$^{(3.0)}$/3.37$^{(0.49)}$ | 41.5$^{(2.0)}$/**3.12**$^{(0.40)}$ | 5.5/4.3 | 62.0/0.27 |
| DynAMO | 12.9$^{(0.8)}$/**0.40**$^{(0.03)}$ | 48.0$^{(1.6)}$/**0.56**$^{(0.01)}$ | 96.7$^{(3.5)}$/**1.82**$^{(0.72)}$ | 81.8$^{(13.4)}$/**2.54**$^{(0.05)}$ | 94.5$^{(0.7)}$/**4.75**$^{(2.16)}$ | 112$^{(7.8)}$/3.29$^{(1.56)}$ | 4.0/2.2 | **74.3/0.62** |

| CoSyNE | TFBind8 | UTR | ChEMBL | Molecule | Superconductor | D'Kitty | Rank ↓ | Opt. Gap ↑ |
|---|---|---|---|---|---|---|---|---|
| Baseline | 24.5$^{(3.5)}$/0.10$^{(0.07)}$ | 49.7$^{(3.1)}$/0.22$^{(0.10)}$ | 98.5$^{(1.6)}$/0.39$^{(0.20)}$ | 86.6$^{(12.7)}$/0.27$^{(0.13)}$ | 93.2$^{(1.0)}$/0.10$^{(0.00)}$ | 91.9$^{(2.0)}$/0.10$^{(0.00)}$ | 3.2/5.3 | 74.1/-1.41 |
| COMs⁻ | 15.4$^{(4.9)}$/**0.23**$^{(0.09)}$ | 44.9$^{(3.5)}$/0.20$^{(0.13)}$ | 90.0$^{(15.2)}$/0.38$^{(0.27)}$ | 86.2$^{(7.7)}$/0.39$^{(0.29)}$ | 93.8$^{(0.6)}$/0.07$^{(0.01)}$ | **101**$^{(6.3)}$/0.06$^{(0.00)}$ | 5.0/5.8 | 71.9/-1.39 |
| COMs⁺ | 12.8$^{(0.6)}$/**0.24**$^{(0.04)}$ | 42.7$^{(1.4)}$/**0.32**$^{(0.03)}$ | 34.1$^{(14.4)}$/**1.31**$^{(0.22)}$ | 50.0$^{(1.9)}$/**1.39**$^{(0.24)}$ | 87.8$^{(3.1)}$/**1.29**$^{(0.37)}$ | 30.8$^{(5.7)}$/0.25$^{(0.09)}$ | 7.5/1.5 | 43.0/**-0.81** |
| RoMA⁻ | 24.6$^{(3.5)}$/0.10$^{(0.07)}$ | 49.7$^{(3.0)}$/0.22$^{(0.11)}$ | **98.6**$^{(1.5)}$/0.40$^{(0.20)}$ | 86.6$^{(12.7)}$/0.27$^{(0.12)}$ | 93.2$^{(1.0)}$/0.10$^{(0.00)}$ | 91.9$^{(2.0)}$/0.10$^{(0.00)}$ | **2.5**/5.2 | 74.1/-1.41 |
| RoMA⁺ | 17.1$^{(3.3)}$/**0.22**$^{(0.05)}$ | 49.3$^{(3.0)}$/0.42$^{(0.09)}$ | 99.5$^{(0.9)}$/0.53$^{(0.13)}$ | 89.6$^{(3.4)}$/0.55$^{(0.22)}$ | 93.2$^{(1.6)}$/0.10$^{(0.00)}$ | 65.5$^{(25.2)}$/0.05$^{(0.03)}$ | 4.0/3.7 | 69.0/-1.30 |
| ROMO | 22.9$^{(6.3)}$/**0.18**$^{(0.09)}$ | 48.9$^{(1.5)}$/0.25$^{(0.10)}$ | 75.8$^{(6.1)}$/**1.20**$^{(0.21)}$ | 92.2$^{(3.3)}$/0.39$^{(0.12)}$ | 84.6$^{(4.4)}$/0.09$^{(0.00)}$ | 31.8$^{(5.6)}$/0.06$^{(0.10)}$ | 5.0/4.5 | 59.4/-1.25 |
| GAMBO | 22.8$^{(2.8)}$/0.12$^{(0.08)}$ | 50.8$^{(1.2)}$/0.22$^{(0.15)}$ | 90.8$^{(14.2)}$/0.53$^{(0.18)}$ | 91.9$^{(3.4)}$/0.14$^{(0.13)}$ | 86.0$^{(3.3)}$/0.10$^{(0.00)}$ | 29.6$^{(3.6)}$/0.02$^{(0.00)}$ | 4.5/6.3 | 62.0/-1.42 |
| DynAMO | 17.8$^{(5.5)}$/**0.21**$^{(0.11)}$ | 48.4$^{(2.3)}$/0.18$^{(0.09)}$ | 96.7$^{(3.5)}$/0.64$^{(0.42)}$ | 80.2$^{(12.9)}$/0.43$^{(0.34)}$ | 94.5$^{(0.7)}$/**1.85**$^{(0.22)}$ | 112$^{(7.8)}$/0.94$^{(0.17)}$ | 4.0/3.3 | **75.0**/-0.90 |

| BO-qEI | TFBind8 | UTR | ChEMBL | Molecule | Superconductor | D'Kitty | Rank ↓ | Opt. Gap ↑ |
|---|---|---|---|---|---|---|---|---|
| Baseline | 21.8$^{(0.5)}$/**0.41**$^{(0.02)}$ | 51.5$^{(0.3)}$/0.55$^{(0.01)}$ | 97.6$^{(0.3)}$/2.37$^{(0.03)}$ | 85.4$^{(1.5)}$/2.11$^{(0.15)}$ | 94.6$^{(0.1)}$/7.84$^{(0.01)}$ | 106$^{(2.9)}$/6.61$^{(0.33)}$ | 3.3/5.0 | 76.2/1.70 |
| COMs⁻ | 21.9$^{(0.5)}$/**0.41**$^{(0.03)}$ | 51.8$^{(0.2)}$/**0.56**$^{(0.01)}$ | 97.3$^{(0.5)}$/**2.44**$^{(0.05)}$ | 85.2$^{(0.5)}$/**2.55**$^{(0.02)}$ | 92.5$^{(1.2)}$/7.61$^{(0.24)}$ | 105$^{(1.6)}$/7.19$^{(0.22)}$ | 4.0/3.0 | 75.6/1.85 |
| COMs⁺ | 12.7$^{(0.4)}$/**0.44**$^{(0.01)}$ | 43.2$^{(0.2)}$/**0.57**$^{(0.01)}$ | 83.3$^{(2.1)}$/**2.50**$^{(0.05)}$ | 79.4$^{(1.3)}$/2.41$^{(0.07)}$ | 85.5$^{(0.4)}$/7.50$^{(0.01)}$ | 35.5$^{(1.2)}$/1.66$^{(0.01)}$ | 7.5/3.8 | 56.6/0.90 |
| RoMA⁻ | 21.6$^{(0.3)}$/0.40$^{(0.03)}$ | 51.7$^{(0.3)}$/**0.55**$^{(0.02)}$ | **97.7**$^{(0.5)}$/2.41$^{(0.04)}$ | 85.9$^{(1.1)}$/0.52$^{(0.00)}$ | 94.6$^{(0.1)}$/7.84$^{(0.01)}$ | 105$^{(3.5)}$/6.45$^{(0.66)}$ | **3.2**/5.2 | 76.1/1.42 |
| RoMA⁺ | 13.6$^{(0.3)}$/0.31$^{(0.02)}$ | 45.0$^{(0.2)}$/**0.56**$^{(0.01)}$ | 98.5$^{(0.4)}$/1.86$^{(0.05)}$ | 88.2$^{(0.6)}$/1.99$^{(0.06)}$ | 94.3$^{(0.1)}$/**7.87**$^{(0.01)}$ | 116$^{(2.5)}$/7.09$^{(0.49)}$ | 3.7/5.0 | 76.0/1.67 |
| ROMO | 15.6$^{(0.4)}$/0.39$^{(0.02)}$ | 47.5$^{(0.2)}$/0.55$^{(0.01)}$ | 87.7$^{(1.3)}$/**2.50**$^{(0.03)}$ | 87.9$^{(1.0)}$/**2.48**$^{(0.05)}$ | 86.6$^{(2.2)}$/7.51$^{(0.09)}$ | 30.8$^{(25.4)}$/2.14$^{(1.34)}$ | 5.5/5.3 | 59.3/0.98 |
| GAMBO | 15.4$^{(0.3)}$/0.40$^{(0.03)}$ | 51.8$^{(0.2)}$/0.55$^{(0.01)}$ | 97.8$^{(0.3)}$/2.38$^{(0.10)}$ | 84.9$^{(0.9)}$/2.53$^{(0.05)}$ | 85.1$^{(0.4)}$/7.45$^{(0.01)}$ | 14.3$^{(1.5)}$/1.29$^{(0.08)}$ | 5.7/6.3 | 58.2/0.82 |
| DynAMO | 21.0$^{(0.5)}$/0.42$^{(0.01)}$ | 51.9$^{(0.2)}$/**0.56**$^{(0.01)}$ | 97.4$^{(0.4)}$/**2.47**$^{(0.03)}$ | 85.2$^{(0.9)}$/**2.54**$^{(0.03)}$ | 94.8$^{(0.1)}$/**7.87**$^{(0.01)}$ | 126$^{(14.6)}$/**7.92**$^{(0.04)}$ | **3.0/2.2** | **79.4/2.02** |

| BO-qUCB | TFBind8 | UTR | ChEMBL | Molecule | Superconductor | D'Kitty | Rank ↓ | Opt. Gap ↑ |
|---|---|---|---|---|---|---|---|---|
| Baseline | 21.6$^{(0.3)}$/0.40$^{(0.02)}$ | 51.7$^{(0.2)}$/0.54$^{(0.01)}$ | 97.9$^{(0.4)}$/2.40$^{(0.05)}$ | 85.3$^{(1.1)}$/**2.52**$^{(0.07)}$ | 93.8$^{(0.6)}$/7.78$^{(0.04)}$ | 98.8$^{(1.1)}$/6.64$^{(0.09)}$ | 3.5/4.7 | 74.8/1.77 |
| COMs⁻ | 21.7$^{(0.3)}$/0.40$^{(0.02)}$ | 51.7$^{(0.2)}$/**0.56**$^{(0.01)}$ | 97.4$^{(0.4)}$/2.40$^{(0.06)}$ | 85.4$^{(1.4)}$/**2.49**$^{(0.04)}$ | 92.9$^{(1.1)}$/7.75$^{(0.09)}$ | 99.2$^{(1.6)}$/6.84$^{(0.11)}$ | 3.8/3.7 | 74.7/1.80 |
| COMs⁺ | 12.6$^{(0.3)}$/**0.44**$^{(0.02)}$ | 43.5$^{(0.3)}$/**0.57**$^{(0.01)}$ | 84.0$^{(2.3)}$/**2.52**$^{(0.05)}$ | 79.2$^{(1.4)}$/2.38$^{(0.08)}$ | 84.6$^{(1.1)}$/7.51$^{(0.02)}$ | 39.4$^{(6.0)}$/1.66$^{(0.01)}$ | 7.5/3.7 | 57.2/0.90 |
| RoMA⁻ | 21.6$^{(0.3)}$/0.39$^{(0.03)}$ | 51.6$^{(0.3)}$/0.55$^{(0.01)}$ | 97.8$^{(0.4)}$/2.37$^{(0.05)}$ | 85.1$^{(1.1)}$/**2.54**$^{(0.07)}$ | 93.8$^{(0.6)}$/7.78$^{(0.04)}$ | 98.8$^{(1.1)}$/6.64$^{(0.09)}$ | 3.5/4.8 | 74.9/1.77 |
| RoMA⁺ | 13.9$^{(0.3)}$/0.31$^{(0.02)}$ | 45.1$^{(0.5)}$/**0.56**$^{(0.01)}$ | **98.8**$^{(0.3)}$/1.85$^{(0.02)}$ | 88.5$^{(0.5)}$/2.01$^{(0.06)}$ | 94.1$^{(0.1)}$/**7.86**$^{(0.01)}$ | 112$^{(2.5)}$/7.23$^{(0.16)}$ | **3.3**/5.2 | 75.4/1.69 |
| ROMO | 16.0$^{(0.4)}$/0.40$^{(0.02)}$ | 47.7$^{(0.2)}$/0.55$^{(0.01)}$ | 88.1$^{(1.2)}$/**2.51**$^{(0.06)}$ | 90.9$^{(0.6)}$/**2.49**$^{(0.05)}$ | 85.4$^{(0.3)}$/7.48$^{(0.02)}$ | 20.9$^{(2.4)}$/1.60$^{(0.03)}$ | 5.7/5.3 | 58.2/0.89 |
| GAMBO | 21.9$^{(0.4)}$/0.40$^{(0.01)}$ | 51.7$^{(0.3)}$/**0.56**$^{(0.01)}$ | 97.5$^{(0.4)}$/2.39$^{(0.05)}$ | 85.2$^{(1.0)}$/**2.52**$^{(0.04)}$ | 81.9$^{(1.9)}$/7.37$^{(0.09)}$ | 25.9$^{(1.4)}$/1.34$^{(0.04)}$ | 4.7/6.0 | 60.7/0.82 |
| DynAMO | 21.4$^{(0.5)}$/0.40$^{(0.02)}$ | 51.7$^{(0.2)}$/0.55$^{(0.01)}$ | 97.1$^{(0.5)}$/**2.47**$^{(0.07)}$ | 85.3$^{(1.1)}$/**2.54**$^{(0.05)}$ | 94.7$^{(0.2)}$/**7.88**$^{(0.03)}$ | 109$^{(4.5)}$/**7.80**$^{(0.23)}$ | 3.7/2.3 | **76.6/2.00** |

for probability distributions $Q, P$.[1] Given a mixed $f$-divergence, we can define a modified MBO objective as in (5):

$$J_f(\pi; \gamma) := \mathbb{E}_{q^\pi}[r_\theta(x)] - \frac{\beta}{\tau} D_f(q^\pi || p_\mathcal{D}^\tau; \gamma) = \mathbb{E}_{q^\pi}[r_\theta(x)] - \frac{\beta}{\tau} D_{\mathrm{KL}}(q^\pi || p_\mathcal{D}^\tau) - \frac{\beta\gamma}{\tau} D_f(q^\pi || p_\mathcal{D}^\tau)$$
$$= J(\pi) - \frac{\beta\gamma}{\tau} D_f(q^\pi || p_\mathcal{D}^\tau) \tag{57}$$

where $r_\theta$ is again the forward surrogate model, $J(\pi)$ is as in (5), $p_\mathcal{D}^\tau(x)$ is the $\tau$-weighted probability distribution as in **Definition 3.2**, and $q^\pi(x)$ is the sampled distribution over designs admitted by the realized sampling policy $\pi$. Given this expression for the modified MBO objective, it is easy to rewrite $J_f(\pi; \gamma)$ similar to **Lemma 3.3** in the main text:

**Lemma D.1** (Generalized Entropy-Divergence Formulation for Mixed $f$-Divergence). *Define $J_f(\pi; \gamma)$ as in (57). An equivalent representation of $J_f(\pi; \gamma)$ is*

$$J_f(\pi) \simeq -\mathcal{H}(q^\pi(x)) - (1 + \beta) D_{KL}(q^\pi(x) || p_\mathcal{D}^\tau(x)) - \beta\gamma D_f(q^\pi(x) || p_\mathcal{D}^\tau(x)) \tag{58}$$

*where $\mathcal{H}(\cdot)$ as the Shannon entropy and $D_f(\cdot||\cdot)$ as the $f$-divergence.*

*Proof.* The proof is trivial using **Lemma 3.3**:

$$J_f(\pi) := \mathbb{E}_{q^\pi}[r_\theta(x)] - \frac{\beta}{\tau} D_f(q^\pi || p_\mathcal{D}^\tau; \pi) = J(\pi) - \frac{\beta\gamma}{\tau} D_f(q^\pi || p_\mathcal{D}^\tau)$$
$$\simeq \tau \cdot J(\pi) - \beta\gamma D_f(q^\pi || p_\mathcal{D}^\tau) \tag{59}$$
$$\simeq -\mathcal{H}(q^\pi) - (1 + \beta) D_{\mathrm{KL}}(q^\pi || p_\mathcal{D}^\tau) - \beta\gamma D_f(q^\pi || p_\mathcal{D}^\tau)$$

up to a constant independent of the policy $\pi$. $\square$

We now consider its derivative optimization problem constrained by source critic feedback analogous to (7):

$$\max_{\pi \in \Pi} \quad J_f(\pi; \gamma) = \mathbb{E}_{q^\pi}[r_\theta(x)] - \frac{\beta}{\tau} D_f(q^\pi || p_\mathcal{D}^\tau; \gamma)$$
$$\text{s.t.} \quad \mathbb{E}_{x \sim p_\mathcal{D}^\tau(x)} c^*(x) - \mathbb{E}_{x \sim q^\pi(x)} c^*(x) \le W_0 \tag{60}$$

where $c^*(x)$ is again an adversarial source critic and $W_0$ is some nonnegative constant. We can show that (60) admits an explicit dual function which can be used to tractably solve this optimization problem.

**Lemma D.2** (Explicit Dual Function of (60)). *Consider the primal problem*

$$\max_{\pi \in \Pi} \quad J_f(\pi; \gamma) = \mathbb{E}_{q^\pi}[r_\theta(x)] - \frac{\beta}{\tau} D_f(q^\pi || p_\mathcal{D}^\tau; \gamma)$$
$$\text{s.t.} \quad \mathbb{E}_{x \sim p_\mathcal{D}^\tau(x)}[c^*(x)] - \mathbb{E}_{x \sim q^\pi(x)}[c^*(x)] \le W_0 \tag{61}$$

*for some convex function $f$ where $0 \notin dom(f)$. The Lagrangian dual function $g(\lambda)$ is bounded from below by the function $g_\ell(\lambda)$ given by*

$$g_\ell(\lambda) := \beta \left[ (1 + \gamma)\lambda(\mathbb{E}_{p_\mathcal{D}^\tau}[c^*(x)] - W_0) - \mathbb{E}_{p_\mathcal{D}^\tau} e^{\lambda c^*(x) - 1} - \gamma \mathbb{E}_{p_\mathcal{D}^\tau} f^\star(\lambda c^*(x)) \right] \tag{62}$$

*where $f^\star(\cdot)$ is the Fenchel conjugate of $f$.*

*Proof.* Recall that the generator function of the mixed $f$-divergence penalty is given by $f_\gamma(u) = \gamma f(u) + u \log u$ for some weighting scalar $\gamma \in [1, +\infty)$. Define $f_{\mathrm{KL}}(u) := u \log u$. From (10), the dual function $g(\lambda) : \mathbb{R}_+ \to \mathbb{R}$ of the primal

---

[1]It is trivial to verify both that $f_\gamma(u)$ is convex and that $0 \notin dom(f_\gamma)$ given a function $f(u)$ that also satisfies both of these conditions.

problem is given by

$$
\begin{aligned}
g(\lambda) &:= \min_{\pi \in \Pi} \left[ (1+\beta)\mathbb{E}_{p_{\mathcal{D}}^{\tau}} \, f_{\mathrm{KL}} \left( \frac{q^{\pi}}{p_{\mathcal{D}}^{\tau}} \right) + \beta\gamma\mathbb{E}_{p_{\mathcal{D}}^{\tau}} f \left( \frac{q^{\pi}}{p_{\mathcal{D}}^{\tau}} \right) - \mathbb{E}_{q^{\pi}} \log\left(q^{\pi}\right) + \beta(1+\gamma)\lambda \left( \mathbb{E}_{p_{\mathcal{D}}^{\tau}} c^*(x) - \mathbb{E}_{q^{\pi}} c^*(x) - W_0 \right) \right] \\
&= \min_{\pi \in \Pi} \left[ (1+\beta)\mathbb{E}_{p_{\mathcal{D}}^{\tau}} \, f_{\mathrm{KL}} \left( \frac{q^{\pi}}{p_{\mathcal{D}}^{\tau}} \right) + \beta\gamma\mathbb{E}_{p_{\mathcal{D}}^{\tau}} f \left( \frac{q^{\pi}}{p_{\mathcal{D}}^{\tau}} \right) - \left( \mathbb{E}_{p_{\mathcal{D}}^{\tau}} f_{\mathrm{KL}} \left( \frac{q^{\pi}}{p_{\mathcal{D}}^{\tau}} \right) + \mathbb{E}_{q^{\pi}} \log p_{\mathcal{D}}^{\tau} \right) \right. \\
&\qquad \left. + \beta(1+\gamma)\lambda \left( \mathbb{E}_{p_{\mathcal{D}}^{\tau}} c^*(x) - \mathbb{E}_{q^{\pi}} c^*(x) - W_0 \right) \right] \\
&= \min_{\pi \in \Pi} \left[ \beta\mathbb{E}_{p_{\mathcal{D}}^{\tau}} \, f_{\mathrm{KL}} \left( \frac{q^{\pi}}{p_{\mathcal{D}}^{\tau}} \right) + \beta\gamma\mathbb{E}_{p_{\mathcal{D}}^{\tau}} f \left( \frac{q^{\pi}}{p_{\mathcal{D}}^{\tau}} \right) - \mathbb{E}_{q^{\pi}} \log p_{\mathcal{D}}^{\tau} + \beta(1+\gamma)\lambda \left( \mathbb{E}_{p_{\mathcal{D}}^{\tau}} c^*(x) - \mathbb{E}_{q^{\pi}} c^*(x) - W_0 \right) \right]
\end{aligned}
\tag{63}
$$

where we define $\beta(1+\gamma)\lambda \in \mathbb{R}_+$ as the Lagrangian multiplier associated with the constraint in (60) (recall that $\mathbb{R}_+$ is closed under multiplication). We rearrange terms to rewrite $g(\lambda)$ as

$$
\begin{aligned}
g(\lambda) = \min_{\pi \in \Pi} \left[ \beta\mathbb{E}_{p_{\mathcal{D}}^{\tau}} \left[ -\left( \lambda c^*(x) \cdot \frac{q^{\pi}}{p_{\mathcal{D}}^{\tau}} \right) + f_{\mathrm{KL}} \left( \frac{q^{\pi}}{p_{\mathcal{D}}} \right) \right] + \beta\gamma\mathbb{E}_{p_{\mathcal{D}}^{\tau}} \left[ -\left( \lambda c^*(x) \cdot \frac{q^{\pi}}{p_{\mathcal{D}}^{\tau}} \right) + f \left( \frac{q^{\pi}}{p_{\mathcal{D}}^{\tau}} \right) \right] \right. \\
\left. - \mathbb{E}_{q^{\pi}} \log p_{\mathcal{D}}^{\tau} + \beta(1+\gamma)\lambda\mathbb{E}_{p_{\mathcal{D}}^{\tau}} c^*(x) - \beta(1+\gamma)\lambda W_0 \right]
\end{aligned}
\tag{64}
$$

The sum of function minima is a lower bound on the minima of the sum:

$$
\begin{aligned}
g(\lambda) &\geq \beta\mathbb{E}_{p_{\mathcal{D}}^{\tau}} \min_{\pi \in \Pi} \left[ -\left( \lambda c^*(x) \cdot \frac{q^{\pi}}{p_{\mathcal{D}}^{\tau}} \right) + f_{\mathrm{KL}} \left( \frac{q^{\pi}}{p_{\mathcal{D}}} \right) \right] + \beta\gamma\mathbb{E}_{p_{\mathcal{D}}^{\tau}} \min_{\pi \in \Pi} \left[ -\left( \lambda c^*(x) \cdot \frac{q^{\pi}}{p_{\mathcal{D}}^{\tau}} \right) + f \left( \frac{q^{\pi}}{p_{\mathcal{D}}} \right) \right] \\
&\quad - \max_{\pi \in \Pi} \mathbb{E}_{q^{\pi}} \log p_{\mathcal{D}}^{\tau} + \min_{\pi \in \Pi} \left[ \beta(1+\gamma)\lambda\mathbb{E}_{p_{\mathcal{D}}^{\tau}} c^*(x) - \beta(1+\gamma)\lambda W_0 \right] \\
&\sim \beta\mathbb{E}_{p_{\mathcal{D}}^{\tau}} \min_{\pi \in \Pi} \left[ -\left( \lambda c^*(x) \cdot \frac{q^{\pi}}{p_{\mathcal{D}}^{\tau}} \right) + f_{\mathrm{KL}} \left( \frac{q^{\pi}}{p_{\mathcal{D}}} \right) \right] + \beta\gamma\mathbb{E}_{p_{\mathcal{D}}^{\tau}} \min_{\pi \in \Pi} \left[ -\left( \lambda c^*(x) \cdot \frac{q^{\pi}}{p_{\mathcal{D}}^{\tau}} \right) + f \left( \frac{q^{\pi}}{p_{\mathcal{D}}} \right) \right] \\
&\quad + \beta(1+\gamma)\lambda\mathbb{E}_{p_{\mathcal{D}}^{\tau}} c^*(x) - \beta(1+\gamma)\lambda W_0
\end{aligned}
\tag{65}
$$

ignoring the term $\max_{\pi \in \Pi} \left[ \mathbb{E}_{q^{\pi}} \log p_{\mathcal{D}}^{\tau} \right]$ that is constant with respect to $\lambda$. We then perform the same tactic of minimizing over the superset $\mathbb{R}_+ \supseteq \{ z \mid \exists \pi \in \Pi \text{ s.t. } q^{\pi}(x)/p_{\mathcal{D}}^{\tau}(x) = z \}$ as in **Appendix A**:

$$
\begin{aligned}
g(\lambda) &\geq \beta\mathbb{E}_{p_{\mathcal{D}}^{\tau}} \min_{z \in \mathbb{R}_+} \left[ -\left( \lambda c^*(x) \cdot z \right) + f_{\mathrm{KL}}(z) \right] + \beta\gamma\mathbb{E}_{p_{\mathcal{D}}^{\tau}} \min_{z \in \mathbb{R}_+} \left[ -\left( \lambda c^*(x) \cdot z \right) + f(z) \right] + \beta(1+\gamma)\lambda(\mathbb{E}_{p_{\mathcal{D}}^{\tau}} c^*(x) - W_0) \\
&= \beta \left[ -\mathbb{E}_{p_{\mathcal{D}}^{\tau}} f_{\mathrm{KL}}^{\star}(\lambda c^*(x)) - \gamma\mathbb{E}_{p_{\mathcal{D}}^{\tau}} f^{\star}(\lambda c^*(x)) + (1+\gamma)\lambda(\mathbb{E}_{p_{\mathcal{D}}^{\tau}} c^*(x) - W_0) \right]
\end{aligned}
\tag{66}
$$

where $f^{\star}(\cdot)$ is the Fenchel conjugate of a convex function $f(\cdot)$. The Fenchel conjugate of $f_{\mathrm{KL}}(u) = u \log u$ is $f_{\mathrm{KL}}^{\star}(v) = e^{v-1}$ (Borwein & Lewis, 2006), so

$$
g(\lambda) \geq \beta \left[ -\mathbb{E}_{p_{\mathcal{D}}^{\tau}} e^{\lambda c^*(x)-1} - \gamma\mathbb{E}_{p_{\mathcal{D}}^{\tau}} f^{\star}(\lambda c^*(x)) + (1+\gamma)\lambda(\mathbb{E}_{p_{\mathcal{D}}^{\tau}} c^*(x) - W_0) \right]
\tag{67}
$$

Define the right hand side of this inequality as the function $g_\ell(\lambda)$ and the result is immediate. $\qquad\square$

**Corollary D.3** (Explicit Dual Function of (60) Using Mixed $\chi^2$-Divergence). *As an example, we can consider the mixed $\chi^2$-divergence defined by $D_{\chi^2}(Q||P; \gamma) = \gamma D_{\chi^2}(Q||P) + D_{KL}(Q||P)$ as used in Huang et al. (2024a). The $\chi^2$-divergence generator function is $f_{\chi^2}(u) = (u-1)^2/2$, and its Fenchel conjugate is $f_{\chi^2}^{\star}(v) = v + (v^2/2)$ from **Lemma C.4**. Directly applying **Lemma D.2**, our lower bound on the our dual function is*

$$
g(\lambda) \geq g_\ell(\lambda) := \beta \left[ -\mathbb{E}_{p_{\mathcal{D}}^{\tau}} e^{\lambda c^*(x)-1} - \gamma\mathbb{E}_{p_{\mathcal{D}}^{\tau}} \left( \frac{1}{2}(\lambda c^*(x))^2 + \lambda c^*(x) \right) + (1+\gamma)\lambda(\mathbb{E}_{p_{\mathcal{D}}^{\tau}} c^*(x) - W_0) \right]
\tag{68}
$$

To experimentally evaluate the utility of distribution matching using a mixed $\chi^2$-KL-Divergence, we substitute the $D_{\mathrm{KL}}(\cdot||\cdot)$ divergence with the mixed $\chi^2$-Divergence $D_{f_{\chi^2}}(\cdot||\cdot; \gamma)$ (setting $\gamma = 1.0$ for experimental evaluation) and its associated

dual function bound from (68) into **Algorithm 1**. Practically, we find that this only requires updating the dual function per **Corollary D.3** and the Lagrangian of (60) given by

$$\mathcal{L}(x; \lambda) = -\mathbb{E}_{q^\pi}[r_\theta(x)] + \frac{\beta}{\tau} \left[ \gamma D_f(q^\pi \| p_{\mathcal{D}}^\tau) + D_{\text{KL}}(q^\pi \| p_{\mathcal{D}}^\tau) \right] + \beta(1 + \gamma)\lambda \left[ \mathbb{E}_{p_{\mathcal{D}}^\tau} c^*(x) - \mathbb{E}_{q^\pi} c^*(x) - W_0 \right] \tag{69}$$

in **Algorithm 1**.

**Experimental Results.** We compare DynAMO implemented with a mixed $\chi^2$ divergence penalty (with $\gamma = 1.0$) against our original DynAMO implementation (i.e., $\gamma = 0$) in **Supplementary Tables A4-A5**. Empirically, we find that using the mixed $\chi^2$-divergence penalty offers limited utility compared with KL-divergence alone: the latter is non-inferior to the former according to both the Rank and Optimality Gap metrics for all 6 optimizers assessed according to the Best@128 oracle score. Furthermore, DynAMO outperforms DynAMO with mixed $\chi^2$-divergence according to the Rank and Optimality Gap metrics for 5 out of the 6 optimizers assessed according to the Pairwise Diversity metric. Based on our qualitative analysis, we hypothesize that the over-conservatism often attributed to $\chi^2$-divergence-based penalties in related literature (Ma et al., 2022; Huang et al., 2024a; Wang et al., 2024) may adversely affect the generative policy's ability to sufficiently explore the design space when compared to using KL-divergence-base distribution matching alone. Further work is needed to tune the relative mixing parameter $\gamma$ and/or explore how other alternative $f$-divergence metrics may be used with DynAMO.

### D.4. Theoretical Guarantees for DynAMO

In this section, we seek to place an upper bound on the difference between *true* diversity-penalized objective

$$J^\star(\pi) := \mathbb{E}_{x \sim q^\pi(x)}[r(x)] - \frac{\beta}{\tau} D_{\text{KL}}(q^\pi(x) \| p^\tau(x)) \tag{70}$$

realized by the final generative policy $\hat{\pi} \in \Pi$ learned by DynAMO (denoted as $\pi^*$ in the main text), and the true diversity-penalized objective realized by the true optimal policy $\pi^\star := \arg\max_{\pi \in \Pi} J^\star(\pi)$. Note that this objective $J^\star(\pi)$ is *not* equivalent to the offline MBO objective $J(\pi)$ introduced in (5); importantly, the objective $J(\pi)$ is a function of the *true*, hidden oracle reward $r(x)$ as opposed to the forward surrogate model $r_\theta(x)$. Furthermore, the KL-divergence penalty is computed with respect to the *true* $\tau$-weighted probability distribution $p^\tau(x)$, as opposed to its empirical estimate computed from the offline dataset $\mathcal{D}$ as in **Definition 3.2**. In principle, (70) captures the true trade-off between diversity and quality of designs that we hope to achieve by the theoretically optimal zero-regret generative policy $\pi^\star$ that maximizes (70) over $\Pi$.

Our main result is in **Theorem D.9** below, although we first step through the relevant assumptions and intermediate results necessary to arrive at our bound on (70). Firstly, we assume the following:

**Assumption D.4** (Surrogate Model Error Bound). There exists a finite $\varepsilon_0^2 \in \mathbb{R}_+$ such that

$$\mathbb{E}_{x \sim p^\tau(x)}[r(x) - r_\theta(x)]^2 \le \varepsilon_0^2 / 4 \tag{71}$$

for any choice in $\tau \ge 0$, where $p^\tau(x)$ is the true $\tau$-weighted probability distribution over $\mathcal{X}$.

**Assumption D.5** (Policy Realizability). Both the true optimal sampling policy $\pi^\star$ according to (70) and optimal sampling policy $\hat{\pi}$ according to (7) are contained in the (finite) policy class $\Pi$.

**Assumption D.6** (Bounded Importance Weights). Define the *importance weight* $w(x)$ as the ratio between probability distributions $q^\pi(x)$ and $p(x)$. There exists a finite $M \in \mathbb{R}_+$ such that for all possible permutations of $\pi \in \{\hat{\pi}, \pi^\star\}$ and $p(x) \in \{p^\tau(x), p_{\mathcal{D}}^\tau(x)\}$, we have $w(x) := q^\pi(x)/p(x) \le M$ for all $x \in \mathcal{X}$.

*Remark.* This assumption is mild assuming that (1) $\mathcal{D}$ is large enough such that $p_{\mathcal{D}}^\tau(x) \approx p^\tau(x)$; and (2) the policy $\pi$ has been learned with sufficiently large $\beta$ according to **Algorithm 1** or a similar distribution matching objective, such that the distribution of designs learned by the generative policy $q^{\hat{\pi}}(x)$ well-approximates the expert distribution $p_{\mathcal{D}}^\tau(x)$. Because the optimal policy $\pi^\star$ should also well-approximate $p^\tau(x)$ (and therefore $p_{\mathcal{D}}^\tau(x)$ by assumption), the assumption that such a finite $M$ exists is reasonable.

Under these assumptions, we first place a bound on the error of the forward surrogate model over the distribution of generated designs from the optimal policies according to both the offline objective $J(\pi)$ and true objective $J^\star(\pi)$:

**Lemma D.7** (Bounded Prediction Error). *Assume there exists an $M \in \mathbb{R}_+$ finite satisfying **Assumption D.6**. Then with probability at least $1 - \delta$ we have (for any $\delta > 0$ and for both $\pi = \pi^\star$ and $\pi = \hat{\pi}$)*

$$\mathbb{E}_{x \sim q^\pi(x)} |r(x) - r_\theta(x)| \le \frac{\varepsilon_0}{2} + M\sqrt{\frac{2\log(2|\Pi|/\delta)}{n}} \tag{72}$$

*Table A4.* **Quality of Design Candidates Using Mixed $\chi^2$-Divergence DynAMO.** Using **Corollary D.3** and (60), we show that it is possible to extend DynAMO to leverage a *mixed $\chi^2$-divergence* that equally weights both $\chi^2$-divergence and KL-divergence to penalize the original MBO objective. We evaluate this specialized implementation of DynAMO against baseline DynAMO and vanilla optimization methods, and report the Best@128 (resp., Median@128) oracle score achieved by the 128 evaluated designs in the top (resp., bottom) table. Metrics are reported mean$^{(95\% \text{ confidence interval})}$ across 10 random seeds, where higher is better. **Bolded** entries indicate average scores with an overlapping 95% confidence interval to the best performing method. **Bolded** (resp., Underlined) Rank and Optimality Gap (Opt. Gap) metrics indicate the best (resp., second best) for a given backbone optimizer.

| Best@128 | TFBind8 | UTR | ChEMBL | Molecule | Superconductor | D'Kitty | Rank ↓ | Opt. Gap ↑ |
|---|---|---|---|---|---|---|---|---|
| Dataset $\mathcal{D}$ | 43.9 | 59.4 | 60.5 | 88.9 | 40.0 | 88.4 | — | — |
| Grad. | $90.0^{(4.3)}$ | $80.9^{(12.1)}$ | $60.2^{(8.9)}$ | $88.8^{(4.0)}$ | $\mathbf{36.0^{(6.8)}}$ | $65.6^{(14.5)}$ | 2.8 | 6.8 |
| DynAMO-Grad. | $90.3^{(4.7)}$ | $\mathbf{86.2^{(0.0)}}$ | $\mathbf{64.4^{(2.5)}}$ | $91.2^{(0.0)}$ | $\mathbf{44.2^{(7.8)}}$ | $\mathbf{89.8^{(3.2)}}$ | **1.2** | **14.2** |
| Mixed $\chi^2$ DynAMO-Grad. | $59.3^{(8.3)}$ | $\mathbf{86.2^{(0.0)}}$ | $\mathbf{64.4^{(2.6)}}$ | $\mathbf{120^{(1.4)}}$ | $\mathbf{42.0^{(5.6)}}$ | $83.6^{(1.4)}$ | 2.0 | 12.5 |
| Adam | $62.9^{(13.0)}$ | $69.7^{(10.5)}$ | $62.9^{(1.9)}$ | $92.3^{(8.9)}$ | $37.8^{(6.3)}$ | $58.4^{(18.5)}$ | 2.7 | 0.5 |
| DynAMO-Adam | $\mathbf{95.2^{(1.7)}}$ | $\mathbf{86.2^{(0.0)}}$ | $\mathbf{65.2^{(1.1)}}$ | $91.2^{(0.0)}$ | $\mathbf{45.5^{(5.7)}}$ | $\mathbf{84.9^{(12.0)}}$ | **1.3** | **14.5** |
| Mixed $\chi^2$ DynAMO-Adam | $59.3^{(8.3)}$ | $86.2^{(0.0)}$ | $\mathbf{64.4^{(2.6)}}$ | $\mathbf{120^{(1.4)}}$ | $\mathbf{42.0^{(5.6)}}$ | $83.6^{(1.4)}$ | 2.0 | 12.5 |
| BO-qEI | $87.3^{(5.8)}$ | $\mathbf{86.2^{(0.0)}}$ | $65.4^{(1.0)}$ | $116^{(3.1)}$ | $\mathbf{53.1^{(3.3)}}$ | $84.4^{(0.9)}$ | 2.8 | 18.7 |
| DynAMO-BO-qEI | $91.9^{(4.4)}$ | $\mathbf{86.2^{(0.0)}}$ | $67.0^{(1.3)}$ | $\mathbf{121^{(0.0)}}$ | $\mathbf{53.5^{(5.0)}}$ | $\mathbf{85.5^{(1.1)}}$ | **1.5** | **20.7** |
| Mixed $\chi^2$ DynAMO-BO-qEI | $\mathbf{92.2^{(4.1)}}$ | $\mathbf{86.3^{(0.1)}}$ | $\mathbf{66.8^{(1.2)}}$ | $\mathbf{123^{(3.0)}}$ | $\mathbf{51.7^{(4.4)}}$ | $\mathbf{85.1^{(1.5)}}$ | 1.7 | **20.7** |
| BO-qUCB | $88.1^{(5.3)}$ | $\mathbf{86.2^{(0.1)}}$ | $66.4^{(0.7)}$ | $\mathbf{121^{(1.3)}}$ | $\mathbf{51.3^{(3.6)}}$ | $84.5^{(0.8)}$ | 2.2 | 19.4 |
| DynAMO-BO-qUCB | $\mathbf{95.1^{(1.9)}}$ | $\mathbf{86.2^{(0.0)}}$ | $\mathbf{66.7^{(1.5)}}$ | $\mathbf{121^{(0.0)}}$ | $48.1^{(4.0)}$ | $\mathbf{86.9^{(4.5)}}$ | 1.7 | **20.5** |
| Mixed $\chi^2$ DynAMO-BO-qUCB | $85.7^{(5.4)}$ | $\mathbf{86.3^{(0.2)}}$ | $\mathbf{66.3^{(0.9)}}$ | $\mathbf{121^{(0.0)}}$ | $\mathbf{51.5^{(4.3)}}$ | $83.8^{(1.1)}$ | 2.0 | 19.0 |
| CMA-ES | $\mathbf{87.6^{(8.3)}}$ | $\mathbf{86.2^{(0.0)}}$ | $\mathbf{66.1^{(1.0)}}$ | $106^{(5.9)}$ | $49.0^{(1.0)}$ | $72.2^{(0.1)}$ | 2.0 | 14.4 |
| DynAMO-CMA-ES | $89.8^{(3.6)}$ | $85.7^{(5.8)}$ | $63.9^{(0.9)}$ | $\mathbf{117^{(6.7)}}$ | $\mathbf{50.6^{(4.8)}}$ | $78.5^{(5.5)}$ | **1.7** | **17.5** |
| Mixed $\chi^2$ DynAMO-CMA-ES | $84.2^{(10.7)}$ | $84.5^{(2.6)}$ | $\mathbf{65.1^{(1.3)}}$ | $\mathbf{113^{(4.8)}}$ | $45.0^{(4.9)}$ | $81.8^{(4.0)}$ | 2.3 | 15.4 |
| CoSyNE | $61.7^{(10.0)}$ | $57.3^{(9.6)}$ | $63.6^{(0.4)}$ | $94.8^{(10.1)}$ | $\mathbf{37.0^{(4.1)}}$ | $62.7^{(1.3)}$ | 2.8 | -0.6 |
| DynAMO-CoSyNE | $91.3^{(4.4)}$ | $\mathbf{77.2^{(11.6)}}$ | $63.9^{(0.9)}$ | $\mathbf{114^{(7.0)}}$ | $\mathbf{40.6^{(8.6)}}$ | $67.5^{(1.4)}$ | **1.5** | **12.3** |
| Mixed $\chi^2$ DynAMO-CoSyNE | $\mathbf{94.3^{(2.3)}}$ | $\mathbf{78.3^{(8.3)}}$ | $63.1^{(2.2)}$ | $\mathbf{100^{(10.9)}}$ | $\mathbf{37.4^{(9.7)}}$ | $82.3^{(4.1)}$ | 1.7 | 12.4 |

| Median@128 | TFBind8 | UTR | ChEMBL | Molecule | Superconductor | D'Kitty | Rank ↓ | Opt. Gap ↑ |
|---|---|---|---|---|---|---|---|---|
| Dataset $\mathcal{D}$ | 33.7 | 42.8 | 50.9 | 87.6 | 6.7 | 77.8 | — | — |
| Grad. | $\mathbf{58.1^{(6.1)}}$ | $58.6^{(13.1)}$ | $\mathbf{59.3^{(8.6)}}$ | $85.3^{(7.7)}$ | $\mathbf{36.0^{(6.7)}}$ | $65.1^{(14.4)}$ | 2.0 | **10.5** |
| DynAMO-Grad. | $47.0^{(2.8)}$ | $69.8^{(6.0)}$ | $61.9^{(2.2)}$ | $85.9^{(0.4)}$ | $23.4^{(8.5)}$ | $68.7^{(12.1)}$ | **1.5** | 9.5 |
| Mixed $\chi^2$ DynAMO-Grad. | $45.2^{(6.9)}$ | $66.9^{(5.2)}$ | $58.3^{(6.5)}$ | $\mathbf{86.6^{(2.1)}}$ | $20.6^{(1.4)}$ | $64.4^{(8.3)}$ | 2.5 | 7.1 |
| Adam | $54.7^{(8.8)}$ | $60.4^{(12.7)}$ | $59.2^{(8.6)}$ | $\mathbf{87.9^{(10.0)}}$ | $\mathbf{37.4^{(6.2)}}$ | $56.8^{(19.8)}$ | 1.8 | **9.5** |
| DynAMO-Adam | $47.7^{(3.0)}$ | $69.0^{(5.2)}$ | $62.4^{(1.9)}$ | $86.4^{(0.6)}$ | $23.0^{(6.0)}$ | $65.6^{(14.1)}$ | **1.7** | 9.1 |
| Mixed $\chi^2$ DynAMO-Adam | $45.2^{(6.9)}$ | $66.9^{(5.2)}$ | $58.3^{(6.5)}$ | $\mathbf{86.6^{(2.1)}}$ | $20.6^{(1.4)}$ | $64.4^{(8.3)}$ | 2.5 | 7.1 |
| BO-qEI | $48.5^{(1.5)}$ | $59.9^{(2.0)}$ | $\mathbf{63.3^{(0.0)}}$ | $\mathbf{86.7^{(0.6)}}$ | $\mathbf{28.7^{(1.8)}}$ | $72.4^{(1.8)}$ | 1.8 | 10.0 |
| DynAMO-BO-qEI | $\mathbf{51.5^{(0.9)}}$ | $65.6^{(3.1)}$ | $\mathbf{63.3^{(0.0)}}$ | $\mathbf{86.7^{(0.6)}}$ | $23.5^{(2.4)}$ | $\mathbf{77.0^{(0.7)}}$ | **1.7** | **11.3** |
| Mixed $\chi^2$ DynAMO-BO-qEI | $44.8^{(1.4)}$ | $\mathbf{66.1^{(3.0)}}$ | $\mathbf{63.3^{(0.0)}}$ | $86.4^{(0.6)}$ | $\mathbf{27.7^{(3.5)}}$ | $73.9^{(2.8)}$ | 2.3 | 10.4 |
| BO-qUCB | $\mathbf{50.3^{(1.8)}}$ | $62.1^{(3.4)}$ | $\mathbf{63.3^{(0.0)}}$ | $\mathbf{86.6^{(0.6)}}$ | $\mathbf{31.7^{(1.2)}}$ | $74.4^{(0.6)}$ | **1.5** | **11.5** |
| DynAMO-BO-qUCB | $48.8^{(1.8)}$ | $65.9^{(3.7)}$ | $\mathbf{63.3^{(0.0)}}$ | $\mathbf{86.5^{(0.5)}}$ | $22.7^{(2.0)}$ | $50.4^{(14.6)}$ | 2.0 | 6.3 |
| Mixed $\chi^2$ DynAMO-BO-qUCB | $44.0^{(0.7)}$ | $\mathbf{68.2^{(3.0)}}$ | $\mathbf{63.3^{(0.0)}}$ | $\mathbf{86.5^{(0.6)}}$ | $20.9^{(1.3)}$ | $\mathbf{74.5^{(2.0)}}$ | 2.0 | 9.6 |
| CMA-ES | $\mathbf{50.7^{(2.7)}}$ | $\mathbf{71.7^{(10.4)}}$ | $\mathbf{63.3^{(0.0)}}$ | $83.9^{(1.0)}$ | $\mathbf{37.9^{(0.7)}}$ | $59.3^{(10.9)}$ | **1.5** | **11.2** |
| DynAMO-CMA-ES | $45.3^{(2.4)}$ | $65.8^{(8.9)}$ | $59.3^{(3.8)}$ | $\mathbf{99.0^{(12.1)}}$ | $22.5^{(5.1)}$ | $60.6^{(15.0)}$ | 2.2 | 8.8 |
| Mixed $\chi^2$ DynAMO-CMA-ES | $\mathbf{48.5^{(3.0)}}$ | $\mathbf{70.0^{(6.5)}}$ | $\mathbf{63.2^{(0.3)}}$ | $\mathbf{87.0^{(2.0)}}$ | $19.4^{(3.8)}$ | $43.7^{(14.6)}$ | 2.3 | 5.4 |
| CoSyNE | $55.3^{(8.0)}$ | $53.6^{(10.2)}$ | $\mathbf{60.8^{(3.2)}}$ | $\mathbf{87.4^{(16.6)}}$ | $\mathbf{36.6^{(4.4)}}$ | $59.3^{(14.5)}$ | 2.3 | 8.9 |
| DynAMO-CoSyNE | $53.8^{(11.0)}$ | $63.4^{(11.5)}$ | $59.3^{(3.8)}$ | $\mathbf{99.0^{(12.1)}}$ | $20.5^{(5.8)}$ | $60.6^{(15.0)}$ | 2.3 | 9.5 |
| Mixed $\chi^2$ DynAMO-CoSyNE | $\mathbf{59.9^{(9.8)}}$ | $\mathbf{65.4^{(9.9)}}$ | $\mathbf{60.9^{(3.1)}}$ | $\mathbf{89.3^{(14.8)}}$ | $23.4^{(4.5)}$ | $66.6^{(5.9)}$ | **1.3** | **11.0** |

*Table A5.* **Diversity of Design Candidates Using Mixed $\chi^2$-Divergence DynAMO.** Using **Corollary D.3** and (60), we show that it is possible to extend DynAMO to leverage a *mixed $\chi^2$-divergence* that equally weights both $\chi^2$-divergence and KL-divergence to penalize the original MBO objective. We evaluate this specialized implementation of DynAMO against baseline DynAMO and vanilla optimization methods, and report the pairwise diversity (resp., minimum novelty and $L_1$ coverage) oracle score achieved by the 128 evaluated designs in the top (resp., middle and bottom) table. Metrics are reported mean$^{(95\% \text{ confidence interval})}$ across 10 random seeds, where higher is better. **Bolded** entries indicate average scores with an overlapping 95% confidence interval to the best performing method. **Bolded** (resp., Underlined) Rank and Optimality Gap (Opt. Gap) metrics indicate the best (resp., second best) for a given backbone optimizer.

| Pairwise Diversity@128 | TFBind8 | UTR | ChEMBL | Molecule | Superconductor | D'Kitty | Rank ↓ | Opt. Gap ↑ |
|---|---|---|---|---|---|---|---|---|
| Dataset $\mathcal{D}$ | 65.9 | 57.3 | 60.0 | 36.7 | 66.0 | 85.7 | — | — |
| Grad. | $12.5^{(8.0)}$ | $7.8^{(8.8)}$ | $7.9^{(7.8)}$ | $24.1^{(13.3)}$ | $0.0^{(0.0)}$ | $0.0^{(0.0)}$ | 3.0 | -53.2 |
| DynAMO-Grad. | $\mathbf{66.9^{(6.9)}}$ | $\mathbf{68.2^{(10.8)}}$ | $\mathbf{77.2^{(21.5)}}$ | $\mathbf{93.0^{(1.2)}}$ | $\mathbf{129^{(55.3)}}$ | $\mathbf{104^{(56.1)}}$ | **1.3** | **27.8** |
| Mixed $\chi^2$ DynAMO-Grad. | $16.8^{(12.6)}$ | $\mathbf{72.6^{(1.1)}}$ | $47.0^{(31.1)}$ | $\mathbf{91.0^{(1.6)}}$ | $\mathbf{182^{(45.9)}}$ | $\mathbf{74.2^{(3.0)}}$ | 1.7 | 18.7 |
| Adam | $12.0^{(12.3)}$ | $11.0^{(12.1)}$ | $4.8^{(3.8)}$ | $16.8^{(12.4)}$ | $6.4^{(14.5)}$ | $6.2^{(14.0)}$ | 3.0 | -52.4 |
| DynAMO-Adam | $\mathbf{54.8^{(8.9)}}$ | $\mathbf{72.3^{(3.4)}}$ | $\mathbf{84.8^{(9.2)}}$ | $\mathbf{89.9^{(5.3)}}$ | $\mathbf{158^{(37.3)}}$ | $\mathbf{126^{(57.3)}}$ | 1.7 | **35.7** |
| Mixed $\chi^2$ DynAMO-Adam | $16.8^{(12.6)}$ | $\mathbf{72.6^{(1.1)}}$ | $\mathbf{99.2^{(0.7)}}$ | $\mathbf{91.0^{(1.6)}}$ | $\mathbf{182^{(45.9)}}$ | $\mathbf{74.2^{(3.0)}}$ | **1.3** | 27.4 |
| BO-qEI | $73.7^{(0.6)}$ | $73.8^{(0.5)}$ | $\mathbf{99.3^{(0.1)}}$ | $\mathbf{93.0^{(0.5)}}$ | $190^{(0.8)}$ | $124^{(7.4)}$ | 2.5 | 47.1 |
| DynAMO-BO-qEI | $\mathbf{74.8^{(0.2)}}$ | $\mathbf{74.6^{(0.3)}}$ | $\mathbf{99.4^{(0.1)}}$ | $\mathbf{93.5^{(0.4)}}$ | $\mathbf{198^{(1.9)}}$ | $\mathbf{277^{(59.7)}}$ | **1.3** | **74.2** |
| Mixed $\chi^2$ DynAMO-BO-qEI | $73.6^{(0.7)}$ | $74.3^{(0.5)}$ | $\mathbf{99.4^{(0.0)}}$ | $\mathbf{93.9^{(0.4)}}$ | $167^{(41.5)}$ | $29.1^{(7.7)}$ | 2.2 | 27.5 |
| BO-qUCB | $\mathbf{73.9^{(0.5)}}$ | $\mathbf{74.3^{(0.4)}}$ | $\mathbf{99.4^{(0.1)}}$ | $\mathbf{93.6^{(0.5)}}$ | $198^{(10.3)}$ | $94.1^{(3.9)}$ | 2.2 | 43.5 |
| DynAMO-BO-qUCB | $\mathbf{74.3^{(0.5)}}$ | $\mathbf{74.4^{(0.6)}}$ | $99.3^{(0.1)}$ | $\mathbf{93.5^{(0.6)}}$ | $\mathbf{211^{(22.8)}}$ | $\mathbf{175^{(44.7)}}$ | **1.7** | **59.4** |
| Mixed $\chi^2$ DynAMO-BO-qUCB | $73.4^{(0.7)}$ | $\mathbf{74.3^{(0.4)}}$ | $\mathbf{99.5^{(0.1)}}$ | $\mathbf{93.7^{(0.5)}}$ | $177^{(25.2)}$ | $28.1^{(7.3)}$ | 2.2 | 29.0 |
| CMA-ES | $47.2^{(11.2)}$ | $44.6^{(15.9)}$ | $\mathbf{93.5^{(2.0)}}$ | $66.2^{(9.4)}$ | $12.8^{(0.6)}$ | $164^{(10.6)}$ | 2.5 | 9.5 |
| DynAMO-CMA-ES | $\mathbf{73.6^{(0.6)}}$ | $\mathbf{73.1^{(3.1)}}$ | $72.0^{(3.1)}$ | $\mathbf{94.0^{(0.5)}}$ | $97.8^{(13.2)}$ | $\mathbf{292^{(83.5)}}$ | **1.3** | **55.2** |
| Mixed $\chi^2$ DynAMO-CMA-ES | $52.9^{(20.8)}$ | $51.5^{(22.1)}$ | $67.7^{(24.7)}$ | $69.7^{(12.4)}$ | $\mathbf{154^{(107)}}$ | $86.9^{(72.9)}$ | 2.2 | 18.6 |
| CoSyNE | $5.6^{(5.0)}$ | $12.7^{(9.8)}$ | $28.2^{(11.3)}$ | $12.2^{(7.3)}$ | $0.0^{(0.0)}$ | $0.0^{(0.0)}$ | 3.0 | -52.1 |
| DynAMO-CoSyNE | $\mathbf{18.1^{(13.0)}}$ | $20.3^{(2.3)}$ | $\mathbf{35.0^{(17.9)}}$ | $\mathbf{22.8^{(11.9)}}$ | $\mathbf{74.4^{(46.3)}}$ | $\mathbf{77.0^{(35.9)}}$ | 1.7 | -20.7 |
| Mixed $\chi^2$ DynAMO-CoSyNE | $\mathbf{22.4^{(8.1)}}$ | $\mathbf{34.1^{(20.1)}}$ | $\mathbf{34.1^{(20.1)}}$ | $\mathbf{23.0^{(17.4)}}$ | $\mathbf{104^{(77.9)}}$ | $\mathbf{49.4^{(25.6)}}$ | **1.3** | **-17.5** |

| Minimum Novelty@128 | TFBind8 | UTR | ChEMBL | Molecule | Superconductor | D'Kitty | Rank ↓ | Opt. Gap ↑ |
|---|---|---|---|---|---|---|---|---|
| Dataset $\mathcal{D}$ | 0.0 | 0.0 | 0.0 | 0.0 | 0.0 | 0.0 | — | — |
| Grad. | $\mathbf{21.2^{(3.0)}}$ | $\mathbf{51.7^{(2.9)}}$ | $\mathbf{97.4^{(3.9)}}$ | $\mathbf{79.5^{(19.7)}}$ | $\mathbf{95.0^{(0.7)}}$ | $\mathbf{102^{(6.1)}}$ | 2.3 | 74.5 |
| DynAMO-Grad. | $\mathbf{21.1^{(1.1)}}$ | $\mathbf{52.2^{(1.3)}}$ | $\mathbf{98.6^{(1.5)}}$ | $\mathbf{85.8^{(1.0)}}$ | $\mathbf{95.0^{(0.4)}}$ | $\mathbf{107^{(6.7)}}$ | **1.3** | **76.7** |
| Mixed $\chi^2$ DynAMO-Grad. | $14.6^{(2.8)}$ | $\mathbf{51.9^{(0.4)}}$ | $\mathbf{99.2^{(0.7)}}$ | $\mathbf{85.2^{(2.1)}}$ | $85.2^{(1.6)}$ | $34.0^{(1.1)}$ | 2.3 | 61.7 |
| Adam | $\mathbf{23.7^{(2.8)}}$ | $\mathbf{51.1^{(3.5)}}$ | $\mathbf{95.5^{(5.3)}}$ | $\mathbf{79.3^{(21.2)}}$ | $94.8^{(0.7)}$ | $\mathbf{103^{(6.3)}}$ | 2.0 | 74.5 |
| DynAMO-Adam | $14.7^{(1.9)}$ | $46.2^{(0.5)}$ | $\mathbf{98.7^{(1.2)}}$ | $\mathbf{85.9^{(1.8)}}$ | $\mathbf{94.9^{(0.4)}}$ | $\mathbf{108^{(7.2)}}$ | **1.7** | **74.7** |
| Mixed $\chi^2$ DynAMO-Adam | $20.4^{(3.3)}$ | $\mathbf{51.9^{(0.4)}}$ | $87.3^{(57.9)}$ | $\mathbf{85.2^{(2.1)}}$ | $85.2^{(1.6)}$ | $34.0^{(1.1)}$ | 2.3 | 60.7 |
| BO-qEI | $\mathbf{21.8^{(0.5)}}$ | $51.5^{(0.3)}$ | $\mathbf{97.6^{(0.3)}}$ | $\mathbf{85.4^{(1.5)}}$ | $\mathbf{94.6^{(0.1)}}$ | $106^{(2.9)}$ | **1.8** | 76.2 |
| DynAMO-BO-qEI | $21.0^{(0.5)}$ | $\mathbf{51.9^{(0.2)}}$ | $97.4^{(0.4)}$ | $85.2^{(0.9)}$ | $\mathbf{94.8^{(0.1)}}$ | $\mathbf{126^{(14.6)}}$ | 2.0 | **79.4** |
| Mixed $\chi^2$ DynAMO-BO-qEI | $14.6^{(0.5)}$ | $\mathbf{51.9^{(0.4)}}$ | $97.4^{(0.5)}$ | $\mathbf{85.5^{(1.3)}}$ | $80.8^{(3.7)}$ | $16.6^{(4.5)}$ | 2.2 | 57.8 |
| BO-qUCB | $\mathbf{21.6^{(0.3)}}$ | $\mathbf{51.7^{(0.2)}}$ | $\mathbf{97.9^{(0.4)}}$ | $85.3^{(1.1)}$ | $93.8^{(0.6)}$ | $98.8^{(1.1)}$ | **1.8** | 74.8 |
| DynAMO-BO-qUCB | $\mathbf{21.4^{(0.5)}}$ | $\mathbf{51.7^{(0.2)}}$ | $97.1^{(0.5)}$ | $85.3^{(1.1)}$ | $\mathbf{94.7^{(0.2)}}$ | $\mathbf{109^{(4.5)}}$ | 2.0 | **76.6** |
| Mixed $\chi^2$ DynAMO-BO-qUCB | $19.8^{(0.3)}$ | $\mathbf{52.0^{(0.1)}}$ | $97.4^{(0.4)}$ | $\mathbf{85.6^{(1.3)}}$ | $79.8^{(3.5)}$ | $14.9^{(3.3)}$ | 2.2 | 58.2 |
| CMA-ES | $16.5^{(2.1)}$ | $47.8^{(1.0)}$ | $96.5^{(0.7)}$ | $\mathbf{73.0^{(18.0)}}$ | $\mathbf{100^{(0.0)}}$ | $\mathbf{100^{(0.0)}}$ | 2.3 | 72.3 |
| DynAMO-CMA-ES | $12.9^{(0.8)}$ | $48.0^{(1.6)}$ | $96.7^{(3.5)}$ | $\mathbf{81.8^{(13.4)}}$ | $94.5^{(0.7)}$ | $\mathbf{112^{(7.8)}}$ | 2.0 | **74.3** |
| Mixed $\chi^2$ DynAMO-CMA-ES | $\mathbf{23.0^{(1.6)}}$ | $\mathbf{51.8^{(0.4)}}$ | $\mathbf{98.0^{(0.9)}}$ | $\mathbf{83.8^{(9.9)}}$ | $87.1^{(2.1)}$ | $48.6^{(16.8)}$ | **1.7** | 65.4 |
| CoSyNE | $\mathbf{24.5^{(3.5)}}$ | $49.7^{(3.1)}$ | $\mathbf{98.5^{(1.6)}}$ | $\mathbf{86.6^{(12.7)}}$ | $93.2^{(1.0)}$ | $91.9^{(2.0)}$ | **1.7** | 74.1 |
| DynAMO-CoSyNE | $17.8^{(5.5)}$ | $48.4^{(2.3)}$ | $96.7^{(3.5)}$ | $\mathbf{80.2^{(12.9)}}$ | $\mathbf{94.5^{(0.7)}}$ | $\mathbf{112^{(7.8)}}$ | 2.3 | **75.0** |
| Mixed $\chi^2$ DynAMO-CoSyNE | $\mathbf{23.5^{(3.7)}}$ | $\mathbf{52.5^{(2.0)}}$ | $\mathbf{99.9^{(0.1)}}$ | $\mathbf{80.6^{(21.5)}}$ | $83.9^{(2.6)}$ | $30.3^{(4.2)}$ | 2.0 | 61.8 |

| $L_1$ Coverage@128 | TFBind8 | UTR | ChEMBL | Molecule | Superconductor | D'Kitty | Rank ↓ | Opt. Gap ↑ |
|---|---|---|---|---|---|---|---|---|
| Dataset $\mathcal{D}$ | 0.42 | 0.31 | 1.42 | 0.68 | 6.26 | 0.58 | — | — |
| Grad. | $0.16^{(0.10)}$ | $0.20^{(0.13)}$ | $0.21^{(0.10)}$ | $0.42^{(0.18)}$ | $0.00^{(0.00)}$ | $0.00^{(0.00)}$ | 3.0 | -1.44 |
| DynAMO-Grad. | $\mathbf{0.36^{(0.04)}}$ | $\mathbf{0.52^{(0.06)}}$ | $\mathbf{1.46^{(0.38)}}$ | $\mathbf{2.49^{(0.06)}}$ | $\mathbf{6.47^{(1.24)}}$ | $\mathbf{5.85^{(1.35)}}$ | 1.3 | 1.25 |
| Mixed $\chi^2$ DynAMO-Grad. | $0.16^{(0.09)}$ | $\mathbf{0.54^{(0.02)}}$ | $0.87^{(0.58)}$ | $2.20^{(0.10)}$ | $\mathbf{6.67^{(1.68)}}$ | $1.61^{(0.03)}$ | 1.7 | 0.40 |
| Adam | $0.11^{(0.06)}$ | $0.22^{(0.09)}$ | $0.23^{(0.15)}$ | $0.48^{(0.31)}$ | $0.27^{(0.55)}$ | $0.24^{(0.49)}$ | 3.0 | -1.35 |
| DynAMO-Adam | $\mathbf{0.33^{(0.05)}}$ | $\mathbf{0.55^{(0.03)}}$ | $\mathbf{1.44^{(0.39)}}$ | $\mathbf{2.40^{(0.16)}}$ | $\mathbf{7.06^{(0.73)}}$ | $\mathbf{6.91^{(0.71)}}$ | 1.0 | 1.50 |
| Mixed $\chi^2$ DynAMO-Adam | $0.16^{(0.09)}$ | $\mathbf{0.54^{(0.02)}}$ | $0.47^{(0.31)}$ | $2.20^{(0.10)}$ | $\mathbf{6.67^{(1.68)}}$ | $1.61^{(0.03)}$ | 2.0 | 0.33 |
| BO-qEI | $\mathbf{0.41^{(0.02)}}$ | $\mathbf{0.55^{(0.01)}}$ | $2.37^{(0.03)}$ | $2.11^{(0.15)}$ | $7.84^{(0.01)}$ | $6.61^{(0.33)}$ | 2.3 | 1.70 |
| DynAMO-BO-qEI | $\mathbf{0.42^{(0.01)}}$ | $\mathbf{0.56^{(0.01)}}$ | $\mathbf{2.47^{(0.03)}}$ | $\mathbf{2.54^{(0.03)}}$ | $\mathbf{7.87^{(0.01)}}$ | $\mathbf{7.92^{(0.04)}}$ | 1.2 | 2.02 |
| Mixed $\chi^2$ DynAMO-BO-qEI | $0.39^{(0.02)}$ | $\mathbf{0.55^{(0.01)}}$ | $2.39^{(0.04)}$ | $\mathbf{2.56^{(0.02)}}$ | $6.11^{(0.64)}$ | $1.36^{(0.12)}$ | 2.5 | 0.62 |
| BO-qUCB | $\mathbf{0.40^{(0.02)}}$ | $0.54^{(0.01)}$ | $\mathbf{2.40^{(0.05)}}$ | $2.52^{(0.07)}$ | $7.78^{(0.04)}$ | $6.64^{(0.09)}$ | 2.5 | 1.77 |
| DynAMO-BO-qUCB | $\mathbf{0.40^{(0.02)}}$ | $\mathbf{0.55^{(0.01)}}$ | $2.47^{(0.07)}$ | $\mathbf{2.54^{(0.05)}}$ | $\mathbf{7.88^{(0.03)}}$ | $\mathbf{7.80^{(0.23)}}$ | 1.2 | 2.00 |
| Mixed $\chi^2$ DynAMO-BO-qUCB | $0.39^{(0.02)}$ | $\mathbf{0.56^{(0.01)}}$ | $\mathbf{2.41^{(0.05)}}$ | $\mathbf{2.53^{(0.06)}}$ | $5.96^{(0.78)}$ | $1.38^{(0.11)}$ | 2.3 | 0.59 |
| CMA-ES | $\mathbf{0.33^{(0.05)}}$ | $0.48^{(0.04)}$ | $\mathbf{2.18^{(0.04)}}$ | $1.82^{(0.12)}$ | $3.26^{(1.42)}$ | $\mathbf{3.77^{(1.36)}}$ | 2.3 | 0.36 |
| DynAMO-CMA-ES | $\mathbf{0.40^{(0.03)}}$ | $\mathbf{0.56^{(0.01)}}$ | $1.82^{(0.72)}$ | $\mathbf{2.54^{(0.05)}}$ | $\mathbf{4.75^{(2.16)}}$ | $\mathbf{3.29^{(1.56)}}$ | 1.7 | 0.62 |
| Mixed $\chi^2$ DynAMO-CMA-ES | $0.30^{(0.09)}$ | $\mathbf{0.52^{(0.05)}}$ | $1.56^{(0.68)}$ | $1.97^{(0.19)}$ | $\mathbf{5.58^{(1.68)}}$ | $\mathbf{4.03^{(3.01)}}$ | 2.0 | 0.71 |
| CoSyNE | $0.10^{(0.07)}$ | $\mathbf{0.22^{(0.10)}}$ | $0.39^{(0.20)}$ | $0.27^{(0.13)}$ | $0.10^{(0.00)}$ | $0.10^{(0.00)}$ | 2.8 | -1.41 |
| DynAMO-CoSyNE | $\mathbf{0.21^{(0.11)}}$ | $0.18^{(0.09)}$ | $\mathbf{0.64^{(0.42)}}$ | $\mathbf{0.43^{(0.34)}}$ | $\mathbf{1.85^{(0.22)}}$ | $\mathbf{0.94^{(0.17)}}$ | 2.0 | -0.90 |
| Mixed $\chi^2$ DynAMO-CoSyNE | $\mathbf{0.20^{(0.04)}}$ | $\mathbf{0.37^{(0.14)}}$ | $\mathbf{0.74^{(0.47)}}$ | $\mathbf{0.53^{(0.42)}}$ | $\mathbf{3.85^{(2.79)}}$ | $\mathbf{1.10^{(0.55)}}$ | **1.2** | **-0.48** |

where $n := |\mathcal{D}|$ is the number of datums in the offline dataset $\mathcal{D}$.

*Proof.* Under **Assumption D.4**, Jensen's inequality gives us

$$\mathbb{E}_{x \sim p^\tau(x)} |r(x) - r_\theta(x)| \leq \sqrt{\mathbb{E}_{x \sim p^\tau(x)}[r(x) - r_\theta(x)]^2} \leq \sqrt{\frac{\varepsilon_0^2}{4}} =: \frac{\varepsilon_0}{2} \tag{73}$$

Furthermore, **Assumption D.6** and Cortes et al. (2010) yield

$$\begin{aligned}
\left| \mathbb{E}_{x \sim p^\tau(x)} |r(x) - r_\theta(x)| - \mathbb{E}_{x \sim q^\pi(x)} |r(x) - r_\theta(x)| \right| & \\
= \left| \mathbb{E}_{x \sim p^\tau(x)} |r(x) - r_\theta(x)| - \mathbb{E}_{x \sim p_{\mathcal{D}}^\tau(x)} \left[ \frac{q^\pi(x)}{p_{\mathcal{D}}^\tau(x)} \Big| r(x) - r_\theta(x) \Big| \right] \right| & \\
\leq M \sqrt{\frac{2 \log(2|\Pi|/\delta)}{n}} &
\end{aligned} \tag{74}$$

with probability at least $1 - \delta$. In the offline setting (as in our work) and assuming that the forward surrogate model $r_\theta(x)$ has been well-trained according to (2) or a similar learning paradigm (e.g., see Trabucco et al. (2021); Yu et al. (2021)), we can reasonably assume that $\varepsilon_0^2 \leq \mathbb{E}_{x \sim q^\pi(x)}[r(x) - r_\theta(x)]^2$. We therefore have an upper bound on the prediction error of the forward surrogate model over the distribution $q^\pi(x)$ over generated designs:

$$\mathbb{E}_{x \sim q^\pi(x)} |r(x) - r_\theta(x)| \leq \mathbb{E}_{x \sim p^\tau(x)} |r(x) - r_\theta(x)| + M \sqrt{\frac{2 \log(2|\Pi|/\delta)}{n}} \leq \frac{\varepsilon_0}{2} + M \sqrt{\frac{2 \log(2|\Pi|/\delta)}{n}} \tag{75}$$

with probability at least $1 - \delta$. $\qquad \square$

Under **Assumption D.6**, we can also place an upper bound on the true and realized KL-divergence penalties:

**Lemma D.8** (Bounded KL-Divergence). *Assume there exists an $M \in \mathbb{R}_+$ finite satisfying **Assumption D.6**. Then with probability at least $1 - \delta$ we have (for any $\delta > 0$ and for both $\pi = \pi^\star$ and $\pi = \hat{\pi}$)*

$$|D_{KL}(q^\pi(x)||p^\tau(x)) - D_{KL}(q^\pi(x)||p_{\mathcal{D}}^\tau(x))| \leq M \sqrt{\log(|\Pi|/\delta)} \tag{76}$$

*Proof.* According to the definition of $M$ and the definition of the KL-divergence from **Definition C.1**,

$$D_{\text{KL}}(q^\pi(x)||p^\tau(x)) = \mathbb{E}_{x \sim p^\tau(x)} \left[ \frac{q^\pi(x)}{p^\tau(x)} \log \left( \frac{q^\pi(x)}{p^\tau(x)} \right) \right] \leq M \log M \tag{77}$$

From Hoeffding's inequality (Hoeffding, 1963),

$$\mathbb{P}\left( |D_{\text{KL}}(q^\pi(x)||p^\tau(x)) - D_{\text{KL}}(q^\pi(x)||p_{\mathcal{D}}^\tau(x))| \geq \varepsilon \right) \leq |\Pi| \cdot \exp \left( -\frac{2\varepsilon^2}{M \log M} \right) \tag{78}$$

for any $\varepsilon > 0$. We can choose to define $\varepsilon := \sqrt{(M \log M) \cdot \log(|\Pi|/\delta)/2}$ such that

$$|D_{\text{KL}}(q^\pi(x)||p^\tau(x)) - D_{\text{KL}}(q^\pi(x)||p_{\mathcal{D}}^\tau(x))| \leq \frac{\sqrt{(M \log M) \cdot \log(|\Pi|/\delta)}}{\sqrt{2}} \leq \frac{M \sqrt{\log(|\Pi|/\delta)}}{\sqrt{2}} \leq M \sqrt{\log(|\Pi|/\delta)} \tag{79}$$

with probability at least $1 - \delta$. $\qquad \square$

We are now ready to prove our main result:

**Theorem D.9** (Bounded Diversity-Penalized Objective $J^\star(\pi)$). *Assume that there exists an $M \in \mathbb{R}_+$ finite satisfying **Assumption D.6**. Then with probability at least $1 - \delta$, we have (for any $\delta > 0$)*

$$J^\star(\pi^\star) - J^\star(\hat{\pi}) \leq \varepsilon_0 + 2M \left( \frac{2}{\sqrt{n}} + \frac{\beta}{\tau} \right) \sqrt{\log \left( \frac{8|\Pi|}{\delta} \right)} \tag{80}$$

*where $n := |\mathcal{D}|$ is the size of the offline dataset $\mathcal{D}$.*

*Proof.* Firstly, we combine **Lemmas D.7** and **D.8** using the triangle inequality to bound the difference between the true reward $J^\star(\hat{\pi})$ and the offline reward $J(\hat{\pi})$, where $\hat{\pi} \in \Pi$ maximizes $J(\pi)$ as defined in (5).

$$
\begin{aligned}
J(\hat{\pi}) - J^\star(\hat{\pi}) &:= \left( \mathbb{E}_{x \sim q^{\hat{\pi}}(x)}[r_\theta(x)] - \frac{\beta}{\tau} D_{\mathrm{KL}}(q^{\hat{\pi}}(x)\|p_\mathcal{D}^\tau) \right) - \left( \mathbb{E}_{x \sim q^{\hat{\pi}}(x)}[r(x)] - \frac{\beta}{\tau} D_{\mathrm{KL}}(q^{\hat{\pi}}(x)\|p^\tau(x)) \right) \\
&\le \mathbb{E}_{x \sim q^{\hat{\pi}}(x)} |r(x) - r_\theta(x)| + \frac{\beta}{\tau} \left| D_{\mathrm{KL}}(q^{\hat{\pi}}(x)\|p^\tau(x)) - D_{\mathrm{KL}}(q^{\hat{\pi}}(x)\|p^\tau(x)) \right| \\
&\le \left( \frac{\varepsilon_0}{2} + M \sqrt{\frac{2 \log(8|\Pi|/\delta)}{n}} \right) + M \cdot \frac{\beta}{\tau} \sqrt{\log(4|\Pi|/\delta)} \\
&\le \frac{\varepsilon_0}{2} + M \left( \frac{2}{\sqrt{n}} + \frac{\beta}{\tau} \right) \sqrt{\log\left( \frac{8|\Pi|}{\delta} \right)}
\end{aligned}
\tag{81}
$$

with probability $1 - (\delta/2)$. Because $\hat{\pi} := \mathrm{argmax}_{\pi \in \Pi} J(\pi)$, we must have $J(\pi^\star) \le J(\hat{\pi})$. Substituting this into the left hand side of (81) gives

$$
J(\pi^\star) - J^\star(\hat{\pi}) \le \frac{\varepsilon_0}{2} + M \left( \frac{2}{\sqrt{n}} + \frac{\beta}{\tau} \right) \sqrt{\log\left( \frac{8|\Pi|}{\delta} \right)}
\tag{82}
$$

Separately, we have (with probability $1 - (\delta/2)$)

$$
\begin{aligned}
J^\star(\pi^\star) - J(\pi^\star) &:= \left( \mathbb{E}_{x \sim q^{\pi^\star}(x)}[r(x)] - \frac{\beta}{\tau} D_{\mathrm{KL}}(q^{\pi^\star}(x)\|p^\tau(x)) \right) - \left( \mathbb{E}_{x \sim q^{\pi^\star}}[r_\theta(x)] - \frac{\beta}{\tau} D_{\mathrm{KL}}(q^{\pi^\star}(x)\|p_\mathcal{D}^\tau(x)) \right) \\
&\le \mathbb{E}_{x \sim q^{\pi^\star}(x)} |r(x) - r_\theta(x)| + \frac{\beta}{\tau} \left| D_{\mathrm{KL}}(q^{\pi^\star}(x)\|p^\tau(x)) - D_{\mathrm{KL}}(q^{\pi^\star}(x)\|p_\mathcal{D}^\tau(x)) \right| \\
&\le \left( \frac{\varepsilon_0}{2} + M \sqrt{\frac{2 \log(8|\Pi|/\delta)}{n}} \right) + M \cdot \frac{\beta}{\tau} \sqrt{\log(|4\Pi|/\delta)} \\
&\le \frac{\varepsilon_0}{2} + M \left( \frac{2}{\sqrt{n}} + \frac{\beta}{\tau} \right) \sqrt{\log\left( \frac{8|\Pi|}{\delta} \right)}
\end{aligned}
\tag{83}
$$

following the derivation in (81) except for $\pi^\star$ (as opposed to $\hat{\pi}$) that maximizes $J^\star(\pi)$ (as opposed to $J(\pi)$). Summing (82) and (83) gives

$$
J^\star(\pi^\star) - J^\star(\hat{\pi}) \le \varepsilon_0 + 2M \left( \frac{2}{\sqrt{n}} + \frac{\beta}{\tau} \right) \sqrt{\log\left( \frac{8|\Pi|}{\delta} \right)}
\tag{84}
$$

with probability $1 - \delta$. □

Note that we only prove **Theorem D.9** in the unconstrained optimization setting; in principle, a tighter bound could exist in the adversarially constrained formulation introduced in (7), as a bound on the 1-Wasserstein distance between $q^{\hat{\pi}}(x)$ and $p_\mathcal{D}^\tau(x)$ will almost surely place a favorably *tighter* bound on the forward surrogate model prediction error than **Lemma D.7**.

### D.5. Comparison with Offline Model-Free Optimization Methods

In our main experimental results reported in **Section 5**, we focus on comparing DynAMO against other *model-based optimization* (MBO) methods—that is, optimization methods that explicitly (1) learn a proxy forward surrogate model $r_\theta(x)$ for the oracle reward function from the offline dataset; and (2) optimize against $r_\theta(x)$ and rank final candidate designs according to a scoring metric involving $r_\theta$. Alternatively, recent work have also proposed methods that instead do *not* learn a forward surrogate model $r_\theta(x)$; we refer to such methods as *model-free* algorithms.

**Survey of Existing Model-Free and Additional Model-Based Methods.** Mashkaria et al. (2023) introduce **BONET** (i.e., **B**lack-box **O**ptimization **Net**works), which learns an auto-regressive model on synthetically constructed optimization trajectories that simulate runs of implicit black-box optimization experiments. The auto-regressive model is trained to learn a rollout of monotonic transitions from low- to high- scoring design candidates using the offline dataset. Nguyen et al. (2023) propose **ExPT** (i.e., **Ex**periment **P**retrained **T**ransformers) as a task-agnostic method of pre-training a transformer

foundation model to learn an inverse modeling of designs from input reward scores and associated contexts. **DDOM** (i.e., **D**enoising **D**iffusion **O**ptimization **M**odels) learns a generative diffusion model conditioned on the oracle reward values in the offline dataset (Krishnamoorthy et al., 2023). Similarly, **GTG** (i.e., **G**uided **T**rajectory **G**eneration) trains a diffusion model to learn from synthetically constructed optimization trajectories conditioned on final scores. **MINs** (i.e., **M**odel **I**nversion **N**etworks) from Kumar & Levine (2019) learn and optimize against an inverse mapping from reward scores to candidate designs.[2] **Tri-Mentoring** and **ICT** (i.e., **I**mportance-aware **C**o-**T**eaching) co-learn an ensemble of multiple surrogate models (Chen et al., 2023a; Yuan et al., 2023). Separately, **PGS** (i.e., **P**olicy-**G**uided **S**earch) from Chemingui et al. (2024) learns a policy to optimize against a surrogate model (although only limit their method to first-order optimization algorithms), and **Match-Opt** from Hoang et al. (2024) proposes a black-box gradient matching algorithm to learn better forward surrogate models. Finally, **RGD** (i.e., **R**obust-**G**uided **D**iffusion) uses a forward surrogate model to guide the generative sampling process from a diffusion model (Chen et al., 2024). Other model-free optimization methods have been proposed specifically for the biological sequence design problems (Kim et al., 2023; Chen et al., 2023b; Jain et al., 2022); we exclude these from our analysis and instead focus on task-agnostic optimization algorithms. We also exclude Design Editing for offline Model-based Optimization (DEMO) from Yuan et al. (2024), Noise-intensified Telescoping density-Ratio Estimation (NTRE) from Yu et al. (2024), and Ranking Models (RaM) from Tan et al. (2024) from our analysis since there are no presently available open-source implementations.

**Experimental Results.** We compare representative implementations of DynAMO (i.e., DynAMO with Gradient Ascent (**DynAMO-Grad.**), Bayesian optimization with Upper Confidence Bound acquisition function (**DynAMO-BO-qUCB**), and Covariance Matrix Adaptation Evolution Strategy (**DynAMO-CMA-ES**)) against other model-based optimization methods using the respective backbone optimizer described by the original authors (i.e., **RoMA** from Yu et al. (2021) using Adam Ascent, **COMs** from Trabucco et al. (2021) using Gradient Ascent, **ROMO** from Chen et al. (2023c) using Gradient Ascent, **GAMBO** from Yao et al. (2024) using BO-qEI) against model-free optimization methods in **Supplementary Tables A6-A8**. We find that DynAMO-augmented optimizers can be competitive in proposing high-quality designs—in particular, DynAMO-BO-qUCB achieves both the second best Rank and Optimality Gap across all six tasks according to the Best@128 oracle score metric. However, the improvement in *diversity* of designs using DynAMO is significant: DynAMO-BO-qUCB achieves the best Rank and Optimality gap according to both the Pairwise Diversity and $L_1$ Coverage metrics, and DynAMO-Grad. achieves the best Rank and Optimality gap according to the Minimum Novelty metric. Furthermore, DynAMO-BO-qUCB attains the best mean Pairwise Diversity score compared to the model-free optimization methods evaluated in 5 out of the 6 tasks assessed. Altogether, our results suggest that DynAMO is a promising technique to propose a diverse set of high-quality designs compared with existing state-of-the-art offline optimization methods.

### D.6. Empirical Computational Cost Analysis

To evaluate the empirical cost associated with running DynAMO, we report both the runtime and maximum GPU utilization of optimization methods both with and without DynAMO augmentation in **Supplementary Table A9**. Briefly, all experimental results reported in **Supplementary Table A9** were conducted on one internal cluster with 8 NVIDIA RTX A6000 GPUs, and one 24-core Intel Xeon CPU—however, only a single GPU was made available for each program instance reported in our experiments. Across all six optimization methods evaluated, DynAMO increases the runtime (resp., maximum GPU usage) of the optimization method (averaged over all six tasks) by a mean of 181.7% (resp. 1.9%). While our experiments reveal that DynAMO is indeed associated with additional computational costs, they also show that DynAMO is empirically tractable to run *even using a single GPU*. Furthermore, we note that as discussed by Yao et al. (2024) and others, the primary real-world application of offline optimization solvers is in generative design tasks where the true oracle reward function is prohibitively expensive or inaccessible. In these settings, we argue that it is often worth leveraging additional compute to use DynAMO (or other offline optimization methods) to generate the best results possible before final oracle evaluation.

### D.7. $\tau$-Weighted Distribution Visualization

In **Definition 3.2**, we define the $\tau$-weighted probability distribution to serve as the reference distribution for a generative policy to learn from in (7). This reference probability distribution is important and should ideally capture the diversity of *high-quality* designs contained in the offline dataset. To investigate if this is indeed the case, we plot the empirical $\tau$-weighted distributions for each of the six offline optimization tasks in our experimental evaluation suite using $\tau = 1.0$,

---

[2]One might argue that MINs (Kumar & Levine, 2019) are also a form of model-based optimization, as the method involves learning a surrogate function $f_\theta^{-1} : \mathbb{R} \to \mathcal{X}$. However, the method proposes a design $x$ given an input score value, and therefore does not make available an output proxy score by which to rank candidate designs. We therefore include MINs as a *model-free* optimization algorithm.

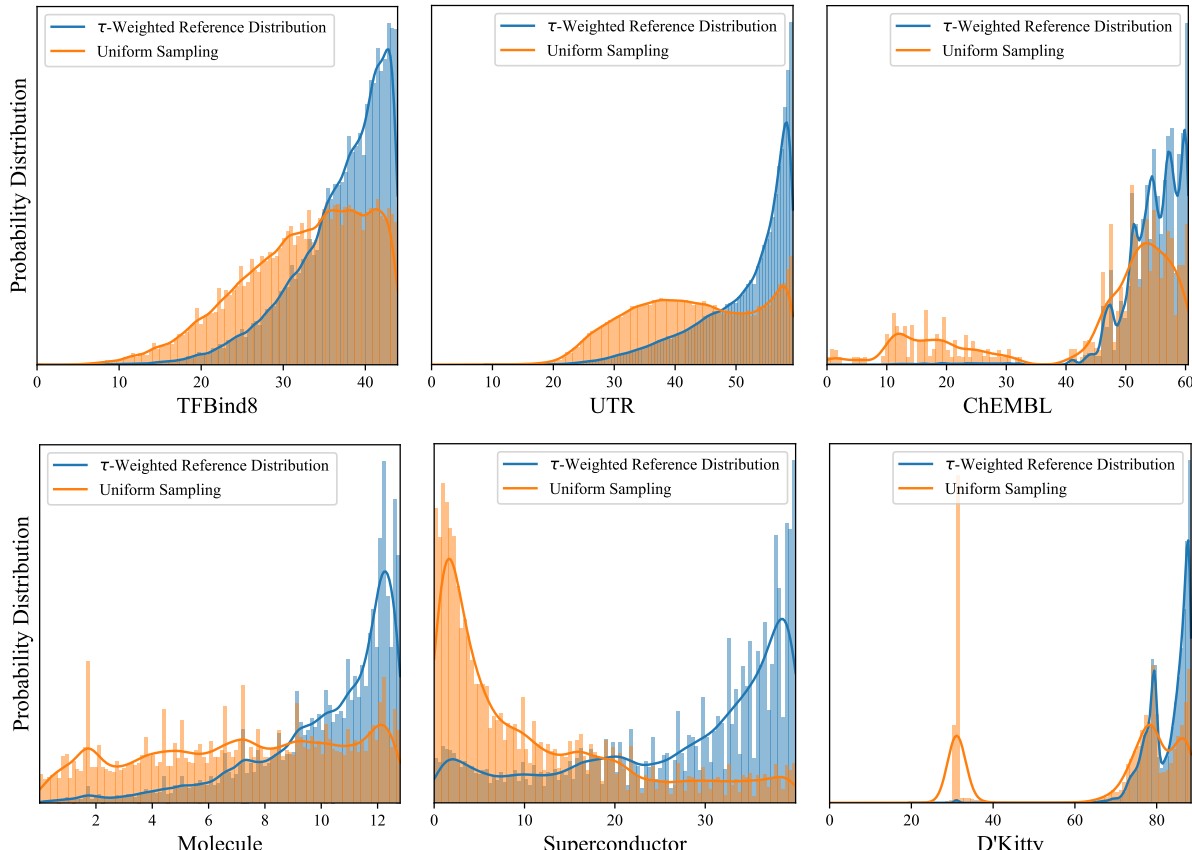

*Figure A1.* **Sample $\tau$-Weighted Probability Distributions.** We plot ($\tau = 1.0$)-weighted distributions $p_{\mathcal{D}}^{\tau}(y)$ (blue) versus the original distribution of oracle scores $y$ in the public offline dataset $\mathcal{D}$ (orange) for the 6 offline optimization tasks in our experimental evaluation suite: (1) **TFBind8** (top left); (2) **UTR** (top middle); (3) **ChEMBL** (top right); (4) **Molecule** (bottom left); (5) **Superconductor** (bottom middle); and (6) **D'Kitty** (bottom right). DynAMO penalizes a model-based optimization objective to encourage sampling policies to match the *diversity* of (high-scoring) designs in the $\tau$-weighted distribution. The $x$-axis represents the normalized oracle scores.

which is the value of the temperature hyperparameter used in our experiments in **Table 1**. The resulting plots are shown in **Figure A1**; in general, we can see that our $\tau$-weighted reference distributions weight optimal and near-optimal designs more heavily (i.e., a distribution with negative skew), while still capturing a variety of different possible designs.

### D.8. Distribution Analysis of Quality and Diversity Results

In our experimental results in the main text and in the Appendices, we primarily focus on reporting summative statistics: for example, the Best@128 oracle score and the average Pairwise Diversity metric over the final batch of $k = 128$ samples. In this section, we isolate a single representative experimental run and plot the distribution of scores achieved by all $k = 128$ designs from a single experimental run to better interrogate the robustness and empirical properties of DynAMO.

In **Supplementary Figure A2**, we first plot the distributions of the oracle reward score $r(x_i^F)$ and minimum novelty score $\min_{x' \in \mathcal{D}} d(x_i^F, x')$ achieved by each of the $k = 128$ designs in the set $\{x_i^F\}_{i=1}^{k}$ proposed by the CMA-ES optimizer with and without DynAMO augmentation in a single experimental run. (Recall that $\mathcal{D}$ is the static, offline dataset of reference designs and $d(\cdot, \cdot)$ is the normalized Levenshtein distance metric for this task.) We see that in general, DynAMO not only enables the optimizer to discover more optimal designs with higher probability, but also yields a wider-tailed distribution of oracle scores compared to the baseline method. Separately, we see that DynAMO augmentation *decreases* both the median and mode Minimum Novelty score compared to the baseline method, in agreement with our discussion in **Appendix D.2**.

In the bottom row of **Supplementary Figure A2**, we visualize a heat map of pairwise diversity scores; that is, the color of pixel $(i, j)$ is correlated with the distance $d(x_i^F, x_j^F)$ for any $1 \le i, j \le k$ pair of generated designs proposed by the

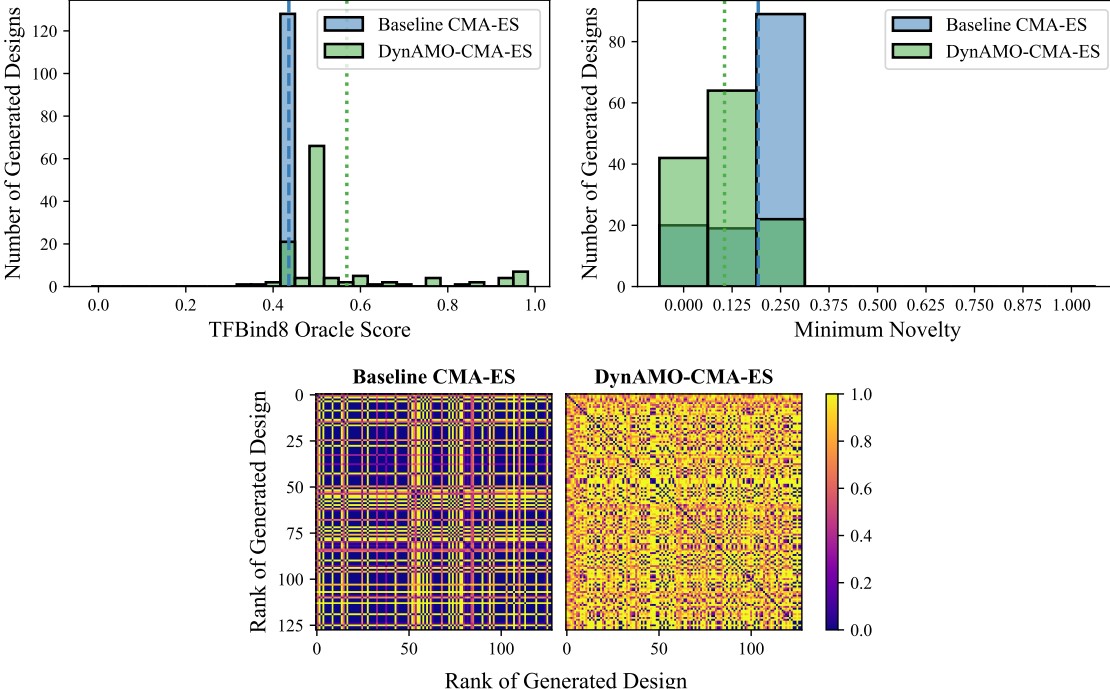

*Figure A2.* **Distribution of Generated Design Quality and Diversity Scores.** We plot the distributions of the (**top left**) oracle score; (**top right**) minimum novelty; and (**bottom**) pairwise diversity of the $k = 128$ proposed designs from a single representative experimental run using the CMA-ES backbone optimizers with and without DynAMO on the TFBind8 task. Dashed blue (resp., dotted green) lines in the top panels represent the mean score achieved by the Baseline CMA-ES (resp., DynAMO-CMA-ES) method from the experimental run.

optimization method. Even a cursory visual inspection reveals that DynAMO augmentation of the backbone CMA-ES optimizer significance improves the pairwise diversity of candidate designs when compared to the baseline method.

### D.9. Why Is Diversity Important?

Our principle motivation for obtaining a diverse sample of designs in offline MBO is to enable downstream **secondary exploration** of other objectives that we might care about in real-world applications. For example, given a batch of proposed candidates drugs that were optimized for maximal therapeutic efficacy in treating a disease, we might then try to quantify each candidate's manufacturing cost, difficulty of synthesize, profile of potential side effects, and other objectives. In this setting, obtaining highly similar designs from offline MBO may result in strong therapeutic efficacy, but also *equally* unacceptable values of other secondary objectives.

To validate this motivating claim that diversity is important to obtain a wide range of secondary objective values, we compare the range and variance of secondary objective values within a batch of proposed candidate designs. We consider the following 3 offline MBO tasks:

1. **Vehicle Safety** is continuous, 5-dimensional optimization problem from Liao et al. (2008) to find an optimal set of car dimensions that minimize the total **Mass** of the vehicle. The problem initially stems from the multi-objective optimization literature (Blank & Deb, 2020; Liao et al., 2008; Huo et al., 2022; Gonzalez de Oliveira et al., 2023), where the secondary goals are to (1) minimize the worst-case **Acceleration**-induced biomechanical damage of the car occupants in the event of a collision; and (2) minimize the worst-case toe board **Intrusion** of the vehicle in the event of an 'offset-frontal crash.' We treat the Mass as the offline MBO optimization objective and Acceleration and Intrusion as the downstream secondary objectives. We negate all objective values prior to max-min normalization as described in **Appendix B** to frame this as a *maximization* problem in accordance with the setup in **Table 1**. An offline dataset of $n = 800$ designs was synthetically constructed, and we used the oracle function from Liao et al. (2008) to compute all 3 objective values.

2. **Welded Beam** is a continuous, 4-dimensional optimization problem from Ray & Liew (2002) to find an optimal set of dimensions for a welded steel beam that minimizes the total manufacturing **Cost**. Similar to the Vehicle task, this problem was initially proposed in the multi-objective optimization literature (Blank & Deb, 2020; Liao et al., 2008; Kamil et al., 2021; Deb et al., 2006) where the secondary goal is to (1) minimize the end **Deflection** of the beam.[3] Again, we negate all objective values prior to max-min normalization as described in **Appendix B** to frame this as a maximization problem. An offline dataset of $n = 800$ designs was synthetically constructed, and we used the oracle function from Ray & Liew (2002) to compute both objective values.

3. **UTR** is a discrete, 50-dimensional optimization problem from Angermüeller et al. (2020) and Sample et al. (2019) with the goal of finding an optimal 50-bp DNA sequence that maximizes the gene expression from a 5' UTR DNA sequence. This is an offline MBO problem from the Design-Bench benchmarking suite (Trabucco et al., 2022) that we use to evaluate offline MBO algorithms in our main experimental results in **Table 1** and elsewhere. However, a secondary objective is to minimize the **GC Content** of the resulting DNA sequence, which is correlated with the difficulty of cloning and sequencing the DNA sequence using standard DNA amplification and analysis methods in the laboratory setting (Benita et al., 2003; Yakovchuk et al., 2006; Gardner et al., 2002). To evaluate this secondary objective, we use the same experimental setting as for the initial UTR experiments described in **Appendix B** and evaluate the GC Content of the $k = 128$ proposed designs as the secondary objective according to Benita et al. (2003).

We used the standard deviation of secondary objective values achieved by a proposed set of designs to quantify the range of secondary objective values, and the pairwise diversity metric (PD@128) to quantify the diversity of designs. We evaluated both baseline and DynAMO-enhanced optimization methods on the three tasks above (**Supplementary Table A10**). Our results consistently demonstrate that a greater diversity score of the final proposed designs (i.e., higher PD@128 score) is correlated with a greater range of captured secondary objective values. As a result, a diverse set of designs (such as those proposed by DynAMO-enhanced optimization methods) can better enable downstream evaluation of the trade-offs between different objectives for a given design.

---

[3]The original problem from Ray & Liew (2002) was proposed as a constrained optimization problem with 5 sets of constraints on the maximum considered shear stress, bending stress, buckling load, and other material testing parameters. To simplify our experimental setting, we consider the *unconstrained* version of the optimization problem here.

*Table A6.* **Comparison of Design Quality Against Model-Free Optimization Methods.** We evaluate DynAMO and other model-based optimization methods against model-free optimization methods. We report the maximum (resp., median) oracle score achieved out of 128 evaluated designs in the top (resp., bottom) table. Metrics are reported mean$^{(95\% \text{ confidence interval})}$ across 10 random seeds, where higher is better. $\max(\mathcal{D})$ reports the top oracle score in the offline dataset. All metrics are multiplied by 100 for easier legibility. **Bolded** entries indicate average scores with an overlapping 95% confidence interval to the best performing method. **Bolded** (resp., Underlined) Rank and Optimality Gap (Opt. Gap) metrics indicate the best (resp., second best) for a given backbone optimizer.

| Best@128 | TFBind8 | UTR | ChEMBL | Molecule | Superconductor | D'Kitty | Rank ↓ | Opt. Gap ↑ |
|---|---|---|---|---|---|---|---|---|
| Dataset $\mathcal{D}$ | 43.9 | 59.4 | 60.5 | 88.9 | 40.0 | 88.4 | — | — |
| Grad. | $90.0^{(4.3)}$ | $80.9^{(12.1)}$ | $60.2^{(8.9)}$ | $88.8^{(4.0)}$ | $36.0^{(6.8)}$ | $65.6^{(14.5)}$ | 16.0 | 6.8 |
| BO-qUCB | $88.1^{(5.3)}$ | $86.2^{(0.1)}$ | $66.4^{(0.7)}$ | $\mathbf{121^{(1.3)}}$ | $\mathbf{51.3^{(3.6)}}$ | $84.5^{(0.8)}$ | 7.3 | 19.4 |
| CMA-ES | $87.6^{(8.3)}$ | $86.2^{(0.0)}$ | $66.1^{(1.0)}$ | $106^{(5.9)}$ | $49.0^{(1.0)}$ | $72.2^{(0.1)}$ | 10.2 | 14.4 |
| BONET | $95.5^{(0.0)}$ | $\mathbf{92.9^{(0.1)}}$ | $63.3^{(0.0)}$ | $97.3^{(0.0)}$ | $39.0^{(0.7)}$ | $93.7^{(0.2)}$ | 8.0 | 16.8 |
| DDOM | $93.0^{(3.6)}$ | $85.3^{(0.5)}$ | $63.5^{(0.4)}$ | $87.9^{(0.6)}$ | $44.7^{(2.2)}$ | $63.0^{(12.1)}$ | 13.0 | 9.4 |
| ExPT | $89.3^{(5.7)}$ | $84.2^{(2.4)}$ | $63.3^{(0.0)}$ | $93.0^{(0.8)}$ | $\mathbf{48.5^{(11.0)}}$ | $82.3^{(2.3)}$ | 12.0 | 13.3 |
| MINs | $89.0^{(3.4)}$ | $68.3^{(0.6)}$ | $63.9^{(0.9)}$ | $93.1^{(0.7)}$ | $45.8^{(2.1)}$ | $91.5^{(1.1)}$ | 11.2 | 11.8 |
| GTG | $92.1^{(0.0)}$ | $70.2^{(0.0)}$ | $63.3^{(0.0)}$ | $85.0^{(0.0)}$ | $52.5^{(0.0)}$ | $\mathbf{96.4^{(0.0)}}$ | 9.7 | 13.1 |
| Tri-Mentoring | $82.4^{(0.0)}$ | $66.6^{(0.0)}$ | $\mathbf{68.4^{(0.0)}}$ | $88.9^{(0.0)}$ | $50.9^{(1.1)}$ | $94.0^{(0.0)}$ | 10.2 | 11.7 |
| ICT | $93.3^{(3.4)}$ | $66.6^{(0.0)}$ | $\mathbf{68.4^{(0.0)}}$ | $88.9^{(0.0)}$ | $48.9^{(1.4)}$ | $\mathbf{95.5^{(1.1)}}$ | 9.0 | 13.4 |
| PGS | $79.6^{(7.5)}$ | $67.1^{(0.8)}$ | $\mathbf{68.4^{(0.0)}}$ | $88.9^{(0.0)}$ | $\mathbf{54.8^{(0.8)}}$ | $72.3^{(0.0)}$ | 11.3 | 8.3 |
| Match-Opt | $90.9^{(3.4)}$ | $68.4^{(0.8)}$ | $63.3^{(0.1)}$ | $87.8^{(0.6)}$ | $35.2^{(2.3)}$ | $72.2^{(0.1)}$ | 15.5 | 6.2 |
| RGD | $87.9^{(4.2)}$ | $68.7^{(0.6)}$ | $63.4^{(0.2)}$ | $90.2^{(0.3)}$ | $43.0^{(2.7)}$ | $88.5^{(1.1)}$ | 13.2 | 10.1 |
| COMs | $93.1^{(3.4)}$ | $67.0^{(0.9)}$ | $64.6^{(1.0)}$ | $97.1^{(1.6)}$ | $41.2^{(4.8)}$ | $91.8^{(0.9)}$ | 10.2 | 12.3 |
| RoMA | $96.5^{(0.0)}$ | $77.8^{(0.0)}$ | $63.3^{(0.0)}$ | $84.7^{(0.0)}$ | $49.8^{(1.4)}$ | $\mathbf{95.7^{(1.6)}}$ | 9.7 | 14.5 |
| ROMO | $\mathbf{98.1^{(0.7)}}$ | $66.8^{(1.0)}$ | $63.0^{(0.8)}$ | $91.8^{(0.9)}$ | $38.7^{(2.5)}$ | $87.8^{(0.9)}$ | 12.7 | 10.9 |
| GAMBO | $94.1^{(1.9)}$ | $86.3^{(0.2)}$ | $66.8^{(0.7)}$ | $\mathbf{121^{(0.0)}}$ | $\mathbf{50.8^{(3.3)}}$ | $86.7^{(1.1)}$ | **4.8** | **20.8** |
| **DynAMO-Grad.** | $90.3^{(4.7)}$ | $86.2^{(0.0)}$ | $64.4^{(2.5)}$ | $91.2^{(0.0)}$ | $44.2^{(7.8)}$ | $89.8^{(3.2)}$ | 9.5 | 14.2 |
| **DynAMO-BO-qUCB** | $95.1^{(1.9)}$ | $86.2^{(0.0)}$ | $66.7^{(1.5)}$ | $\mathbf{121^{(0.0)}}$ | $48.1^{(4.0)}$ | $86.9^{(4.5)}$ | 6.3 | 20.5 |
| **DynAMO-CMA-ES** | $89.8^{(3.6)}$ | $85.7^{(5.8)}$ | $63.9^{(0.9)}$ | $117^{(6.7)}$ | $50.6^{(4.8)}$ | $78.5^{(5.5)}$ | 9.3 | 17.5 |

| Median@128 | TFBind8 | UTR | ChEMBL | Molecule | Superconductor | D'Kitty | Rank ↓ | Opt. Gap ↑ |
|---|---|---|---|---|---|---|---|---|
| Dataset $\mathcal{D}$ | 33.7 | 42.8 | 50.9 | 87.6 | 6.7 | 77.8 | — | — |
| Grad. | $\mathbf{58.1^{(6.1)}}$ | $58.6^{(13.1)}$ | $\mathbf{59.3^{(8.6)}}$ | $\mathbf{85.3^{(7.7)}}$ | $\mathbf{36.0^{(6.7)}}$ | $65.1^{(14.4)}$ | 10.7 | 10.5 |
| BO-qUCB | $50.3^{(1.8)}$ | $62.1^{(3.4)}$ | $\mathbf{63.3^{(0.0)}}$ | $\mathbf{86.6^{(0.6)}}$ | $31.7^{(1.2)}$ | $74.4^{(0.6)}$ | 6.8 | 11.5 |
| CMA-ES | $50.7^{(2.7)}$ | $\mathbf{71.7^{(10.4)}}$ | $\mathbf{63.3^{(0.0)}}$ | $83.9^{(1.0)}$ | $\mathbf{37.9^{(0.7)}}$ | $59.3^{(10.9)}$ | 7.2 | 11.2 |
| BONET | $53.1^{(0.0)}$ | $46.5^{(0.6)}$ | $\mathbf{63.3^{(0.0)}}$ | $\mathbf{91.2^{(0.1)}}$ | $37.9^{(0.0)}$ | $92.1^{(0.0)}$ | **5.2** | 14.1 |
| DDOM | $55.9^{(0.7)}$ | $57.6^{(0.8)}$ | $\mathbf{63.3^{(0.0)}}$ | $83.4^{(0.1)}$ | $21.9^{(1.7)}$ | $57.6^{(13.2)}$ | 12.2 | 6.7 |
| ExPT | $44.7^{(6.8)}$ | $57.0^{(4.8)}$ | $\mathbf{63.3^{(0.0)}}$ | $89.8^{(3.3)}$ | $34.1^{(12.3)}$ | $67.8^{(14.1)}$ | 9.3 | 9.5 |
| MINs | $41.3^{(1.2)}$ | $58.0^{(0.5)}$ | $\mathbf{63.3^{(0.0)}}$ | $88.3^{(0.3)}$ | $32.3^{(1.6)}$ | $68.9^{(15.9)}$ | 9.7 | 8.8 |
| GTG | $43.4^{(0.3)}$ | $64.2^{(0.0)}$ | $\mathbf{63.3^{(0.0)}}$ | $83.1^{(0.0)}$ | $28.0^{(0.0)}$ | $90.6^{(0.0)}$ | 9.3 | 12.2 |
| Tri-Mentoring | $46.1^{(0.0)}$ | $61.1^{(0.0)}$ | $\mathbf{63.3^{(0.0)}}$ | $\mathbf{88.9^{(0.0)}}$ | $34.2^{(1.2)}$ | $88.4^{(0.0)}$ | 6.5 | 13.7 |
| ICT | $\mathbf{59.5^{(3.0)}}$ | $61.1^{(0.0)}$ | $57.1^{(0.0)}$ | $\mathbf{88.9^{(0.0)}}$ | $37.0^{(1.2)}$ | $88.4^{(0.1)}$ | 7.0 | **15.4** |
| PGS | $40.7^{(2.6)}$ | $60.3^{(0.7)}$ | $58.4^{(0.0)}$ | $\mathbf{88.9^{(0.0)}}$ | $28.2^{(0.5)}$ | $70.9^{(0.7)}$ | 12.3 | 8.0 |
| Match-Opt | $40.7^{(1.2)}$ | $57.9^{(0.5)}$ | $\mathbf{63.3^{(0.0)}}$ | $83.2^{(0.1)}$ | $14.5^{(0.9)}$ | $60.5^{(9.0)}$ | 15.0 | 3.4 |
| RGD | $41.1^{(1.3)}$ | $57.4^{(0.9)}$ | $\mathbf{63.3^{(0.0)}}$ | $86.4^{(0.1)}$ | $20.7^{(0.6)}$ | $70.9^{(1.8)}$ | 12.3 | 6.7 |
| COMs | $43.9^{(0.0)}$ | $59.0^{(0.5)}$ | $\mathbf{63.3^{(0.0)}}$ | $93.2^{(7.7)}$ | $21.3^{(5.6)}$ | $89.9^{(1.0)}$ | 8.5 | 11.8 |
| RoMA | $50.1^{(4.3)}$ | $\mathbf{77.4^{(0.0)}}$ | $\mathbf{63.3^{(0.0)}}$ | $84.7^{(0.0)}$ | $34.9^{(1.8)}$ | $63.7^{(6.2)}$ | 7.3 | 12.4 |
| ROMO | $\mathbf{58.7^{(3.3)}}$ | $37.7^{(0.3)}$ | $27.4^{(1.2)}$ | $61.8^{(2.6)}$ | $27.0^{(0.6)}$ | $46.0^{(11.7)}$ | 15.8 | -6.8 |
| GAMBO | $46.4^{(1.8)}$ | $63.4^{(3.3)}$ | $\mathbf{63.3^{(0.0)}}$ | $86.3^{(0.5)}$ | $28.9^{(1.1)}$ | $79.1^{(0.7)}$ | 7.8 | 11.3 |
| **DynAMO-Grad.** | $47.0^{(2.8)}$ | $69.8^{(6.0)}$ | $61.9^{(2.2)}$ | $85.9^{(0.4)}$ | $23.4^{(8.5)}$ | $68.7^{(12.1)}$ | 11.0 | 9.5 |
| **DynAMO-BO-qUCB** | $48.8^{(1.8)}$ | $65.9^{(3.7)}$ | $\mathbf{63.3^{(0.0)}}$ | $86.5^{(0.5)}$ | $22.7^{(2.0)}$ | $50.4^{(14.6)}$ | 9.7 | 6.3 |
| **DynAMO-CMA-ES** | $45.3^{(2.4)}$ | $65.8^{(8.9)}$ | $59.3^{(3.8)}$ | $\mathbf{99.0^{(12.1)}}$ | $22.5^{(5.1)}$ | $60.6^{(15.0)}$ | 11.0 | 8.8 |

*Table A7.* **Comparison of Design Diversity Against Model-Free Optimization Methods.** We evaluate DynAMO and other model-based optimization methods against model-free optimization methods. We report the pairwise diversity (resp., minimum novelty) oracle score achieved by the 128 evaluated designs in the top (resp., bottom) table. Metrics are reported mean[95% confidence interval] across 10 random seeds, where higher is better. All metrics are multiplied by 100 for easier legibility. **Bolded** entries indicate average scores with an overlapping 95% confidence interval to the best performing method. **Bolded** (resp., Underlined) Rank and Optimality Gap (Opt. Gap) metrics indicate the best (resp., second best) for a given backbone optimizer.

| Pairwise Diversity@128 | TFBind8 | UTR | ChEMBL | Molecule | Superconductor | D'Kitty | Rank ↓ | Opt. Gap ↑ |
|---|---|---|---|---|---|---|---|---|
| Dataset $\mathcal{D}$ | 65.9 | 57.3 | 60.0 | 36.7 | 66.0 | 85.7 | — | — |
| Grad. | $12.5^{(8.0)}$ | $7.8^{(8.8)}$ | $7.9^{(7.8)}$ | $24.1^{(13.3)}$ | $0.0^{(0.0)}$ | $0.0^{(0.0)}$ | 18.7 | -53.2 |
| BO-qUCB | $\mathbf{73.9^{(0.5)}}$ | $\mathbf{74.3^{(0.4)}}$ | $99.4^{(0.1)}$ | $93.6^{(0.5)}$ | $\mathbf{198^{(10.3)}}$ | $94.1^{(3.9)}$ | $\underline{3.8}$ | 43.5 |
| CMA-ES | $47.2^{(11.2)}$ | $44.6^{(15.9)}$ | $93.5^{(2.0)}$ | $66.2^{(9.4)}$ | $12.8^{(0.6)}$ | $164^{(10.6)}$ | 10.8 | 9.5 |
| BONET | $46.7^{(2.5)}$ | $24.6^{(0.4)}$ | $14.9^{(1.6)}$ | $5.5^{(0.2)}$ | $0.2^{(0.0)}$ | $0.1^{(0.0)}$ | 17.3 | -46.6 |
| DDOM | $51.3^{(1.0)}$ | $47.1^{(0.3)}$ | $21.9^{(3.6)}$ | $97.2^{(0.0)}$ | $1.9^{(0.1)}$ | $50.7^{(13.1)}$ | 12.5 | -16.9 |
| ExPT | $15.3^{(5.6)}$ | $16.5^{(1.6)}$ | $21.3^{(1.8)}$ | $5.3^{(0.7)}$ | $8.1^{(2.3)}$ | $0.2^{(0.0)}$ | 17.5 | -50.8 |
| MINs | $67.0^{(0.3)}$ | $56.8^{(0.4)}$ | $53.5^{(3.1)}$ | $34.1^{(2.6)}$ | $84.6^{(21.1)}$ | $4.3^{(0.3)}$ | 11.8 | -11.9 |
| GTG | $60.9^{(0.0)}$ | $44.6^{(0.0)}$ | $2.8^{(0.0)}$ | $0.9^{(0.0)}$ | $114.8^{(0.1)}$ | $2.7^{(0.0)}$ | 15.0 | -24.2 |
| Tri-Mentoring | $58.5^{(0.0)}$ | $57.6^{(0.0)}$ | $85.5^{(0.0)}$ | $39.9^{(0.0)}$ | $47.7^{(0.0)}$ | $62.5^{(0.0)}$ | 10.5 | -3.3 |
| ICT | $44.8^{(6.0)}$ | $57.5^{(0.0)}$ | $89.9^{(1.8)}$ | $70.3^{(8.6)}$ | $78.9^{(3.7)}$ | $164^{(0.8)}$ | 9.3 | 22.3 |
| PGS | $65.8^{(1.6)}$ | $57.4^{(0.3)}$ | $63.2^{(0.0)}$ | $39.9^{(0.0)}$ | $36.7^{(0.6)}$ | $162^{(0.7)}$ | 10.5 | 8.9 |
| Match-Opt | $65.1^{(0.4)}$ | $55.9^{(0.1)}$ | $\mathbf{99.8^{(0.0)}}$ | $97.2^{(0.0)}$ | $10.9^{(0.4)}$ | $202^{(0.5)}$ | 7.5 | 26.6 |
| RGD | $67.1^{(0.2)}$ | $58.4^{(0.2)}$ | $\mathbf{99.8^{(0.0)}}$ | $\mathbf{97.3^{(0.0)}}$ | $88.4^{(3.8)}$ | $76.2^{(0.7)}$ | 5.0 | 19.3 |
| COMs | $66.6^{(1.0)}$ | $57.4^{(0.2)}$ | $81.6^{(4.9)}$ | $3.8^{(0.9)}$ | $99.5^{(25.8)}$ | $21.1^{(23.5)}$ | 10.7 | -6.9 |
| RoMA | $21.3^{(0.3)}$ | $3.8^{(0.0)}$ | $5.9^{(0.2)}$ | $1.8^{(0.0)}$ | $49.4^{(6.1)}$ | $14.8^{(0.6)}$ | 17.2 | -45.8 |
| ROMO | $62.1^{(0.8)}$ | $57.1^{(0.1)}$ | $53.9^{(0.6)}$ | $48.7^{(0.1)}$ | $51.7^{(31.7)}$ | $22.1^{(5.5)}$ | 11.5 | -12.7 |
| GAMBO | $\mathbf{74.0^{(0.6)}}$ | $\mathbf{74.3^{(0.4)}}$ | $99.3^{(0.1)}$ | $93.3^{(0.4)}$ | $193^{(1.2)}$ | $17.7^{(3.5)}$ | 5.5 | 30.0 |
| **DynAMO-Grad.** | $66.9^{(6.9)}$ | $\mathbf{68.2^{(10.8)}}$ | $77.2^{(21.5)}$ | $93.0^{(1.2)}$ | $129^{(55.3)}$ | $104^{(56.1)}$ | 6.8 | 27.8 |
| **DynAMO-BO-qUCB** | $\mathbf{74.3^{(0.5)}}$ | $\mathbf{74.4^{(0.6)}}$ | $99.3^{(0.1)}$ | $93.5^{(0.6)}$ | $\mathbf{211^{(22.8)}}$ | $175^{(44.7)}$ | **2.8** | **59.4** |
| **DynAMO-CMA-ES** | $73.6^{(0.6)}$ | $\mathbf{73.1^{(3.1)}}$ | $72.0^{(3.1)}$ | $94.0^{(0.5)}$ | $97.8^{(13.2)}$ | $292^{(83.5)}$ | 5.2 | $\underline{55.2}$ |

| Minimum Novelty@128 | TFBind8 | UTR | ChEMBL | Molecule | Superconductor | D'Kitty | Rank ↓ | Opt. Gap ↑ |
|---|---|---|---|---|---|---|---|---|
| Dataset $\mathcal{D}$ | 0.0 | 0.0 | 0.0 | 0.0 | 0.0 | 0.0 | — | — |
| Grad. | $21.2^{(3.0)}$ | $\mathbf{51.7^{(2.9)}}$ | $\mathbf{97.4^{(3.9)}}$ | $79.5^{(19.7)}$ | $95^{(0.7)}$ | $102.2^{(6.1)}$ | 5.8 | 74.5 |
| BO-qUCB | $21.6^{(0.3)}$ | $\mathbf{51.7^{(0.2)}}$ | $97.9^{(0.4)}$ | $85.3^{(1.1)}$ | $93.8^{(0.6)}$ | $98.8^{(1.1)}$ | 6.0 | 74.8 |
| CMA-ES | $16.5^{(2.1)}$ | $47.8^{(1.0)}$ | $96.5^{(0.7)}$ | $73.0^{(18.0)}$ | $\mathbf{100^{(0.0)}}$ | $100^{(0.0)}$ | 7.8 | 72.3 |
| BONET | $\mathbf{94.6^{(1.3)}}$ | $38.8^{(0.1)}$ | $41.2^{(0.3)}$ | $10.3^{(0.1)}$ | $1.3^{(0.0)}$ | $1.1^{(0.0)}$ | 14.7 | 31.2 |
| DDOM | $11.1^{(0.4)}$ | $38.6^{(0.1)}$ | $96.5^{(0.9)}$ | $94.2^{(0.1)}$ | $98.0^{(0.1)}$ | $100^{(0.0)}$ | 9.3 | 73.1 |
| ExPT | $11.1^{(1.6)}$ | $39.0^{(0.7)}$ | $54.2^{(2.0)}$ | $15.6^{(1.2)}$ | $69.8^{(6.8)}$ | $3.5^{(0.9)}$ | 15.2 | 32.2 |
| MINs | $12.2^{(0.4)}$ | $38.3^{(0.2)}$ | $48.7^{(2.9)}$ | $22.1^{(1.7)}$ | $6.1^{(1.1)}$ | $0.6^{(0.1)}$ | 16.5 | 21.3 |
| GTG | $13.8^{(0.0)}$ | $37.8^{(0.0)}$ | $\mathbf{99.3^{(0.0)}}$ | $\mathbf{99.4^{(0.0)}}$ | $67.7^{(0.0)}$ | $0.2^{(0.0)}$ | 10.5 | 53.0 |
| Tri-Mentoring | $13.9^{(0.0)}$ | $31.8^{(0.0)}$ | $74.7^{(0.0)}$ | $75.5^{(0.0)}$ | $44.0^{(0.0)}$ | $64.3^{(0.0)}$ | 14.3 | 50.7 |
| ICT | $19.3^{(1.2)}$ | $31.7^{(0.1)}$ | $70.8^{(0.4)}$ | $75.6^{(0.0)}$ | $46.2^{(0.4)}$ | $66.2^{(0.7)}$ | 13.2 | 51.6 |
| PGS | $11.5^{(0.4)}$ | $33.1^{(1.8)}$ | $16.0^{(0.0)}$ | $15.8^{(0.0)}$ | $45.0^{(0.2)}$ | $74.8^{(1.1)}$ | 16.3 | 32.7 |
| Match-Opt | $12.1^{(0.5)}$ | $40.0^{(0.1)}$ | $98.5^{(0.1)}$ | $94.9^{(0.1)}$ | $91.9^{(0.1)}$ | $85.8^{(2.6)}$ | 8.7 | 70.5 |
| RGD | $12.3^{(0.5)}$ | $39.0^{(0.2)}$ | $98.6^{(0.1)}$ | $94.2^{(0.1)}$ | $90.4^{(0.3)}$ | $90.5^{(2.3)}$ | 8.5 | 70.9 |
| COMs | $10.9^{(0.3)}$ | $31.7^{(0.8)}$ | $52.4^{(11.0)}$ | $13.7^{(1.1)}$ | $99.6^{(0.3)}$ | $100^{(0.0)}$ | 14.0 | 51.4 |
| RoMA | $18.3^{(0.5)}$ | $40.1^{(0.2)}$ | $18.9^{(0.2)}$ | $95.3^{(0.0)}$ | $47.6^{(2.4)}$ | $5.1^{(0.2)}$ | 11.0 | 37.6 |
| ROMO | $16.1^{(0.5)}$ | $32.9^{(0.1)}$ | $5.0^{(0.7)}$ | $23.1^{(0.0)}$ | $78.5^{(0.5)}$ | $\mathbf{153.3^{(0.4)}}$ | 12.3 | 51.5 |
| GAMBO | $15.4^{(0.3)}$ | $\mathbf{51.8^{(0.2)}}$ | $97.8^{(0.3)}$ | $84.9^{(0.9)}$ | $85.1^{(0.4)}$ | $14.3^{(1.5)}$ | 8.8 | 58.2 |
| **DynAMO-Grad.** | $21.1^{(1.1)}$ | $\mathbf{52.2^{(1.3)}}$ | $\mathbf{98.6^{(1.5)}}$ | $85.8^{(1.0)}$ | $95.0^{(0.4)}$ | $107.2^{(6.7)}$ | **3.8** | **76.7** |
| **DynAMO-BO-qUCB** | $21.4^{(0.5)}$ | $\mathbf{51.7^{(0.2)}}$ | $97.1^{(0.5)}$ | $85.3^{(1.1)}$ | $94.7^{(0.2)}$ | $109^{(4.5)}$ | $\underline{5.3}$ | $\underline{76.6}$ |
| **DynAMO-CMA-ES** | $12.9^{(0.8)}$ | $48.0^{(1.6)}$ | $\mathbf{96.7^{(3.5)}}$ | $81.8^{(13.4)}$ | $94.5^{(0.7)}$ | $112^{(7.8)}$ | 7.8 | 74.3 |

*Table A8.* **Comparison of Design Diversity Against Model-Free Optimization Methods (cont.).** We evaluate DynAMO and other model-based optimization methods against model-free optimization methods. We report the $L_1$ coverage score achieved by the 128 evaluated designs. Metrics are reported mean$^{(95\% \text{ confidence interval})}$ across 10 random seeds, where higher is better. All metrics are multiplied by 100 for easier legibility. **Bolded** entries indicate average scores with an overlapping 95% confidence interval to the best performing method. **Bolded** (resp., Underlined) Rank and Optimality Gap (Opt. Gap) metrics indicate the best (resp., second best) for a given backbone optimizer.

| $L_1$ **Coverage@128** | **TFBind8** | **UTR** | **ChEMBL** | **Molecule** | **Superconductor** | **D'Kitty** | **Rank** $\downarrow$ | **Opt. Gap** $\uparrow$ |
|---|---|---|---|---|---|---|---|---|
| Dataset $\mathcal{D}$ | 0.42 | 0.31 | 1.42 | 0.68 | 6.26 | 0.58 | — | — |
| Grad. | $0.16^{(0.10)}$ | $0.20^{(0.13)}$ | $0.21^{(0.10)}$ | $0.42^{(0.18)}$ | $0.00^{(0.00)}$ | $0.00^{(0.00)}$ | 19.5 | -1.44 |
| BO-qUCB | $0.40^{(0.02)}$ | $\mathbf{0.54^{(0.01)}}$ | $\mathbf{2.40^{(0.05)}}$ | $\mathbf{2.52^{(0.07)}}$ | $7.79^{(0.04)}$ | $6.64^{(0.09)}$ | 4.3 | 1.77 |
| CMA-ES | $0.33^{(0.05)}$ | $0.48^{(0.04)}$ | $2.18^{(0.04)}$ | $1.82^{(0.12)}$ | $3.26^{(1.42)}$ | $3.78^{(1.36)}$ | 8.5 | 0.36 |
| BONET | $0.11^{(0.00)}$ | $0.22^{(0.00)}$ | $0.72^{(0.02)}$ | $0.54^{(0.00)}$ | $0.03^{(0.00)}$ | $0.07^{(0.00)}$ | 17.8 | -1.33 |
| DDOM | $0.43^{(0.01)}$ | $0.29^{(0.00)}$ | $0.68^{(0.05)}$ | $0.85^{(0.02)}$ | $\mathbf{9.24^{(0.23)}}$ | $0.66^{(0.11)}$ | 10.8 | 0.41 |
| ExPT | $0.22^{(0.05)}$ | $0.25^{(0.01)}$ | $0.60^{(0.03)}$ | $0.29^{(0.03)}$ | $0.45^{(0.01)}$ | $0.10^{(0.00)}$ | 18.0 | -1.29 |
| MINs | $0.43^{(0.02)}$ | $0.30^{(0.01)}$ | $1.32^{(0.03)}$ | $0.70^{(0.03)}$ | $0.86^{(0.13)}$ | $0.43^{(0.01)}$ | 11.0 | -0.94 |
| GTG | $0.42^{(0.00)}$ | $0.30^{(0.00)}$ | $1.61^{(0.00)}$ | $1.96^{(0.00)}$ | $8.77^{(0.00)}$ | $0.31^{(0.00)}$ | 8.7 | 0.62 |
| Tri-Mentoring | $\mathbf{0.48^{(0.00)}}$ | $0.30^{(0.00)}$ | $1.08^{(0.00)}$ | $0.53^{(0.00)}$ | $1.94^{(0.00)}$ | $3.79^{(0.00)}$ | 10.8 | -0.26 |
| ICT | $0.34^{(0.02)}$ | $0.30^{(0.00)}$ | $0.90^{(0.02)}$ | $0.80^{(0.05)}$ | $0.56^{(0.01)}$ | $3.73^{(0.02)}$ | 12.8 | -0.51 |
| PGS | $\mathbf{0.45^{(0.04)}}$ | $0.30^{(0.01)}$ | $1.40^{(0.00)}$ | $0.60^{(0.00)}$ | $1.92^{(0.00)}$ | $3.66^{(0.04)}$ | 10.0 | -0.22 |
| Match-Opt | $0.41^{(0.01)}$ | $0.32^{(0.00)}$ | $0.69^{(0.01)}$ | $0.86^{(0.02)}$ | $5.26^{(0.02)}$ | $5.22^{(0.07)}$ | 8.7 | 0.52 |
| RGD | $0.42^{(0.01)}$ | $0.33^{(0.00)}$ | $0.69^{(0.01)}$ | $0.86^{(0.02)}$ | $4.08^{(0.13)}$ | $4.90^{(0.07)}$ | 9.0 | 0.27 |
| COMs | $\mathbf{0.49^{(0.02)}}$ | $0.31^{(0.00)}$ | $1.11^{(0.16)}$ | $0.61^{(0.09)}$ | $0.37^{(0.11)}$ | $0.81^{(0.76)}$ | 11.0 | -1.00 |
| RoMA | $0.28^{(0.00)}$ | $0.46^{(0.01)}$ | $0.41^{(0.02)}$ | $0.42^{(0.01)}$ | $1.87^{(0.06)}$ | $0.79^{(0.01)}$ | 14.8 | -0.91 |
| ROMO | $0.33^{(0.02)}$ | $0.30^{(0.00)}$ | $1.31^{(0.02)}$ | $0.62^{(0.02)}$ | $0.34^{(0.16)}$ | $6.13^{(2.81)}$ | 11.8 | -0.11 |
| GAMBO | $0.40^{(0.03)}$ | $\mathbf{0.55^{(0.01)}}$ | $\mathbf{2.38^{(0.10)}}$ | $\mathbf{2.53^{(0.05)}}$ | $7.45^{(0.01)}$ | $1.29^{(0.08)}$ | 6.3 | 0.82 |
| **DynAMO-Grad.** | $0.36^{(0.04)}$ | $\mathbf{0.53^{(0.06)}}$ | $1.46^{(0.38)}$ | $\mathbf{2.50^{(0.06)}}$ | $6.47^{(1.24)}$ | $5.85^{(1.35)}$ | 6.7 | 1.25 |
| **DynAMO-BO-qUCB** | $0.40^{(0.03)}$ | $\mathbf{0.55^{(0.01)}}$ | $\mathbf{2.47^{(0.07)}}$ | $\mathbf{2.54^{(0.04)}}$ | $7.88^{(0.03)}$ | $\mathbf{7.80^{(0.23)}}$ | **2.8** | **2.00** |
| **DynAMO-CMA-ES** | $0.40^{(0.03)}$ | $\mathbf{0.56^{(0.01)}}$ | $\mathbf{1.82^{(0.72)}}$ | $\mathbf{2.54^{(0.05)}}$ | $4.75^{(2.16)}$ | $3.29^{(1.56)}$ | 6.3 | 0.62 |

*Table A9.* **Computational Requirements of DynAMO.** To evaluate the computational cost of augmenting an MBO problem with DynAMO, we compare both the total runtime and maximum GPU utilization of vanilla optimizers with that of their DynAMO equivalents on six MBO problems. Runtime is reported mean$^{(95\% \text{ confidence interval})}$ in seconds across 10 random seeds. Maximum GPU utilization is reported for a single experimental run. The average metric across all six tasks is reported in the final column.

| Runtime (seconds) | TFBind8 | UTR | ChEMBL | Molecule | Superconductor | D'Kitty | Average $\downarrow$ |
|---|---|---|---|---|---|---|---|
| Grad. | $58.1^{(24.6)}$ | $240.0^{(11.1)}$ | $104.7^{(8.3)}$ | $458.7^{(28.2)}$ | $24.7^{(3.7)}$ | $10357^{(2044)}$ | 1874 |
| DynAMO-Grad. | $517.0^{(68.7)}$ | $2686^{(868.3)}$ | $297.0^{(100.0)}$ | $1091^{(243.1)}$ | $917.5^{(554.4)}$ | $13302^{(4301)}$ | 3135 |
| Adam | $26.1^{(0.5)}$ | $249.2^{(17.5)}$ | $75.3^{(13.2)}$ | $748.2^{(34.6)}$ | $24.4^{(2.6)}$ | $8797^{(1333)}$ | 1653 |
| DynAMO-Adam | $646.7^{(185.6)}$ | $616.6^{(106.1)}$ | $709.6^{(410.3)}$ | $953.7^{(140.9)}$ | $6941^{(205)}$ | $24444^{(2466)}$ | 5719 |
| BO-qEI | $182.4^{(51.6)}$ | $339.6^{(61.7)}$ | $494.3^{(34.2)}$ | $1548^{(56.4)}$ | $567.8^{(119.3)}$ | $3196^{(158.4)}$ | 838.0 |
| DynAMO-BO-qEI | $449.9^{(111.0)}$ | $952.8^{(140.6)}$ | $494.4^{(161.8)}$ | $1307^{(232.1)}$ | $1122^{(150.0)}$ | $5973^{(1245)}$ | 1717 |
| BO-qUCB | $186.9^{(36.4)}$ | $414.4^{(73.7)}$ | $289.8^{(80.0)}$ | $589.8^{(50.2)}$ | $524.9^{(232.5)}$ | $8424^{(1336)}$ | 1738 |
| DynAMO-BO-qUCB | $408.4^{(66.3)}$ | $1531^{(146.6)}$ | $932.3^{(323.1)}$ | $1525^{(357.1)}$ | $2100^{(1547)}$ | $11648^{(5091)}$ | 3024 |
| CMA-ES | $109.4^{(66.0)}$ | $130.0^{(3.2)}$ | $66.0^{(1.4)}$ | $604.1^{(12.4)}$ | $20.0^{(0.3)}$ | $14027^{(2651)}$ | 2493 |
| DynAMO-CMA-ES | $10744^{(7904)}$ | $9551^{(2717)}$ | $751.3^{(381.0)}$ | $8088^{(1533)}$ | $4796^{(445)}$ | $13727^{(5870)}$ | 7943 |
| CoSyNE | $133.9^{(81.3)}$ | $187.0^{(1.7)}$ | $110.3^{(2.6)}$ | $716.8^{(30.0)}$ | $25.8^{(0.5)}$ | $9876.8^{(735.3)}$ | 1842 |
| DynAMO-CoSyNE | $8823^{(3370)}$ | $11388^{(2375)}$ | $1374^{(584.6)}$ | $8002^{(1802)}$ | $6261^{(3202)}$ | $17147^{(3856)}$ | 8832 |

| Max GPU Utilization (MB) | TFBind8 | UTR | ChEMBL | Molecule | Superconductor | D'Kitty | Average $\downarrow$ |
|---|---|---|---|---|---|---|---|
| Grad. | 307.9 | 786.8 | 503.9 | 1920 | 133.6 | 131.7 | 630.7 |
| DynAMO-Grad. | 374.1 | 994.8 | 503.9 | 1920 | 208.3 | 160.1 | 693.6 |
| Adam | 307.9 | 786.8 | 503.9 | 1920 | 133.6 | 131.7 | 630.7 |
| DynAMO-Adam | 374.4 | 995.0 | 503.9 | 1920 | 208.4 | 160.1 | 693.7 |
| BO-qEI | 307.9 | 786.8 | 519.1 | 1920 | 372.6 | 544.1 | 741.8 |
| DynAMO-BO-qEI | 371.4 | 789.0 | 503.9 | 1920 | 479.4 | 445.2 | 751.6 |
| BO-qUCB | 597.3 | 786.8 | 642.3 | 1920 | 712.4 | 879.5 | 923.1 |
| DynAMO-BO-qUCB | 516.2 | 786.8 | 646.9 | 1920 | 718.0 | 406.1 | 832.4 |
| CMA-ES | 307.9 | 786.8 | 503.9 | 1920 | 133.6 | 131.7 | 630.7 |
| DynAMO-CMA-ES | 307.9 | 786.8 | 503.9 | 1920 | 133.6 | 131.7 | 630.7 |
| CoSyNE | 307.9 | 786.8 | 503.9 | 1920 | 133.6 | 131.7 | 630.7 |
| DynAMO-CoSyNE | 307.9 | 786.8 | 503.9 | 1920 | 133.6 | 131.7 | 630.7 |

*Table A10.* **Pairwise Diversity as a Predictor for Downstream Secondary Exploration.** We evaluate the pairwise diversity achieved by 128 proposed designs (**PD@128**); and also the variance of the distribution of oracle secondary objective values of those same 128 proposed designs. Note that the secondary objectives are *not* explicitly optimized against in the offline MBO setting. Metrics are reported mean[95% confidence interval] across 10 random seeds, where higher is better (i.e., more diverse designs and better capture of the range of secondary objective values). All metrics are multiplied by 100 for easier legibility.

| | Vehicle Safety | | | Welded Beam | | UTR | |
| --- | --- | --- | --- | --- | --- | --- | --- |
| Method | PD@128 | Acceleration | Intrusion | PD@128 | Deflection | PD@128 | GC Content |
| Grad. | $0.0^{(0.0)}$ | $0.1^{(0.0)}$ | $0.0^{(0.0)}$ | $0.0^{(0.0)}$ | $0.1^{(0.3)}$ | $7.8^{(8.8)}$ | $0.7^{(0.7)}$ |
| DynAMO-Grad. | $2.5^{(0.2)}$ | $12.6^{(0.5)}$ | $10.7^{(0.9)}$ | $19.6^{(33.3)}$ | $31.7^{(9.3)}$ | $68.2^{(10.8)}$ | $3.4^{(0.7)}$ |
| Adam | $0.0^{(0.0)}$ | $0.1^{(0.1)}$ | $0.1^{(0.1)}$ | $0.0^{(0.0)}$ | $1.5^{(1.6)}$ | $11.0^{(12.1)}$ | $4.0^{(5.9)}$ |
| DynAMO-Adam | $2.2^{(0.1)}$ | $12.6^{(0.6)}$ | $10.3^{(1.0)}$ | $11.1^{(3.5)}$ | $49.5^{(21.3)}$ | $72.3^{(3.4)}$ | $14.0^{(3.1)}$ |
| CMA-ES | $0.0^{(0.0)}$ | $0.5^{(0.2)}$ | $0.4^{(0.2)}$ | $0.1^{(0.1)}$ | $0.0^{(0.0)}$ | $44.6^{(15.9)}$ | $36.5^{(8.3)}$ |
| DynAMO-CMA-ES | $8.6^{(6.0)}$ | $28.5^{(10.8)}$ | $30.8^{(20.0)}$ | $43.9^{(6.0)}$ | $19.8^{(18.8)}$ | $73.1^{(3.1)}$ | $42.0^{(2.3)}$ |
| CoSyNE | $0.0^{(0.0)}$ | $0.6^{(0.1)}$ | $0.5^{(0.1)}$ | $0.1^{(0.0)}$ | $16.8^{(6.5)}$ | $12.7^{(9.8)}$ | $1.6^{(2.5)}$ |
| DynAMO-CoSyNE | $1.9^{(2.7)}$ | $4.0^{(4.2)}$ | $2.2^{(2.6)}$ | $55.1^{(65.3)}$ | $65.8^{(13.2)}$ | $20.3^{(2.3)}$ | $1.0^{(1.3)}$ |
| BO-qEI | $1.3^{(0.1)}$ | $7.5^{(0.4)}$ | $7.2^{(0.4)}$ | $46.3^{(2.0)}$ | $8.0^{(0.3)}$ | $73.8^{(0.5)}$ | $44.8^{(0.8)}$ |
| DynAMO-BO-qEI | $2.4^{(0.1)}$ | $13.3^{(0.2)}$ | $10.5^{(0.2)}$ | $78.1^{(18.0)}$ | $26.6^{(2.1)}$ | $74.6^{(0.3)}$ | $45.7^{(0.6)}$ |
| BO-qUCB | $1.2^{(0.1)}$ | $5.7^{(0.4)}$ | $6.1^{(0.4)}$ | $46.5^{(7.5)}$ | $7.8^{(0.2)}$ | $74.3^{(0.2)}$ | $45.3^{(0.4)}$ |
| DynAMO-BO-qUCB | $2.8^{(0.1)}$ | $12.3^{(0.2)}$ | $10.9^{(0.1)}$ | $63.9^{(4.2)}$ | $29.0^{(1.9)}$ | $74.4^{(0.6)}$ | $45.2^{(0.5)}$ |

# E. Ablation Experiments

## E.1. Sampling Batch Size Ablation

Recall from (7) that a key component of our DynAMO algorithm is the estimation of the empirical KL-divergence between the $\tau$-weighted probability distribution of real designs from the offline dataset and the distribution of sampled designs from the generative policy. The latter distribution of generated designs is fundamentally dependent on our sampling batch size $b$ in **Algorithm 1**—the larger the batch size per sampling step, the better our empirical estimate of the KL divergence between our two distributions. However, as the batch size increases, there also exists a greater likelihood of significant regret in the sampling policy when compared to the optimal sequential policy (Gonzalez et al., 2016; Wilson et al., 2017). To better evaluate the impact of the sampling batch size parameter $b$ DynAMO, we experimentally evaluate sampling batch size values logarithmically ranging between $2 \leq b \leq 512$. We use a BO-qEI sampling policy with the DynAMO-modified objective on the TFBind8 optimization task, and evaluate both the Best@128 oracle score and Pairwise Diversity of the 128 final proposed design candidates across 10 random seeds.

Our ablation experiment results are shown in **Figure A3**. We find that the Best@128 design quality scores do not vary significantly as a function of the batch sizes that were evaluated; however, there exists an optimal batch size ($b = 64$ in our experiments) that maximizes the diversity of designs according to the pairwise diversity metric.

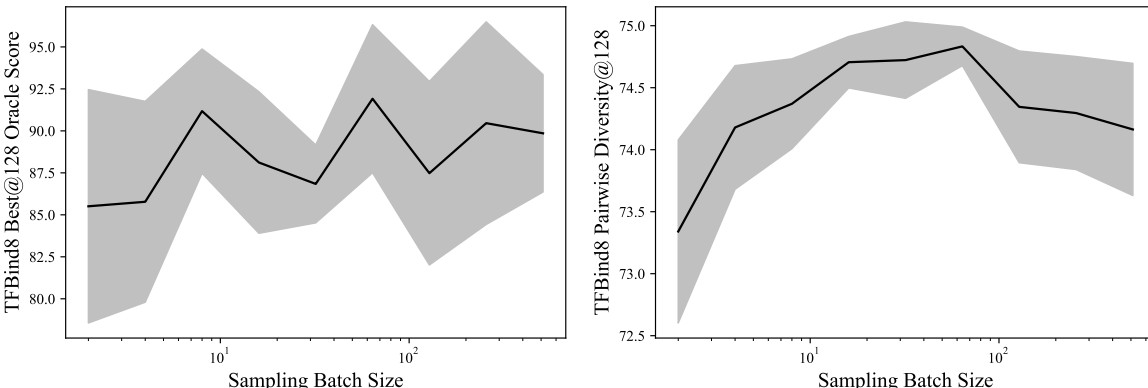

*Figure A3.* **Sampling Batch Size Ablation.** We vary the sampling batch size $b$ in **Algorithm 1** between 2 and 512, and report both the (**left**) Best@128 Oracle Score and (**right**) Pairwise Diversity score for 128 final designs proposed by a DynAMO-BO-qEI policy on the TFBind8 optimization task. We plot the mean $\pm$ 95% confidence interval over 10 random seeds.

## E.2. Adversarial Critic Feedback and Distribution Matching Ablation

Recall that instead of solving the original MBO optimization problem in (3), DynAMO leverages weak Lagrangian duality to solve the constrained optimization problem in (7)—copied below for convenience:

$$\max_{\pi \in \Pi} \quad J(\pi) = \mathbb{E}_{q^\pi}[r_\theta(x)] - \frac{\beta}{\tau} D_{\text{KL}}(q^\pi || p_{\mathcal{D}}^\tau)$$
$$\text{s.t.} \quad \mathbb{E}_{x \sim p_{\mathcal{D}}^\tau(x)} c^*(x) - \mathbb{E}_{x \sim q^\pi(x)} c^*(x) \leq W_0 \tag{85}$$

We can think of this problem formulation as as the fusion of two separable components: (1) **A**dversarial feedback via a source-critic model $c^*(x)$ to prevent out-of-distribution evaluation of $r_\theta(x)$; and (2) **D**iversit**y** (via KL-divergence-based distribution matching with a diverse reference distribution $p_{\mathcal{D}}^\tau$) **in** **M**odel-based **O**ptimization. These two components together form the foundation of **DynAMO** presented in **Algorithm 1**. To better understand how each of these two components affects the performance of DynAMO-augmented optimizers, we can separate these two components and study them individually.

**A**MO is our ablation method that solves the related optimization problem

$$\max_{\pi \in \Pi} \quad \mathbb{E}_{q^\pi}[r_\theta(x)]$$
$$\text{s.t.} \quad \mathbb{E}_{x \sim p_{\mathcal{D}}^\tau(x)} c^*(x) - \mathbb{E}_{x \sim q^\pi(x)} c^*(x) \leq W_0 \tag{86}$$

instead of (85). Note that AMO solves the same constrained optimization problem as DynAMO in the setting where $\beta = 0$. We note that our derivation of the Lagrange dual function of (7) in **Appendix A** is invalid when $\beta = 0$, and so we cannot

exactly solve (86) using the same methodology presented in **Algorithm 1**. Instead, we leverage the **adaptive Source Critic Regularization (aSCR)** algorithm from Yao et al. (2024) to *approximate* a solution to (86) in the Lagrangian dual space—we use their publicly available implementation of aSCR at `github.com/michael-s-yao/gabo` and defer to their work for additional discussion regarding the specific implementation details of aSCR.

**Dyn**MO is our separate ablation method that solves the related (unconstrained) optimization problem

$$\max_{\pi \in \Pi} \quad J(\pi) = \mathbb{E}_{q^\pi}[r_\theta(x)] - \frac{\beta}{\tau} D_{\mathrm{KL}}(q^\pi || p_\mathcal{D}^\tau) \tag{87}$$

instead of (85). To implement DynMO empirically, we modify **Algorithm 1** by ignoring the subroutine to solve for the globally optimal Lagrange multiplier $\lambda$ using (13), and instead fixing $\lambda = 0$ for the entire optimization process to effectively remove any contributions from the adversarial source critic $c^*(x)$. All other implementation details were kept constant.

We compare DynAMO with AMO and **Dyn**MO in **Supplementary Tables A11-A13**. Firstly, we note that DynAMO and AMO are competitive in proposing the high-quality designs according to the Best@128 oracle scores, alternating between having the highest and second best Rank and Optimality Gaps across all six tasks when compared with DynMO and the baseline optimizer for all optimizers evaluated. This makes sense, as the purpose of the adversarial source critic-dependent constraint in (85) and (86) is to minimize out-of-distribution evaluation of $r_\theta(x)$ during optimization—as a result, the forward surrogate model $r_\theta(x)$ can provide a better estimate of the quality of sampled designs, leading to higher quality designs according to the true oracle function $r(x)$. Separately, we find that DynMO and DynAMO also perform similarly in terms of the all 3 diversity metrics evaluated. However, we find that DynMO (resp., AMO) struggles on proposing high-quality (resp., diverse) sets of final designs. **These experimental results collectively allow us to conclude that both the adversarial source critic supervision and KL-divergence-based distribution matching are important for DynAMO to propose *both* high-quality and diverse sets of designs.**

### E.3. $\beta$ and $\tau$ Hyperparameter Ablation

Fundamentally, DynAMO relies on two important hyperparameters that define the constrained optimization problem in (7): (1) the $\beta$ hyperparameter dictates the relative weighting of the KL-divergence penalty relative to the original MBO objective; and (2) the $\tau$ temperature hyperparameter describes the distribution of reference designs weighted according to their oracle scores in the offline dataset. To better interrogate how these hyperparameters impact the performance of DynAMO-augmented MBO optimizers, we (independently) ablate the values of both $\beta$ and $\tau$ logarithmically between $0.01 \le \beta, \tau \le 100$. Similar to our experiments in **Appendix E.1**, we use a BO-qEI sampling policy with the DynAMO-modified objective on the TFBind8 optimization task, and evaluate both the Best@128 oracle score and Pairwise Diversity of the 128 final proposed design candidates across 10 random seeds.

Our experimental results for our $\beta$ ablation study are shown in **Supplementary Figure A4**. Firstly, as the strength of the KL-divergence term $\beta$ increases, the diversity of proposed designs (according to the Pairwise Diversity metric) increases roughly proportional to the logarithm of $\beta$ (**Supplementary Fig. A4**). This is expected: as the distribution matching objective becomes more important relative to the $r_\theta(x)$ forward surrogate model, the generative policy is rewarded for finding an increasingly diverse set of designs that matches the $\tau$-weighted reference distribution. Similarly, we found that for sufficiently large values of $\beta$ (i.e., $\beta \ge 0.03$ in our particular experimental setting), the *quality* of designs (according to the Best@128 oracle score) decreases due to the inherent trade-off between design quality (according to $r_\theta(x)$) and diversity (according to the KL-divergence in (7)). Interestingly, for small values of $\beta$ (i.e., $\beta \le 0.03$) the quality of designs actually *increases* with $\beta$. This is because in this regime, naïvely optimizing against primarily $r_\theta(x)$ leads to the policy exploiting suboptimal regions of the design space—penalizing the optimization objective with a 'small amount of' the diversity objective helps the policy explore new regions of the design space that can contain more optimal designs according to the hidden oracle objective $r(x)$.

Separately, the experimental results for our $\tau$ ablation study are shown in **Supplementary Figure A5**. (Note that in these experiments, we fix $\beta = \tau$ so that the ratio $\beta/\tau$ in **Algorithm 1** remains constant.) As the value of $\tau$ increases, the diversity of designs captured by the reference $\tau$-weighted probability distribution decreases and approaches a (potential mixture of) Dirac delta functions with non-zero support at the optimal designs in the offline dataset. As a result, distribution matching via the KL-divergence objective no longer encourages the generative policy to find a diverse sample of designs, as the reference distribution is no longer diverse itself for $\tau \gg 1$. Similar to our $\beta$ ablation study, we find that there is a unique exploration-exploitation trade-off phenomenon according to the Best@128 oracle score as a function of $\tau$: in our particular experimental setting, we find that for $\tau \le 1$, the Best@128 oracle score (modestly) increases, while for $\tau \ge 1$, the score

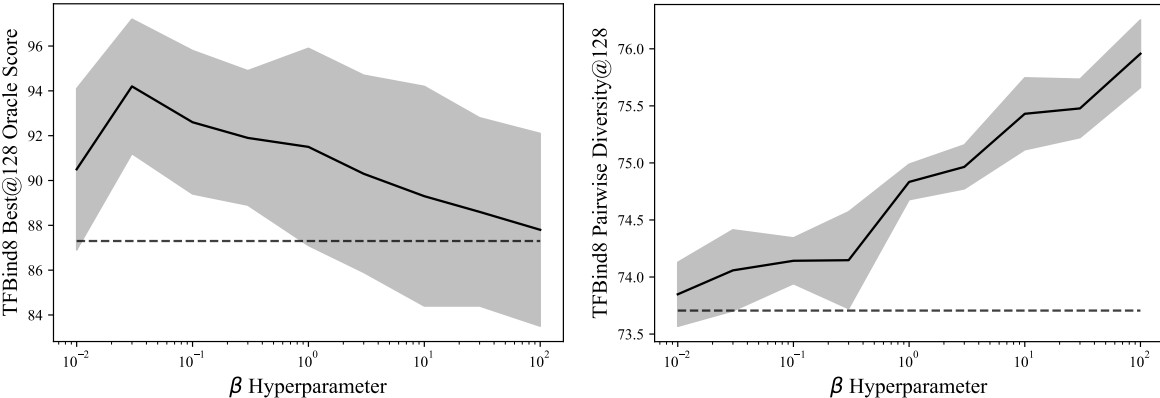

*Figure A4. β* **Hyperparameter Ablation.** We vary the value of the KL-divergence regularization strength hyperparameter $\beta$ in **Algorithm 1** between 0.01 and 100, and report both the (**left**) Best@128 Oracle Score and (**right**) Pairwise Diversity score for 128 final design candidates proposed by a DynAMO-BO-qEI policy on the TFBind8 optimization task. We plot the mean $\pm$ 95% confidence interval over 10 random seeds in both plots. The dotted horizontal line corresponds to the $\beta = 0$ experimental mean score, which could not be plotted as a point on the logarithmic $x$-axis.

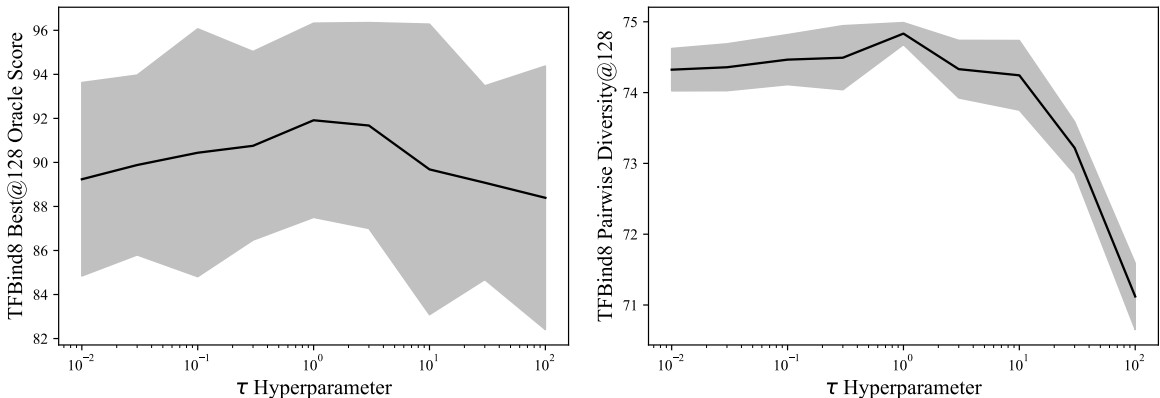

*Figure A5. τ* **Temperature Hyperparameter Ablation.** We vary the value of the temperature hyperparameter $\tau$ in **Algorithm 1** between 0.01 and 100, and report both the (**left**) Best@128 Oracle Score and (**right**) Pairwise Diversity score for 128 final designs proposed by a DynAMO-BO-qEI policy on the TFBind8 optimization task. We plot the mean $\pm$ 95% confidence interval over 10 random seeds.

decreases. For $\tau \approx 1$, we find that the generative policy is encouraged to match a sample of high-quality samples that is simultaneously *diverse* enough for the generative policy to explore new regions of the design space.

### E.4. Oracle Evaluation Budget Ablation

Recall that in our experiments, we evaluate DynAMO and baseline methods using an oracle evaluation budget of $k = 128$ samples consistent with prior work (Mashkaria et al., 2023; Yu et al., 2021; Trabucco et al., 2021; Chen et al., 2023c; Yao et al., 2024). More specifically, this means that any offline optimization method proposes exactly $k$ design candidates that are evaluated by the hidden oracle function $r(x)$ as the final step for experimental evaluation. In **Table 1**, we reported both the Best@$k$ and Pairwise Diversity@$k$ metrics, where Best@$k$ represents the maximum oracle score achieved by the $k$ final design candidates; and Pairwise Diversity@$k$ represents the pairwise diversity averaged over the $k$ candidates.

However, in different experimental settings we might have a different evaluation budget available—larger values of $k$ are more costly but enable us to evaluate more designs that are potentially promising, whereas smaller, more practical budgets may preclude the evaluation of optimal designs according to $r(x)$. In this section, we evaluate the performance of DynAMO as a function of the allowed evaluation budget $16 \leq k \leq 1024$. We compare DynAMO-augmented optimizers against the corresponding vanilla backbone optimization method on the TFBind8 task, and plot the mean and 95% confidence interval Best@$k$ and Pairwise Diversity@$k$ metrics as a function of $k$ in **Supplementary Figure A6**.

As expected, the Best@$k$ oracle score is monotonically non-decreasing as a function of $k$ for all DynAMO-enhanced and baseline optimizers (**Supplementary Fig. A6**). We also find that in the limit of $k \gg 1$, the DynAMO optimizers are able to propose best designs that are more optimal than the designs by their baseline counterparts for first-order, evolutionary, and Bayesian optimization algorithms. Furthermore, DynAMO achieves a mean Best@$k$ score non-inferior to that of the baseline method for all $k \geq 128$ across all the optimization methods evaluated on the TFBind8 task.

Separately, we find that the Pairwise Diversity of the $k$ designs proposed by DynAMO-augmented first-order optimizers (i.e., **DynAMO-Grad.** and **DynAMO-Adam**) increases as a function of $k$. This makes sense, as first-order methods generally produce optimization trajectories that are simple curves in the design space as a function of the acquisition step. As a result, increasing $k$ can be informally thought of as increasing the fraction of the trajectory curve connecting the initial and final samples during the acquisition process. In contrast, we find that the Pairwise Diversity *decreases* after a certain optimizer-dependent threshold $k$ for evolutionary and Bayesian optimization-based backbone optimizers. This is because as both classes of optimization methods do not necessarily sample repeatedly from any given region of the input space; as a result, the pairwise diversity between any two sampled points may decrease as more of the design space has been explored as a function of $k$. Finally, we found that leveraging DynAMO improves the Pairwise Diversity of designs compared to the baseline objective for almost all optimizers and values of $k$ assessed, as expected.

Altogether, these results suggest that DynAMO helps optimization methods discover both high-quality and diverse sets of designs across a wide range of oracle evaluation budgets.

### E.5. Optimization Initialization Ablation

In **Algorithm 1**, we initialize DynAMO by sampling the initial batch of $b = 64$ designs according to a pseudo-random Sobol sequence as described in **Appendix B**. This initial batch of designs is used as the 'starting point' in our first-order optimization experiments. However, most first-order offline MBO algorithms reported in prior work (Trabucco et al., 2021; Yu et al., 2021) do not follow this same initialization schema. Instead, they perform a *top-$k$* initialization strategy where the top $k = b$ designs in the dataset with the highest associated reward score constitute the initial batch of designs. First-order optimization is then performed on these initial top-$k$ designs. However, it is possible that for many MBO problems, these top-$k$ initial designs constitute only a small 'area' of the overall search space, resulting in a lower diversity of final designs when compared to Sobol sequence initialization.

To interrogate whether the gains in diversity of designs obtained with DynAMO are due to our Sobol sequence-based initialization strategy, we evaluated baseline Gradient Ascent, COMs, RoMA, ROMO, GAMBO with Gradient Ascent, and DynAMO with Gradient Ascent using both Sobol sequence-based and top-$k$-based initialization strategies. All algorithms were initialized using $k = b = 64$ samples and used Gradient Ascent as the backbone optimizer (except for RoMA from Yu et al. (2021), which used Adam Ascent in line with the original method proposed by the authors).

Our results are shown in **Supplementary Table A14**. Empirically, we found that the relative performance of Sobol sequence-initialized and Top-$k$-initialized optimizer largely depends on the specific algorithm; for example, COMs and RoMA strongly benefit from using Top-$k$ initialization in obtaining high-quality designs. This makes sense, as the original authors for both methods use Top-$k$ initialization for all their experiments. In contrast, the quality of designs proposed by GAMBO and DynAMO is better with Sobol sequence initialization.

While DynAMO using Sobol sequence initialization does indeed outperform the Top-$k$-initialized counterpart across all tasks, both initialization strategies consistently propose batches of designs with competitive pairwise diversity scores when compared to other first-order optimization algorithms. This suggests that DynAMO is able to provide a significant advantage in proposing diverse designs that extend beyond the choice of initialization strategy alone. Separately for the other first-order optimization methods assessed, there is no clear advantage in obtaining diverse designs when using Sobol sequence initialization according to the pairwise diversity metric across all tasks. In summary, these results suggest that DynAMO is able to propose both high-quality and diverse sets of designs with performance exceeding what is possible with a switching to a Sobol sequence initialization alone.

*Table A11.* **Quality of Design Candidates in Ablation of A**dversarial Critic Feedback (**A**MO) and **D**iversity in (**Dyn**MO) **M**odel-**based O**ptimization. We evaluate our method (1) with the KL-divergence penalized-MBO objective as in (11) only (**Dyn**MO); (2) with the adversarial source critic-dependent constraint as introduced by Yao et al. (2024) only (**A**MO); and (3) with both algorithmic components as in **Dyn**AMO described in **Algorithm 1**. We report the Best@128 (resp., Median@128) oracle score achieved by the 128 evaluated designs in the top (resp., bottom) table. Metrics are reported mean$^{(95\% \text{ confidence interval})}$ across 10 random seeds, where higher is better. **Bolded** entries indicate average scores with an overlapping 95% confidence interval to the best performing method. **Bolded** (resp., Underlined) Rank and Optimality Gap (Opt. Gap) metrics indicate the best (resp., second best) for a given backbone optimizer.

| Best@128 | TFBind8 | UTR | ChEMBL | Molecule | Superconductor | D'Kitty | Rank ↓ | Opt. Gap ↑ |
|---|---|---|---|---|---|---|---|---|
| Dataset $\mathcal{D}$ | 43.9 | 59.4 | 60.5 | 88.9 | 40.0 | 88.4 | — | — |
| Grad. | $90.0^{(4.3)}$ | $80.9^{(12.1)}$ | $60.2^{(8.9)}$ | $88.8^{(4.0)}$ | $36.0^{(6.8)}$ | $65.6^{(14.5)}$ | 3.2 | 6.8 |
| AMO-Grad. | $73.1^{(12.8)}$ | $77.1^{(9.6)}$ | $64.4^{(1.5)}$ | $92.8^{(8.0)}$ | $46.0^{(6.8)}$ | $90.6^{(14.5)}$ | 1.8 | 10.5 |
| DynMO-Grad. | $61.3^{(9.7)}$ | $63.6^{(11.6)}$ | $59.8^{(8.6)}$ | $89.3^{(5.6)}$ | $36.3^{(6.9)}$ | $70.4^{(12.0)}$ | 3.5 | 0.0 |
| DynAMO-Grad. | $90.3^{(4.7)}$ | $86.2^{(0.0)}$ | $64.4^{(2.5)}$ | $91.2^{(0.0)}$ | $44.2^{(7.8)}$ | $89.8^{(3.2)}$ | 1.5 | 14.2 |
| Adam | $62.9^{(13.0)}$ | $69.7^{(10.5)}$ | $62.9^{(1.9)}$ | $92.3^{(8.9)}$ | $37.8^{(6.3)}$ | $58.4^{(18.5)}$ | 2.8 | 0.5 |
| AMO-Adam | $94.0^{(2.2)}$ | $60.0^{(12.6)}$ | $60.9^{(8.7)}$ | $91.4^{(6.3)}$ | $37.8^{(6.3)}$ | $88.4^{(13.8)}$ | 2.8 | 8.6 |
| DynMO-Adam | $66.6^{(12.9)}$ | $68.7^{(10.1)}$ | $63.7^{(0.4)}$ | $92.0^{(8.3)}$ | $38.6^{(5.7)}$ | $66.5^{(14.6)}$ | 2.5 | 2.5 |
| DynAMO-Adam | $95.2^{(1.7)}$ | $86.2^{(0.0)}$ | $65.2^{(1.1)}$ | $91.2^{(0.0)}$ | $45.5^{(5.7)}$ | $84.9^{(12.0)}$ | 1.7 | 14.5 |
| BO-qEI | $87.3^{(5.8)}$ | $86.2^{(0.0)}$ | $65.4^{(1.0)}$ | $117^{(3.1)}$ | $53.1^{(3.3)}$ | $84.4^{(0.9)}$ | 3.5 | 18.7 |
| AMO-BO-qEI | $94.1^{(1.9)}$ | $86.3^{(0.2)}$ | $66.8^{(0.7)}$ | $121^{(0.0)}$ | $50.8^{(3.3)}$ | $86.7^{(1.1)}$ | 1.5 | 20.8 |
| DynMO-BO-qEI | $93.2^{(3.3)}$ | $86.2^{(0.0)}$ | $66.0^{(0.8)}$ | $121^{(0.0)}$ | $49.6^{(2.6)}$ | $85.9^{(1.0)}$ | 2.7 | 20.2 |
| DynAMO-BO-qEI | $91.9^{(4.4)}$ | $86.2^{(0.0)}$ | $67.0^{(1.3)}$ | $121^{(0.0)}$ | $53.5^{(5.0)}$ | $85.5^{(1.1)}$ | 1.8 | 20.7 |
| BO-qUCB | $88.1^{(5.3)}$ | $86.2^{(0.1)}$ | $66.4^{(0.7)}$ | $121^{(1.3)}$ | $51.3^{(3.6)}$ | $84.5^{(0.8)}$ | 2.2 | 19.4 |
| AMO-BO-qUCB | $95.4^{(1.6)}$ | $86.2^{(0.0)}$ | $66.3^{(1.1)}$ | $121^{(1.3)}$ | $50.2^{(2.8)}$ | $83.6^{(1.0)}$ | 2.7 | 20.2 |
| DynMO-BO-qUCB | $93.6^{(3.0)}$ | $86.2^{(0.1)}$ | $66.0^{(0.9)}$ | $121^{(0.0)}$ | $49.9^{(3.0)}$ | $83.9^{(1.1)}$ | 2.7 | 20.0 |
| DynAMO-BO-qUCB | $95.1^{(1.9)}$ | $86.2^{(0.0)}$ | $66.7^{(1.5)}$ | $121^{(0.0)}$ | $48.1^{(4.0)}$ | $86.9^{(4.5)}$ | 2.2 | 20.5 |
| Baseline-CMA-ES | $87.6^{(8.3)}$ | $86.2^{(0.0)}$ | $66.1^{(1.0)}$ | $106^{(5.9)}$ | $49.0^{(1.0)}$ | $72.2^{(0.1)}$ | 2.8 | 14.4 |
| AMO-CMA-ES | $90.4^{(4.4)}$ | $86.2^{(0.0)}$ | $66.2^{(1.6)}$ | $121^{(0.0)}$ | $45.2^{(3.5)}$ | $72.2^{(0.1)}$ | 1.8 | 16.7 |
| DynMO-CMA-ES | $85.2^{(10.1)}$ | $86.2^{(0.0)}$ | $65.0^{(0.6)}$ | $104^{(7.8)}$ | $51.6^{(2.0)}$ | $83.6^{(3.1)}$ | 2.7 | 15.8 |
| DynAMO-CMA-ES | $89.8^{(3.6)}$ | $85.7^{(5.8)}$ | $63.9^{(0.9)}$ | $117^{(6.7)}$ | $50.6^{(4.8)}$ | $78.5^{(5.5)}$ | 2.7 | 17.5 |
| CoSyNE | $61.7^{(10.0)}$ | $57.3^{(9.6)}$ | $63.6^{(0.4)}$ | $94.8^{(10.1)}$ | $37.0^{(4.1)}$ | $62.7^{(13.1)}$ | 3.5 | -0.6 |
| AMO-CoSyNE | $79.8^{(10.6)}$ | $68.0^{(12.5)}$ | $64.2^{(0.9)}$ | $99.4^{(15.0)}$ | $37.0^{(4.1)}$ | $62.7^{(13.1)}$ | 2.2 | 5.0 |
| DynMO-CoSyNE | $63.6^{(10.1)}$ | $59.3^{(10.8)}$ | $63.9^{(1.6)}$ | $90.1^{(12.7)}$ | $37.0^{(4.1)}$ | $62.7^{(13.1)}$ | 3.0 | -0.7 |
| DynAMO-CoSyNE | $91.3^{(4.4)}$ | $77.2^{(11.6)}$ | $63.9^{(0.9)}$ | $114^{(7.0)}$ | $40.6^{(8.6)}$ | $67.5^{(14.1)}$ | 1.2 | 12.3 |

| Median@128 | TFBind8 | UTR | ChEMBL | Molecule | Superconductor | D'Kitty | Rank ↓ | Opt. Gap ↑ |
|---|---|---|---|---|---|---|---|---|
| Dataset $\mathcal{D}$ | 33.7 | 42.8 | 50.9 | 87.6 | 6.7 | 77.8 | — | — |
| Grad. | $58.1^{(6.1)}$ | $58.6^{(13.1)}$ | $59.3^{(8.6)}$ | $85.3^{(7.7)}$ | $36.0^{(6.7)}$ | $65.1^{(14.4)}$ | 3.2 | 10.5 |
| AMO-Grad. | $63.8^{(13.7)}$ | $75.3^{(9.9)}$ | $60.1^{(3.3)}$ | $91.6^{(11.2)}$ | $46.0^{(6.7)}$ | $90.1^{(14.4)}$ | 1.2 | 21.2 |
| DynMO-Grad. | $50.5^{(6.5)}$ | $58.6^{(13.1)}$ | $59.7^{(8.7)}$ | $85.1^{(8.1)}$ | $36.3^{(6.9)}$ | $70.0^{(12.0)}$ | 3.0 | 10.1 |
| DynAMO-Grad. | $47.0^{(2.8)}$ | $69.8^{(6.0)}$ | $61.9^{(2.2)}$ | $85.9^{(0.4)}$ | $23.4^{(8.5)}$ | $68.7^{(12.1)}$ | 2.7 | 9.5 |
| Adam | $54.7^{(8.8)}$ | $60.4^{(12.7)}$ | $59.2^{(8.6)}$ | $87.9^{(10.0)}$ | $37.4^{(6.2)}$ | $56.8^{(19.8)}$ | 2.3 | 9.5 |
| AMO-Adam | $49.5^{(8.9)}$ | $55.7^{(12.7)}$ | $57.7^{(9.1)}$ | $84.3^{(9.6)}$ | $37.4^{(6.2)}$ | $87.8^{(4.3)}$ | 3.0 | 12.1 |
| DynMO-Adam | $54.0^{(9.9)}$ | $60.5^{(12.6)}$ | $59.3^{(8.6)}$ | $85.9^{(10.8)}$ | $37.7^{(6.4)}$ | $63.6^{(15.6)}$ | 2.2 | 10.2 |
| DynAMO-Adam | $47.7^{(3.0)}$ | $69.0^{(5.2)}$ | $62.4^{(1.9)}$ | $86.4^{(0.6)}$ | $23.0^{(6.0)}$ | $65.6^{(14.1)}$ | 2.3 | 9.1 |
| BO-qEI | $48.5^{(1.5)}$ | $59.9^{(2.0)}$ | $63.3^{(0.0)}$ | $86.7^{(0.6)}$ | $28.7^{(1.8)}$ | $72.4^{(1.8)}$ | 2.5 | 10.0 |
| AMO-BO-qEI | $46.4^{(1.8)}$ | $63.4^{(3.3)}$ | $63.3^{(0.0)}$ | $86.3^{(0.5)}$ | $28.9^{(1.1)}$ | $79.1^{(0.7)}$ | 2.2 | 11.3 |
| DynMO-BO-qEI | $50.5^{(1.5)}$ | $61.1^{(2.8)}$ | $63.3^{(0.0)}$ | $86.4^{(0.7)}$ | $28.4^{(0.7)}$ | $79.1^{(0.9)}$ | 2.3 | 11.5 |
| DynAMO-BO-qEI | $51.5^{(0.9)}$ | $65.6^{(3.1)}$ | $63.3^{(0.0)}$ | $86.7^{(0.6)}$ | $23.5^{(2.4)}$ | $77.0^{(0.7)}$ | 2.0 | 11.3 |
| BO-qUCB | $50.3^{(1.8)}$ | $62.1^{(3.4)}$ | $63.3^{(0.0)}$ | $86.6^{(0.6)}$ | $31.7^{(1.2)}$ | $74.4^{(0.6)}$ | 1.7 | 11.5 |
| AMO-BO-qUCB | $47.9^{(1.9)}$ | $59.8^{(1.2)}$ | $63.3^{(0.0)}$ | $86.0^{(0.6)}$ | $33.1^{(2.9)}$ | $73.8^{(1.2)}$ | 2.8 | 10.7 |
| DynMO-BO-qUCB | $50.3^{(1.7)}$ | $60.1^{(2.2)}$ | $63.3^{(0.0)}$ | $86.4^{(0.6)}$ | $32.4^{(2.7)}$ | $74.3^{(0.8)}$ | 2.0 | 11.2 |
| DynAMO-BO-qUCB | $48.8^{(1.8)}$ | $65.9^{(3.7)}$ | $63.3^{(0.0)}$ | $86.5^{(0.5)}$ | $22.7^{(2.0)}$ | $50.4^{(14.6)}$ | 2.5 | 6.3 |
| CMA-ES | $50.7^{(2.7)}$ | $71.7^{(10.4)}$ | $63.3^{(0.0)}$ | $83.9^{(1.0)}$ | $37.9^{(0.7)}$ | $59.3^{(10.9)}$ | 2.2 | 11.2 |
| AMO-CMA-ES | $44.2^{(0.8)}$ | $72.7^{(3.8)}$ | $62.7^{(1.1)}$ | $86.1^{(0.5)}$ | $21.4^{(2.0)}$ | $54.9^{(9.6)}$ | 3.2 | 7.1 |
| DynMO-CMA-ES | $50.7^{(2.7)}$ | $75.2^{(9.1)}$ | $63.3^{(0.0)}$ | $82.5^{(1.2)}$ | $38.7^{(3.9)}$ | $65.8^{(9.1)}$ | 1.7 | 12.8 |
| DynAMO-CMA-ES | $45.3^{(2.4)}$ | $65.8^{(8.9)}$ | $59.3^{(3.8)}$ | $99.0^{(12.1)}$ | $22.5^{(5.1)}$ | $60.6^{(15.0)}$ | 2.8 | 8.8 |
| CoSyNE | $55.3^{(8.0)}$ | $53.6^{(10.2)}$ | $60.8^{(3.2)}$ | $87.4^{(16.6)}$ | $36.6^{(4.4)}$ | $59.3^{(14.5)}$ | 2.5 | 8.9 |
| AMO-CoSyNE | $59.5^{(12.0)}$ | $63.5^{(11.2)}$ | $55.4^{(9.6)}$ | $84.2^{(17.2)}$ | $36.6^{(4.4)}$ | $59.3^{(14.5)}$ | 2.2 | 9.8 |
| DynMO-CoSyNE | $59.2^{(10.8)}$ | $55.6^{(8.2)}$ | $60.4^{(5.9)}$ | $87.9^{(12.5)}$ | $36.6^{(4.4)}$ | $59.3^{(14.5)}$ | 2.3 | 9.9 |
| DynAMO-CoSyNE | $53.8^{(11.0)}$ | $63.4^{(11.5)}$ | $59.3^{(3.8)}$ | $99.0^{(12.1)}$ | $20.5^{(5.8)}$ | $60.6^{(15.0)}$ | 2.5 | 9.5 |

*Table A12.* **Diversity of Design Candidates in Ablation of Adversarial Critic Feedback (AMO) and Diversity in (DynMO) Model-based Optimization.** We evaluate our method (1) with the KL-divergence penalized-MBO objective as in (11) only (**Dyn**MO); (2) with the adversarial source critic-dependent constraint as introduced by Yao et al. (2024) only (**A**MO); and (3) with both algorithmic components as in **DynAMO** described in **Algorithm 1**. We report the pairwise diversity (resp., minimum novelty) oracle score achieved by the 128 evaluated designs in the top (resp., bottom) table. Metrics are reported mean$^{(\text{95\% confidence interval})}$ across 10 random seeds, where higher is better. **Bolded** entries indicate average scores with an overlapping 95% confidence interval to the best performing method. **Bolded** (resp., Underlined) Rank and Optimality Gap (Opt. Gap) metrics indicate the best (resp., second best) for a given optimizer.

| Pairwise Diversity@128 | TFBind8 | UTR | ChEMBL | Molecule | Superconductor | D'Kitty | Rank ↓ | Opt. Gap ↑ |
|---|---|---|---|---|---|---|---|---|
| Dataset $\mathcal{D}$ | 65.9 | 57.3 | 60.0 | 36.7 | 66.0 | 85.7 | — | — |
| Grad. | $12.5^{(8.0)}$ | $7.8^{(8.8)}$ | $7.9^{(7.8)}$ | $24.1^{(13.3)}$ | $0.0^{(0.0)}$ | $0.0^{(0.0)}$ | 3.0 | -53.2 |
| AMO-Grad. | $17.3^{(12.8)}$ | $11.2^{(10.3)}$ | $6.9^{(7.7)}$ | $22.1^{(10.5)}$ | $0.0^{(0.0)}$ | $1.5^{(3.2)}$ | 2.7 | -52.1 |
| **Dyn**MO-Grad. | $20.9^{(15.1)}$ | $3.0^{(3.2)}$ | $\mathbf{58.2^{(24.0)}}$ | $13.5^{(8.6)}$ | $0.0^{(0.0)}$ | $0.0^{(0.0)}$ | 3.0 | -46.0 |
| **DynA**MO-Grad. | $\mathbf{66.9^{(6.9)}}$ | $\mathbf{68.2^{(10.8)}}$ | $\mathbf{77.2^{(21.5)}}$ | $\mathbf{93.0^{(1.2)}}$ | $\mathbf{129^{(55.3)}}$ | $\mathbf{104^{(56.1)}}$ | **1.0** | **27.8** |
| Adam | $12.0^{(12.3)}$ | $11.0^{(12.1)}$ | $4.8^{(3.8)}$ | $16.8^{(12.4)}$ | $6.4^{(14.5)}$ | $6.2^{(14.0)}$ | 3.0 | -52.4 |
| AMO-Adam | $15.1^{(11.2)}$ | $10.3^{(11.5)}$ | $12.1^{(11.3)}$ | $19.6^{(15.2)}$ | $0.3^{(0.8)}$ | $2.6^{(3.9)}$ | 3.0 | -51.9 |
| **Dyn**MO-Adam | $13.1^{(11.4)}$ | $10.3^{(9.6)}$ | $57.0^{(26.0)}$ | $23.8^{(15.1)}$ | $6.4^{(14.5)}$ | $0.0^{(0.0)}$ | 2.8 | -43.5 |
| **DynA**MO-Adam | $\mathbf{54.8^{(8.9)}}$ | $\mathbf{72.3^{(3.4)}}$ | $\mathbf{84.8^{(9.2)}}$ | $\mathbf{89.9^{(5.3)}}$ | $\mathbf{158^{(37.3)}}$ | $\mathbf{126^{(57.3)}}$ | **1.0** | **35.7** |
| BO-qEI | $73.7^{(0.6)}$ | $\mathbf{73.8^{(0.5)}}$ | $99.3^{(0.1)}$ | $93.0^{(0.5)}$ | $190^{(0.8)}$ | $124^{(7.4)}$ | 3.7 | 47.1 |
| AMO-BO-qEI | $74.0^{(0.6)}$ | $\mathbf{74.3^{(0.4)}}$ | $99.3^{(0.1)}$ | $93.3^{(0.4)}$ | $193^{(1.2)}$ | $17.7^{(3.5)}$ | 2.8 | 30.0 |
| **Dyn**MO-BO-qEI | $\mathbf{74.5^{(0.3)}}$ | $\mathbf{74.3^{(0.6)}}$ | $99.3^{(0.1)}$ | $\mathbf{93.3^{(0.7)}}$ | $\mathbf{200^{(3.0)}}$ | $135^{(11.2)}$ | 2.3 | 50.8 |
| **DynA**MO-BO-qEI | $\mathbf{74.8^{(0.2)}}$ | $\mathbf{74.6^{(0.3)}}$ | $99.4^{(0.1)}$ | $93.5^{(0.4)}$ | $198^{(1.9)}$ | $277^{(59.7)}$ | **1.2** | **74.2** |
| BO-qUCB | $73.9^{(0.5)}$ | $\mathbf{74.3^{(0.4)}}$ | $99.4^{(0.1)}$ | $93.6^{(0.5)}$ | $198^{(10.3)}$ | $94.1^{(3.9)}$ | 2.5 | 43.5 |
| AMO-BO-qUCB | $74.0^{(0.5)}$ | $\mathbf{74.3^{(0.3)}}$ | $99.3^{(0.1)}$ | $93.4^{(0.4)}$ | $190^{(9.3)}$ | $22.0^{(2.1)}$ | 3.7 | 30.3 |
| **Dyn**MO-BO-qUCB | $\mathbf{74.7^{(0.2)}}$ | $\mathbf{74.3^{(0.4)}}$ | $99.2^{(0.1)}$ | $93.6^{(0.5)}$ | $198^{(12.0)}$ | $92.6^{(3.8)}$ | 2.2 | 43.5 |
| **DynA**MO-BO-qUCB | $74.3^{(0.5)}$ | $\mathbf{74.4^{(0.6)}}$ | $99.3^{(0.1)}$ | $93.5^{(0.6)}$ | $211^{(22.8)}$ | $175^{(44.7)}$ | **1.7** | **59.4** |
| CMA-ES | $47.2^{(11.2)}$ | $44.6^{(15.9)}$ | $\mathbf{93.5^{(2.0)}}$ | $66.2^{(9.4)}$ | $12.8^{(0.6)}$ | $164^{(10.6)}$ | 2.3 | 9.5 |
| AMO-CMA-ES | $39.6^{(15.5)}$ | $53.4^{(8.4)}$ | $84.8^{(4.8)}$ | $61.3^{(14.6)}$ | $\mathbf{173^{(19.4)}}$ | $59.9^{(19.6)}$ | 2.3 | 16.8 |
| **Dyn**MO-CMA-ES | $33.5^{(2.6)}$ | $11.1^{(1.1)}$ | $34.5^{(2.8)}$ | $4.5^{(5.0)}$ | $38.1^{(5.4)}$ | $14.4^{(1.2)}$ | 3.8 | -39.3 |
| **DynA**MO-CMA-ES | $\mathbf{73.6^{(0.6)}}$ | $\mathbf{73.1^{(3.1)}}$ | $72.0^{(3.1)}$ | $\mathbf{94.0^{(0.5)}}$ | $97.8^{(13.2)}$ | $292^{(83.5)}$ | **1.5** | **55.2** |
| CoSyNE | $5.6^{(5.0)}$ | $\mathbf{12.7^{(9.8)}}$ | $28.2^{(11.3)}$ | $12.2^{(7.3)}$ | $0.0^{(0.0)}$ | $0.0^{(0.0)}$ | 2.8 | -52.1 |
| AMO-CoSyNE | $5.2^{(5.7)}$ | $9.1^{(9.0)}$ | $28.4^{(15.7)}$ | $7.1^{(8.0)}$ | $0.0^{(0.0)}$ | $0.0^{(0.0)}$ | 3.7 | -53.6 |
| **Dyn**MO-CoSyNE | $\mathbf{27.4^{(10.3)}}$ | $\mathbf{13.0^{(11.8)}}$ | $\mathbf{53.3^{(24.2)}}$ | $18.1^{(13.8)}$ | $0.0^{(0.0)}$ | $0.0^{(0.0)}$ | 2.2 | -43.3 |
| **DynA**MO-CoSyNE | $18.1^{(13.0)}$ | $\mathbf{20.3^{(2.3)}}$ | $35.0^{(17.9)}$ | $\mathbf{22.8^{(11.9)}}$ | $\mathbf{74.4^{(46.3)}}$ | $\mathbf{77.0^{(35.9)}}$ | **1.3** | **-20.7** |

| Minimum Novelty@128 | TFBind8 | UTR | ChEMBL | Molecule | Superconductor | D'Kitty | Rank ↓ | Opt. Gap ↑ |
|---|---|---|---|---|---|---|---|---|
| Dataset $\mathcal{D}$ | 0.0 | 0.0 | 0.0 | 0.0 | 0.0 | 0.0 | — | — |
| Grad. | $\mathbf{21.2^{(3.0)}}$ | $\mathbf{51.7^{(2.9)}}$ | $\mathbf{97.4^{(3.9)}}$ | $\mathbf{79.5^{(19.7)}}$ | $\mathbf{95.0^{(0.7)}}$ | $102^{(6.1)}$ | 2.3 | 74.5 |
| AMO-Grad. | $14.0^{(2.0)}$ | $46.7^{(2.7)}$ | $\mathbf{96.8^{(3.9)}}$ | $\mathbf{76.8^{(19.7)}}$ | $83.8^{(6.8)}$ | $31.5^{(3.6)}$ | 3.8 | 58.3 |
| **Dyn**MO-Grad. | $\mathbf{21.9^{(3.0)}}$ | $\mathbf{53.6^{(2.1)}}$ | $93.1^{(6.6)}$ | $86.4^{(10.2)}$ | $\mathbf{95.0^{(0.7)}}$ | $102^{(6.1)}$ | 1.8 | 75.4 |
| **DynA**MO-Grad. | $\mathbf{21.1^{(1.1)}}$ | $52.2^{(1.3)}$ | $\mathbf{98.6^{(1.5)}}$ | $\mathbf{85.8^{(1.0)}}$ | $\mathbf{95.0^{(0.4)}}$ | $107^{(6.7)}$ | **1.7** | **76.7** |
| Adam | $\mathbf{23.7^{(2.8)}}$ | $51.1^{(3.5)}$ | $95.5^{(5.3)}$ | $\mathbf{79.3^{(21.2)}}$ | $\mathbf{94.8^{(0.7)}}$ | $103^{(6.3)}$ | 2.7 | 74.5 |
| AMO-Adam | $\mathbf{23.7^{(3.1)}}$ | $51.3^{(3.4)}$ | $95.0^{(5.1)}$ | $\mathbf{80.0^{(20.6)}}$ | $84.8^{(6.4)}$ | $27.3^{(3.4)}$ | 3.0 | 60.4 |
| **Dyn**MO-Adam | $22.9^{(2.3)}$ | $51.6^{(2.9)}$ | $\mathbf{99.2^{(0.6)}}$ | $87.3^{(9.7)}$ | $\mathbf{94.7^{(0.7)}}$ | $103^{(6.3)}$ | 1.8 | 76.4 |
| **DynA**MO-Adam | $14.7^{(1.9)}$ | $46.2^{(0.5)}$ | $\mathbf{98.7^{(1.2)}}$ | $85.9^{(1.8)}$ | $\mathbf{94.9^{(0.4)}}$ | $108^{(7.2)}$ | 2.3 | 74.7 |
| BO-qEI | $\mathbf{21.8^{(0.5)}}$ | $51.5^{(0.3)}$ | $\mathbf{97.6^{(0.3)}}$ | $85.4^{(1.5)}$ | $94.6^{(0.1)}$ | $106^{(2.9)}$ | 2.5 | 76.2 |
| AMO-BO-qEI | $15.4^{(0.3)}$ | $\mathbf{51.8^{(0.2)}}$ | $\mathbf{97.8^{(0.3)}}$ | $84.9^{(0.9)}$ | $85.1^{(0.4)}$ | $14.3^{(1.5)}$ | 3.3 | 58.2 |
| **Dyn**MO-BO-qEI | $20.4^{(0.4)}$ | $\mathbf{51.8^{(0.1)}}$ | $\mathbf{97.7^{(0.3)}}$ | $85.7^{(1.3)}$ | $94.4^{(0.5)}$ | $108^{(3.2)}$ | 2.2 | 76.4 |
| **DynA**MO-BO-qEI | $\mathbf{21.0^{(0.5)}}$ | $\mathbf{51.9^{(0.2)}}$ | $\mathbf{97.4^{(0.4)}}$ | $85.2^{(0.9)}$ | $94.8^{(0.1)}$ | $126^{(14.6)}$ | 2.0 | **79.4** |
| BO-qUCB | $\mathbf{21.6^{(0.3)}}$ | $\mathbf{51.7^{(0.2)}}$ | $\mathbf{97.9^{(0.4)}}$ | $85.3^{(1.1)}$ | $93.8^{(0.6)}$ | $98.8^{(1.1)}$ | 2.0 | 74.8 |
| AMO-BO-qUCB | $\mathbf{21.9^{(0.4)}}$ | $\mathbf{51.7^{(0.3)}}$ | $97.5^{(0.4)}$ | $85.2^{(1.0)}$ | $81.9^{(1.9)}$ | $25.9^{(1.4)}$ | 2.7 | 60.7 |
| **Dyn**MO-BO-qUCB | $20.7^{(0.4)}$ | $\mathbf{51.8^{(0.2)}}$ | $97.1^{(0.5)}$ | $84.9^{(0.6)}$ | $93.2^{(1.4)}$ | $98.0^{(1.8)}$ | 3.2 | 74.3 |
| **DynA**MO-BO-qUCB | $\mathbf{21.4^{(0.5)}}$ | $\mathbf{51.7^{(0.2)}}$ | $97.1^{(0.5)}$ | $85.3^{(1.1)}$ | $94.7^{(0.2)}$ | $109^{(4.5)}$ | 2.2 | **76.6** |
| CMA-ES | $16.5^{(2.1)}$ | $47.8^{(1.0)}$ | $96.5^{(0.7)}$ | $73.0^{(18.0)}$ | $\mathbf{100^{(0.0)}}$ | $100^{(0.0)}$ | 2.3 | 72.3 |
| AMO-CMA-ES | $\mathbf{24.3^{(0.9)}}$ | $\mathbf{53.3^{(1.4)}}$ | $95.0^{(1.5)}$ | $72.5^{(23.6)}$ | $85.6^{(3.0)}$ | $41.5^{(2.0)}$ | 3.0 | 62.0 |
| **Dyn**MO-CMA-ES | $14.3^{(0.3)}$ | $46.1^{(0.4)}$ | $\mathbf{98.2^{(1.0)}}$ | $83.3^{(1.0)}$ | $\mathbf{100^{(0.0)}}$ | $100^{(0.0)}$ | 2.0 | 73.7 |
| **DynA**MO-CMA-ES | $12.9^{(0.8)}$ | $48.0^{(1.6)}$ | $96.7^{(3.5)}$ | $81.8^{(13.4)}$ | $94.5^{(0.7)}$ | $112^{(7.8)}$ | 2.3 | **74.3** |
| CoSyNE | $\mathbf{24.5^{(3.5)}}$ | $49.7^{(3.1)}$ | $\mathbf{98.5^{(1.6)}}$ | $86.6^{(12.7)}$ | $\mathbf{93.2^{(1.0)}}$ | $91.9^{(2.0)}$ | 1.8 | 74.1 |
| AMO-CoSyNE | $22.8^{(2.8)}$ | $\mathbf{50.8^{(1.5)}}$ | $90.8^{(14.2)}$ | $\mathbf{91.9^{(3.4)}}$ | $86.0^{(3.3)}$ | $29.6^{(3.6)}$ | 2.5 | 62.0 |
| **Dyn**MO-CoSyNE | $19.3^{(5.5)}$ | $46.3^{(2.3)}$ | $93.4^{(3.7)}$ | $88.3^{(5.2)}$ | $85.7^{(3.2)}$ | $29.6^{(3.6)}$ | 3.2 | 60.4 |
| **DynA**MO-CoSyNE | $17.8^{(5.5)}$ | $48.4^{(2.3)}$ | $\mathbf{96.7^{(3.5)}}$ | $80.2^{(12.9)}$ | $94.5^{(0.7)}$ | $112^{(7.8)}$ | 2.5 | **75.0** |

*Table A13.* **Diversity of Design Candidates in Ablation of Adversarial Critic Feedback (AMO) and Diversity in (Dyn MO) Model-based Optimization (cont.).** We evaluate our method (1) with the KL-divergence penalized-MBO objective as in (11) only (**Dyn**MO); (2) with the adversarial source critic-dependent constraint as introduced by Yao et al. (2024) only (AMO); and (3) with both algorithmic components as in **DynAMO** described in **Algorithm 1**. We report the $L_1$ coverage score achieved by the 128 evaluated designs as mean$^{(95\% \text{ confidence interval})}$ across 10 random seeds, where higher is better. **Bolded** entries indicate average scores with an overlapping 95% confidence interval to the best performing method. **Bolded** (resp., Underlined) Rank and Optimality Gap (Opt. Gap) metrics indicate the best (resp., second best) for a given backbone optimizer.

| $L_1$ **Coverage@128** | **TFBind8** | **UTR** | **ChEMBL** | **Molecule** | **Superconductor** | **D'Kitty** | **Rank ↓** | **Opt. Gap ↑** |
|---|---|---|---|---|---|---|---|---|
| Dataset $\mathcal{D}$ | 0.42 | 0.31 | 1.42 | 0.68 | 6.26 | 0.58 | — | — |
| Grad. | $0.16^{(0.10)}$ | $0.20^{(0.13)}$ | $0.21^{(0.10)}$ | $0.42^{(0.18)}$ | $0.00^{(0.00)}$ | $0.00^{(0.00)}$ | 3.3 | -1.44 |
| AMO-Grad. | $0.17^{(0.10)}$ | $0.24^{(0.13)}$ | $0.25^{(0.16)}$ | $0.37^{(0.11)}$ | $0.00^{(0.00)}$ | $0.09^{(0.14)}$ | 2.5 | -1.42 |
| **Dyn**MO-Grad. | $\mathbf{0.20^{(0.13)}}$ | $0.22^{(0.15)}$ | $\mathbf{1.23^{(0.63)}}$ | $0.28^{(0.17)}$ | $0.00^{(0.00)}$ | $0.00^{(0.00)}$ | 2.8 | -1.29 |
| **DynAMO**-Grad. | $\mathbf{0.36^{(0.04)}}$ | $\mathbf{0.52^{(0.06)}}$ | $\mathbf{1.46^{(0.38)}}$ | $\mathbf{2.49^{(0.06)}}$ | $\mathbf{6.47^{(1.24)}}$ | $\mathbf{5.85^{(1.35)}}$ | **1.0** | **1.25** |
| Adam | $0.11^{(0.06)}$ | $0.22^{(0.09)}$ | $0.23^{(0.15)}$ | $0.48^{(0.31)}$ | $0.27^{(0.55)}$ | $0.24^{(0.49)}$ | 2.8 | -1.35 |
| AMO-Adam | $0.14^{(0.06)}$ | $0.22^{(0.10)}$ | $0.35^{(0.24)}$ | $0.50^{(0.34)}$ | $0.26^{(0.53)}$ | $0.09^{(0.12)}$ | 3.0 | -1.35 |
| **Dyn**MO-Adam | $\mathbf{0.20^{(0.12)}}$ | $0.21^{(0.15)}$ | $\mathbf{1.10^{(0.60)}}$ | $0.45^{(0.29)}$ | $0.27^{(0.55)}$ | $0.03^{(0.00)}$ | 3.0 | -1.23 |
| **DynAMO**-Adam | $\mathbf{0.33^{(0.05)}}$ | $\mathbf{0.55^{(0.03)}}$ | $\mathbf{1.44^{(0.39)}}$ | $\mathbf{2.40^{(0.16)}}$ | $\mathbf{7.06^{(0.73)}}$ | $\mathbf{6.91^{(0.71)}}$ | **1.0** | **1.50** |
| BO-qEI | $\mathbf{0.41^{(0.02)}}$ | $\mathbf{0.55^{(0.01)}}$ | $2.37^{(0.03)}$ | $2.11^{(0.15)}$ | $7.84^{(0.01)}$ | $6.61^{(0.33)}$ | 2.8 | 1.70 |
| AMO-BO-qEI | $\mathbf{0.40^{(0.03)}}$ | $\mathbf{0.55^{(0.01)}}$ | $2.38^{(0.10)}$ | $\mathbf{2.53^{(0.05)}}$ | $7.45^{(0.01)}$ | $1.29^{(0.08)}$ | 3.7 | 0.82 |
| **Dyn**MO-BO-qEI | $\mathbf{0.40^{(0.02)}}$ | $\mathbf{0.55^{(0.01)}}$ | $\mathbf{2.42^{(0.05)}}$ | $\mathbf{2.55^{(0.03)}}$ | $7.83^{(0.02)}$ | $6.72^{(0.48)}$ | 2.3 | 1.80 |
| **DynAMO**-BO-qEI | $\mathbf{0.42^{(0.01)}}$ | $\mathbf{0.56^{(0.01)}}$ | $\mathbf{2.47^{(0.03)}}$ | $\mathbf{2.54^{(0.03)}}$ | $\mathbf{7.87^{(0.01)}}$ | $\mathbf{7.92^{(0.04)}}$ | **1.2** | **2.02** |
| BO-qUCB | $\mathbf{0.40^{(0.02)}}$ | $0.54^{(0.01)}$ | $\mathbf{2.40^{(0.05)}}$ | $\mathbf{2.52^{(0.07)}}$ | $7.78^{(0.04)}$ | $6.64^{(0.09)}$ | 2.8 | 1.77 |
| AMO-BO-qUCB | $\mathbf{0.40^{(0.01)}}$ | $\mathbf{0.56^{(0.01)}}$ | $2.39^{(0.05)}$ | $\mathbf{2.52^{(0.04)}}$ | $7.37^{(0.09)}$ | $1.34^{(0.04)}$ | 3.3 | 0.82 |
| **Dyn**MO-BO-qUCB | $0.39^{(0.02)}$ | $\mathbf{0.55^{(0.00)}}$ | $\mathbf{2.40^{(0.08)}}$ | $\mathbf{2.52^{(0.04)}}$ | $7.76^{(0.07)}$ | $6.64^{(0.13)}$ | 2.5 | 1.77 |
| **DynAMO**-BO-qUCB | $\mathbf{0.40^{(0.02)}}$ | $\mathbf{0.55^{(0.01)}}$ | $\mathbf{2.47^{(0.07)}}$ | $\mathbf{2.54^{(0.05)}}$ | $\mathbf{7.88^{(0.03)}}$ | $\mathbf{7.80^{(0.23)}}$ | **1.3** | **2.00** |
| CMA-ES | $\mathbf{0.33^{(0.05)}}$ | $0.48^{(0.04)}$ | $\mathbf{2.18^{(0.04)}}$ | $1.82^{(0.12)}$ | $3.26^{(1.42)}$ | $3.77^{(1.36)}$ | 2.3 | 0.36 |
| AMO-CMA-ES | $0.31^{(0.04)}$ | $0.51^{(0.01)}$ | $2.17^{(0.06)}$ | $1.83^{(0.15)}$ | $3.37^{(0.49)}$ | $3.12^{(0.40)}$ | 2.5 | 0.27 |
| **Dyn**MO-CMA-ES | $\mathbf{0.34^{(0.09)}}$ | $0.39^{(0.12)}$ | $0.66^{(0.43)}$ | $0.60^{(0.42)}$ | $1.85^{(2.28)}$ | $0.94^{(1.73)}$ | 3.7 | -0.81 |
| **DynAMO**-CMA-ES | $\mathbf{0.40^{(0.03)}}$ | $\mathbf{0.56^{(0.01)}}$ | $1.82^{(0.72)}$ | $\mathbf{2.54^{(0.05)}}$ | $4.75^{(2.16)}$ | $3.29^{(1.56)}$ | **1.5** | **0.62** |
| CoSyNE | $0.10^{(0.07)}$ | $\mathbf{0.22^{(0.10)}}$ | $\mathbf{0.39^{(0.20)}}$ | $\mathbf{0.27^{(0.13)}}$ | $0.10^{(0.00)}$ | $0.10^{(0.00)}$ | 2.8 | -1.41 |
| AMO-CoSyNE | $\mathbf{0.12^{(0.08)}}$ | $\mathbf{0.22^{(0.15)}}$ | $\mathbf{0.53^{(0.18)}}$ | $0.14^{(0.13)}$ | $0.10^{(0.00)}$ | $0.02^{(0.00)}$ | 2.8 | -1.42 |
| **Dyn**MO-CoSyNE | $\mathbf{0.28^{(0.09)}}$ | $\mathbf{0.32^{(0.13)}}$ | $0.33^{(0.23)}$ | $\mathbf{0.79^{(0.65)}}$ | $0.10^{(0.00)}$ | $0.02^{(0.00)}$ | 2.3 | -1.30 |
| **DynAMO**-CoSyNE | $\mathbf{0.21^{(0.11)}}$ | $0.18^{(0.09)}$ | $\mathbf{0.64^{(0.42)}}$ | $0.43^{(0.34)}$ | $\mathbf{1.85^{(0.22)}}$ | $\mathbf{0.94^{(0.17)}}$ | **1.8** | **-0.90** |

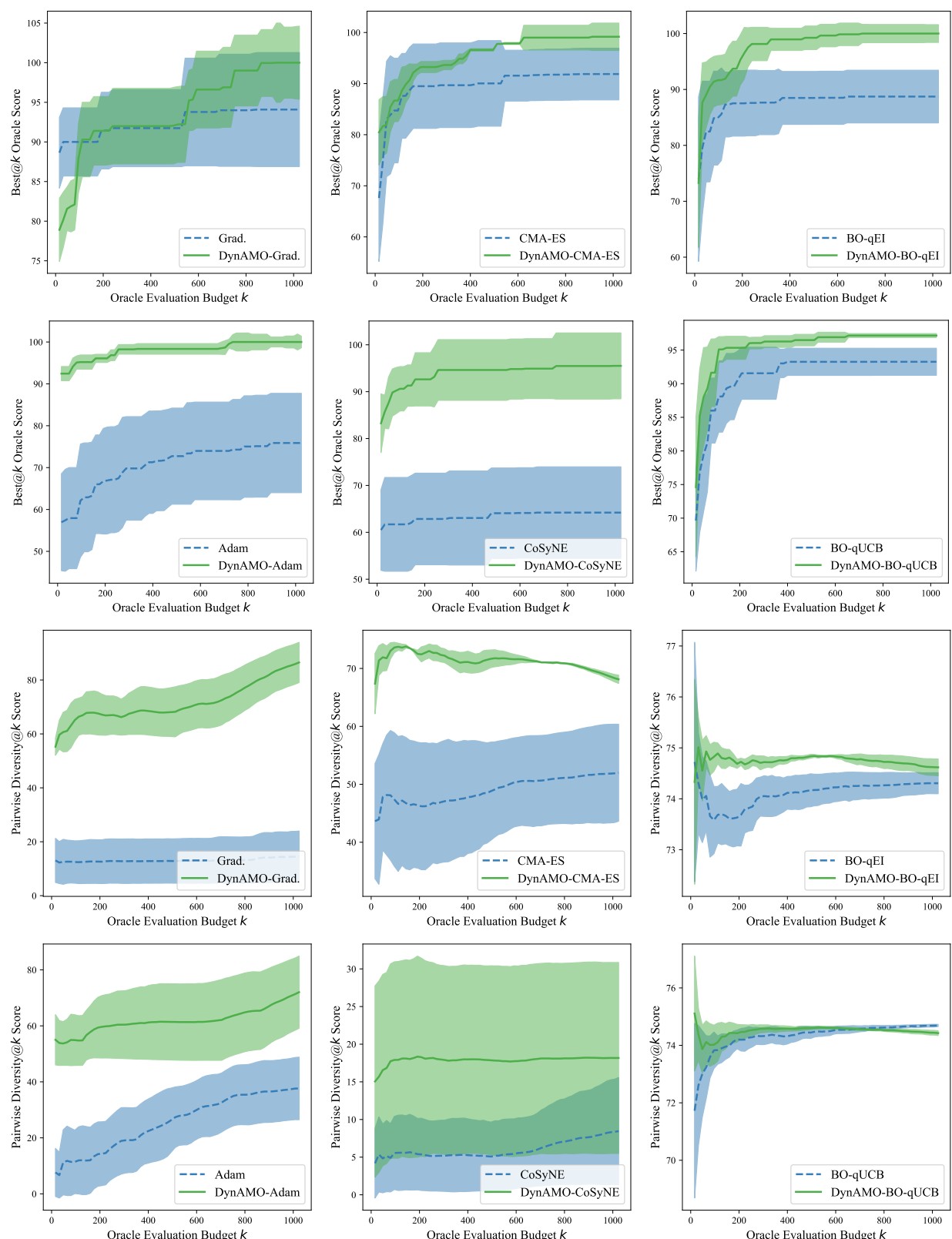

*Figure A6.* **Oracle Evaluation Budget Ablation.** We vary the allowed oracle evaluation budget $k$ in **Algorithm 1** between 16 and 1024, and report both the (**first two rows**) Best@128 Oracle Score and (**last two rows**) Pairwise Diversity score for $k$ final designs proposed by both DynAMO-augmented and base optimizers on the TFBind8 task. We plot the mean $\pm$ 95% confidence interval over 10 random seeds.

*Table A14.* **Optimization Initialization Ablation.** We evaluate both Sobol sequence-based and Top-$k$ initialization strategies for DynAMO with Gradient Ascent and other first-order MBO methods. We report the maximum oracle score (resp., pairwise diversity score) achieved out of 128 evaluated designs in the top (resp., bottom) table. Metrics are reported mean$^{(\text{95\% confidence interval})}$ across 10 random seeds, where higher is better. $\max(\mathcal{D})$ reports the top oracle score in the offline dataset. All metrics are multiplied by 100 for easier legibility. **Bolded** entries indicate the higher average scores for a given optimization method.

| Best@128 | TFBind8 | UTR | ChEMBL | Molecule | Superconductor | D'Kitty | Win Rate ↑ |
|---|---|---|---|---|---|---|---|
| Dataset $\mathcal{D}$ | 43.9 | 59.4 | 60.5 | 88.9 | 40.0 | 88.4 | — |
| Grad. (Sobol) | **90.0**$^{(4.3)}$ | **80.9**$^{(12.1)}$ | 60.2$^{(8.9)}$ | 88.8$^{(4.0)}$ | **36.0**$^{(6.8)}$ | 65.6$^{(14.5)}$ | 3/6 |
| Grad. (Top-$k$) | 85.1$^{(3.1)}$ | 64.0$^{(0.7)}$ | **63.3**$^{(0.0)}$ | **90.1**$^{(0.3)}$ | 27.1$^{(1.0)}$ | **67.8**$^{(0.0)}$ | 3/6 |
| COMs (Sobol) | 84.7$^{(5.3)}$ | 60.4$^{(2.2)}$ | 63.3$^{(0.0)}$ | 91.4$^{(0.4)}$ | 17.3$^{(0.5)}$ | 82.8$^{(2.9)}$ | 0/6 |
| COMs (Top-$k$) | **93.1**$^{(3.4)}$ | **67.0**$^{(0.9)}$ | **64.6**$^{(1.0)}$ | **97.1**$^{(1.6)}$ | **41.2**$^{(4.8)}$ | **91.8**$^{(0.9)}$ | 6/6 |
| RoMA (Sobol) | **96.5**$^{(0.0)}$ | **77.8**$^{(0.0)}$ | **63.3**$^{(0.0)}$ | **85.5**$^{(2.4)}$ | 46.5$^{(2.5)}$ | 93.9$^{(1.0)}$ | 4/6 |
| RoMA (Top-$k$) | **96.5**$^{(0.0)}$ | **77.8**$^{(0.0)}$ | **63.3**$^{(0.0)}$ | 84.7$^{(0.0)}$ | **49.8**$^{(1.4)}$ | **95.7**$^{(1.6)}$ | 6/6 |
| ROMO (Sobol) | 97.7$^{(1.2)}$ | **67.0**$^{(1.3)}$ | **68.3**$^{(0.5)}$ | 90.8$^{(0.4)}$ | **45.5**$^{(1.6)}$ | 86.1$^{(0.5)}$ | 3/6 |
| ROMO (Top-$k$) | **98.1**$^{(0.7)}$ | 66.8$^{(1.0)}$ | 63.0$^{(0.8)}$ | **91.8**$^{(0.9)}$ | 38.7$^{(2.5)}$ | **87.8**$^{(0.9)}$ | 3/6 |
| GAMBO (Sobol) | 73.1$^{(12.8)}$ | **77.1**$^{(9.6)}$ | **64.4**$^{(1.5)}$ | **92.8**$^{(8.0)}$ | **46.0**$^{(6.8)}$ | **90.6**$^{(14.5)}$ | 5/6 |
| GAMBO (Top-$k$) | **78.5**$^{(9.3)}$ | 68.3$^{(0.5)}$ | 63.0$^{(0.0)}$ | 90.6$^{(0.3)}$ | 27.1$^{(1.0)}$ | 77.8$^{(0.0)}$ | 1/6 |
| DynAMO (Sobol) | **90.3**$^{(4.7)}$ | **86.2**$^{(0.0)}$ | **64.4**$^{(2.5)}$ | **91.2**$^{(0.0)}$ | **44.2**$^{(7.8)}$ | **89.8**$^{(3.2)}$ | 6/6 |
| DynAMO (Top-$k$) | 81.9$^{(8.4)}$ | 64.4$^{(1.2)}$ | 63.3$^{(0.0)}$ | 90.8$^{(0.3)}$ | 29.4$^{(4.4)}$ | 75.3$^{(11.6)}$ | 0/6 |

| Pairwise Diversity@128 | TFBind8 | UTR | ChEMBL | Molecule | Superconductor | D'Kitty | Win Rate ↑ |
|---|---|---|---|---|---|---|---|
| Dataset $\mathcal{D}$ | 33.7 | 42.8 | 50.9 | 87.6 | 6.7 | 77.8 | — |
| Grad. (Sobol) | **12.5**$^{(8.0)}$ | 7.8$^{(8.8)}$ | 7.9$^{(7.8)}$ | 24.1$^{(13.3)}$ | **0.0**$^{(0.0)}$ | **0.0**$^{(0.0)}$ | 3/6 |
| Grad. (Top-$k$) | 8.3$^{(4.7)}$ | **40.3**$^{(3.6)}$ | **63.1**$^{(8.3)}$ | **28.4**$^{(6.0)}$ | **0.0**$^{(0.0)}$ | **0.0**$^{(0.0)}$ | 5/6 |
| COMs (Sobol) | 65.4$^{(0.5)}$ | 57.3$^{(0.1)}$ | 59.3$^{(1.1)}$ | **72.6**$^{(0.7)}$ | 43.9$^{(16.5)}$ | **33.8**$^{(1.7)}$ | 2/6 |
| COMs (Top-$k$) | **66.6**$^{(1.0)}$ | **57.4**$^{(0.2)}$ | **81.6**$^{(4.9)}$ | 3.8$^{(0.9)}$ | **99.5**$^{(25.8)}$ | 21.1$^{(23.5)}$ | 4/6 |
| RoMA (Sobol) | 21.0$^{(0.2)}$ | **3.8**$^{(0.0)}$ | **5.9**$^{(0.0)}$ | **1.8**$^{(0.0)}$ | **70.3**$^{(13.6)}$ | 8.1$^{(0.3}$ | 4/6 |
| RoMA (Top-$k$) | **21.3**$^{(0.3)}$ | **3.8**$^{(0.0)}$ | **5.9**$^{(0.2)}$ | **1.8**$^{(0.0)}$ | 49.4$^{(6.1)}$ | **14.8**$^{(0.6)}$ | 5/6 |
| ROMO (Sobol) | **64.4**$^{(1.3)}$ | 56.9$^{(0.2)}$ | **59.3**$^{(0.9)}$ | 39.0$^{(0.9)}$ | **58.3**$^{(12.9)}$ | 10.9$^{(0.5)}$ | 3/6 |
| ROMO (Top-$k$) | 62.1$^{(0.8)}$ | **57.1**$^{(0.1)}$ | 53.9$^{(0.6)}$ | **48.7**$^{(0.1)}$ | 51.7$^{(31.7)}$ | **22.1**$^{(5.5)}$ | 3/6 |
| GAMBO (Sobol) | 15.1$^{(11.2)}$ | 10.3$^{(11.5)}$ | 12.1$^{(11.3)}$ | 19.6$^{(15.2)}$ | **0.3**$^{(0.8)}$ | **2.6**$^{(3.9)}$ | 2/6 |
| GAMBO (Top-$k$) | **59.2**$^{(5.0)}$ | **54.1**$^{(2.5)}$ | **79.3**$^{(3.4)}$ | **33.4**$^{(1.9)}$ | 0.0$^{(0.0)}$ | 0.0$^{(0.0)}$ | 4/6 |
| DynAMO (Sobol) | **66.9**$^{(6.9)}$ | **68.2**$^{(10.8)}$ | **77.2**$^{(21.5)}$ | **93.0**$^{(1.2)}$ | **129**$^{(55.3)}$ | **104**$^{(56.1)}$ | 6/6 |
| DynAMO (Top-$k$) | 55.2$^{(10.5)}$ | 46.4$^{(5.2)}$ | 76.8$^{(4.5)}$ | 36.4$^{(3.1)}$ | 120$^{(30.0)}$ | 85.7$^{(50.0)}$ | 0/6 |

