# OpenReview forum: "Diversity By Design: Leveraging Distribution Matching for Offline Model-Based Optimization"
_ICML.cc/2025/Conference — ICML 2025 poster_

### Official Review · Reviewer_sKFn · 2025-03-09

**Overall Recommendation:** 4

**Summary:**

This paper introduces DynAMO - a method for offline model-based optimization whose objective is to produce a diverse distribution of various designs. The objective is clearly formulated, the method is carefully derived, and empirically evaluated. The empirical evaluation is very extensive and it is certainly impressive. The results show that the method is very strong in terms of the reward of the designs, and unusually strong in terms of design diversity. Some elements of the paper pertaining the technical clarity and the rigor of the mathematical statements could be slightly improved, but it definitely qualifies as a good paper.

## Update after rebuttal
The authors addressed my comments and did extra work to communicate that to me. The proposed method is now well-, and correctly-motivated theoretically. It showcases the desired properties (performance + diversity), the empirical evaluation is extensive, and the limitations are described. I strongly advocate for accepting this paper.

**Claims And Evidence:**

The claims are supported by evidence. There are several theoretical claims, all of which come with mathematical proofs. Some things like a bit problematic:

> Lemma 3.1: You need to make some assumptions about q(x) or loosen the statement to make it hold. q(x) could be non-zero on some measure-zero set of designs, and then your Inequality (16) in the proof does not hold.

> The formulation of the objective in Equation (5) is neat. But we can prove that its solution is an EBM distribution like the one in Equation (6), but with a different temperature parameter. Thus, I do not understand why to have a separate pure reward term. Lemma 3.3 makes my point even more explicit.

**Essential References Not Discussed:**

N/A

**Experimental Designs Or Analyses:**

I confirm the experimental design is valid. I had one question about clarity (see "Methods and Evaluation Criteria"). I have one qualm though:

> As your Table A1 shows, your method trades off reward for diversity. Maybe you get a good argmax but the quality of remaining designs, like the median, can be unimpressive. Your results in A1, especially in comparison to baseline Adam, show that you generally produce suboptimal, but very diverse, designs. It is not a deal breaker but it is an important limitation. I offer a deal: if you write it in Limitations, I will raise your score, and if you address my other concerns, I will raise your score further. But the Limitations paragraph is a condition.

**Methods And Evaluation Criteria:**

The experimental design is very reasonable, and broad. The authors cover enough tasks and many baselines. The results are satisfying. I would, however, ask to clarify what is happening in the experiments:

> Do I understand correctly? Does "Baseline" (e.g. Adam) mean the reward is trained with the Baseline optimizer and then the designs are trained with that optimizer too?

**Other Comments Or Suggestions:**

Algorithm 1. Could you please make it more legible? Right now it looks messy. And I doubt you have while loops inside while loops. I haven’t found it in the code either.

Table 1. Adam. TFBind8. Shouldn’t ROMO and GAMBO be in bold?

Line 403 (right). “Trabucco et al. (2022) used REINFORCE-style methods similar to Williams (1992) to learn a myopic sampling policy…” - what do you mean by myopic here?

**Other Strengths And Weaknesses:**

N/A

**Questions For Authors:**

If the limitation of the trade-off between diversity and quality is acknowledged and discussed, I will raise my score. Then, if my other concerns, regarding clarity and rigor, are addressed, I will consider raising my score further.

**Relation To Broader Scientific Literature:**

The paper is relevant for the field of offline MBO. It addresses an important problem of design diversity that is often omitted in the MBO literature.

**Theoretical Claims:**

I have read the theorems and the proofs. Read my comments about their rigor in "Claims and Evidence".

---

> ### Author Rebuttal · Authors · 2025-04-01
>
> We thank Reviewer sKFn for their helpful and constructive feedback on our manuscript, and appreciate their efforts in helping us improve our work. Please find our responses below - we would be happy to discuss further and answer any follow-up questions as necessary. Thank you!
>
> 1. **(Assumption in Lemma 3.1)** Thank you for this comment. The inequality in Equation (16) follows directly from monotonicity of the integral and holds even if $q^\pi(x)$ is non-zero in a measure-zero set. If we have misunderstood your concern, please let us know.
>
> 2. **(Reward Term in Eq. (5))** Thank you for this comment. We include a pure reward term in (5) because the forward surrogate model $r_\theta(x)$ provides an important (albeit sometimes inaccurate) signal of design quality outside the dataset $\mathcal{D}$. This strategy builds off prior work in offline MBO (e.g., [Trabucco et al. ICML (2021).](https://proceedings.mlr.press/v139/trabucco21a.html), [Yu et al. NeurIPS (2021).](https://arxiv.org/abs/2110.14188), [Yao et al. NeurIPS (2024).](https://arxiv.org/abs/2402.06532)). Without this reward term, the Shannon entropy term in **Lemma 3.3** would not appear in our derivation and so the generative policy is not explicitly encouraged to find designs that maximize against the $r_\theta(x)$ surrogate signal.
>
> 3. **(Clarification of "Baseline")** Thank you for this question. In general, there are two "optimizers" used in our experiments: (1) a ***model-training*** optimizer used to train a surrogate reward function on the offline dataset; and (2) a ***generative*** optimizer used to generate new designs by optimizing against the trained surrogate function from step (1). "Baseline" only refers to the latter **generative** optimizer.
>
> The reward functions used for all the experiments in **Table 1** were trained to approximate the oracle reward function using Adam as the model-training optimizer.
>
> We have better clarified this point in our revised manuscript.
>
> 4. **(Limitations of DynAMO)** We thank the Reviewer for this comment. You are correct - our method does indeed trade reward for diversity. In many applications of offline MBO, we argue it is most important to obtain a good maximum score over a good median score. We highlight this potential limitation of DynAMO in our revised Limitations section, included at [this link](https://postimg.cc/YjfWXyf4).
>
> 5. **(Algorithm 1 Presentation)** Thank you for this comment. We have revised **Algorithm 1** [here](https://postimg.cc/5jGz48wM) to improve its legibility. Regarding the while loops, the outer while loop terminates when the optimizer $a^b$ no longer improves, and is implemented as lines 281-308 of our `main.py` source code file in our Supplementary Material. The inner while loop terminates when the optimal Lagrange multiplier is found, and is implemented in lines 226-256 in the `src/dogambo/models/mlp.py` file (which is invoked in line 304 of our `main.py` source code file).
>
> We would be happy to make any additional changes to the presentation of Algorithm 1 to improve its legibility.
>
> 6. **(Table 1 Formatting)** We appreciate your careful eye to detail! For the TFBind8 optimization task using the Adam optimizer, the top performing method in terms of design quality was RoMA+ with a score of $96.5\pm 0.0$ (lower bound on 95% CI is 96.5). However, ROMO scores $95.6\pm 0.0$ (upper bound on 95% CI is 95.6) and GAMBO scores $94.0\pm 2.2$ (upper bound on 95% CI is 96.2). Because the 95% CI intervals of ROMO and RoMA+ (and also GAMBO and RoMA+) are non-intersecting, ROMO and GAMBO are correctly left unbolded.
>
> 7. **(Meaning of "Myopic")** We follow [Deisenroth et al.](https://dl.acm.org/doi/10.1561/2300000021), [Ngo](https://www.alignmentforum.org/posts/GqxuDtZvfgL2bEQ5v/arguments-against-myopic-training), and others to refer to policies that are trained according a value function with an discount factor of $\gamma=0$ in reinforcement learning as "myopic". Intuitively, this means that the surrogate reward function used by the REINFORCE algorithm only considers the current input design in computing the reward, and not future possible states like in traditional RL applications. We have better clarified this point in our revised manuscript.

---

> > ### Comment · Reviewer_sKFn · 2025-04-02
> >
> > Dear Authors,
> > Thank you for your response. Most importantly, thank you for adding the limitation of your work. I will raise my score. Before that, I would like to ask a few more questions.
> > >1. I disagree with you about the inequality in Equation (16), although it is not a big issue. Suppose the space $\mathcal{X}$ is continuous and the only optimum is $x^{\star}$. Then, your probability distribution can be given by $q^{\pi}(x) = \delta (x-x^{\star}) + I(x=x_0)$, where $x_0$ is a sub-optimal point and $I(\cdot)$ is an indicator function. This function is non-negative and integrates out to 1, so it is a valid density function, even though $x_0$ shows up with probability zero. This is a minor thing, but if you want to be maximally correct, you should prove that the distribution $q^{\pi^{\star}}$ has no support on sub-optimal points almost surely (no sub-optimal non-measure-zero set gets support). This would actually strengthen your theorem because then you can just say that $q^{\pi^{\star}}$ has positive support on a, possibly infinite, set of optimal solutions, but no support on a non-measure-zero set of sub-optimal ones. The proof technique is the same, you just have to change the wording.
> > >2. Regarding the explicit reward term in your optimization objective - I disagree again. You say that it is necessary for the entropy term to show up in Lemma 3.3. My point is, this term does not do much. Using linearity of expectation (which entropy and the KL divergence are) you can represent Equation (11) as a single KL divergence between $q^{\pi}$ and, let's call it $\hat{p}^{\tau}(x)$ - an EBM distribution like $p^{\tau}(x)$ but with a different temperature. Thus, the same optimization objective can be obtained from trying to minimize the KL divergence between an EBM, albeit with a different temperature than $\tau$.
> >
> > I'm curious to see what your thoughts are!

---

> > > ### Author Response · Authors · 2025-04-03
> > >
> > > Thank you for the additional feedback and opportunity for discussion! We are grateful for your support of our work and appreciate your efforts in helping us improve our manuscript. Please find our responses below:
> > >
> > > 1. **(Equation (16))** Thank you for the clarification, we apologize for misunderstanding your original comment. We have revised Lemma 3.1 in accordance with your feedback [here](https://postimg.cc/sQ5QdDQr).
> > >
> > > 2. **(Reward Term)**: We apologize for the confusion and believe that we might have misunderstood your initial comment. Indeed, we agree with you that our modified MBO optimization objective in Equation (5) is equivalent to a distribution-matching objective (i.e., minimizing the KL divergence with an EBM) for some different distribution $\hat{p}^{\tau}\_{\mathcal{D}}$. We show how to arrive at this result [here](https://postimg.cc/XXWCbdVB) in our revised Appendix. However, we cannot guarantee that
> > > $\hat{p}\_{\mathcal{D}}^{\tau}$ has the form $p_{\mathcal{D}}^{\tau'}$ for some $\tau'$ in the most general case. Separately, we include a pure reward term in Equation (5) since it explicitly shows that our objective both maximizes reward while also performing distribution matching, which is consistent with the formalism of prior MBO literature (e.g., [Trabucco et al. ICML (2022)](https://proceedings.mlr.press/v162/trabucco22a); [Yao et al. NeurIPS (2024)](https://arxiv.org/abs/2402.06532); [Trabucco et al. ICML (2021)](https://proceedings.mlr.press/v139/trabucco21a.html); [Yu et al. NeurIPS (2021)](https://arxiv.org/abs/2110.14188)), as recommended by Reviewer kDhU (Other Comments Or Suggestions).
> > >
> > > We would be happy to answer any additional questions you might have!

---

### Official Review · Reviewer_DFtV · 2025-03-10

**Overall Recommendation:** 3

**Summary:**

This paper proposes a distributed matching based adversarial optimization framework (DynAMO) aimed at addressing the issue of insufficient design diversity in offline model optimization (MBO) tasks. By explicitly modeling diversity objectives as a matching problem between generative design and offline dataset distribution, this method significantly improves diversity while ensuring the quality of candidate designs, and validates its effectiveness in multiple scientific fields such as biological sequence design.

**Claims And Evidence:**

Yes.

**Essential References Not Discussed:**

NAN.

**Experimental Designs Or Analyses:**

Yes.

**Methods And Evaluation Criteria:**

Yes.

**Other Comments Or Suggestions:**

NAN.

**Other Strengths And Weaknesses:**

Strengths:
* The experiment covers multiple scientific fields (such as Table A10) to verify the generalization ability of the method.
* Table A10 clearly distinguishes the performance of different methods (DynMO, AMO, DynAMO) and enhances readability through confidence intervals and bold annotations.


Weaknesses:
* The method relies on existing techniques such as KL divergence penalty and adversarial constraints, and further evidence is needed to demonstrate the unique value of their combination.
* The theoretical guarantee section (D.4) was not elaborated in the abstract, which may lack original breakthroughs in theoretical depth.
* The limitations of methods in ultra-high dimensional design spaces or sparse reward scenarios have not been discussed.
* The trade-off parameters between diversity objectives and quality objectives (such as τ value) may rely on domain experience tuning and lack universal guidance.

**Questions For Authors:**

See weaknesses.

**Relation To Broader Scientific Literature:**

* By reparameterizing the optimization objective as a combination of weighted entropy and divergence using Lemma 3.3, a closed form solution (Lemma 3.4) is provided, which solves the algebraic complexity problem that previous studies (such as those related to chi square divergence) could not directly apply. This theoretical contribution expands the application scope of f-divergence in offline optimization, complementing broader probability distribution alignment studies such as divergence optimization in generative models.
* D. Section 6 validated the computational feasibility of the method in large-scale design spaces through empirical comparisons (such as with the model proposed by Yu et al., 2021), filling the practical gap of model independent methods (such as Fu&Levine, 2021).

**Theoretical Claims:**

Yes.

---

> ### Author Rebuttal · Authors · 2025-03-31
>
> We thank Reviewer DFtV for their thoughtful feedback and careful consideration of our work. Please find our responses to your comments below.
>
> 1. **(Combination of KL Divergence and Adversarial Constraints)** Thank you for this comment - indeed, we believe that it is a *strength* of our method that DynAMO is able to leverage existing individual techniques--such as KL divergence penalization from imitation learning and adversarial constraints from the GAN literature--and combine them together in a unique way to solve a new problem (offline MBO). Namely, leveraging KL divergence penalization has not been shown in prior MBO literature, and leveraging adversarial regularization has only recently been explored by [Yao et al. NeurIPS (2024)](https://arxiv.org/abs/2402.06532). **Combining them together in a meaningful and principled way to solve the problem of offline MBO is the main novelty and contribution of our work.**
>
> Regarding the unique value of combining both techniques, we refer the Reviewer to **Supp. Tables A8-A10** in **Appendix E.2**, which shows that using both KL-divergence and adversarial regularization **together in combination** significantly improves the performance of DynAMO than when using either technique alone.
>
> 2. **(Theoretical Guarantee in Section D.4)** To clarify, our primary contribution of this work is the DynAMO algorithm. The goal of the theoretical bound presented in Section D.4 is to use standard, validated techniques from the literature to show theoretically what we already observe in our presented experiments: DynAMO can improve the diversity of designs obtained using different generative optimization methods. Indeed, we use fairly standard methods and techniques to arrive at our result in Section D.4 - for example, see [Ma et al. Proc NeurIPS (2022)](https://arxiv.org/abs/2206.03023), [Huang et al. (2024)](https://arxiv.org/abs/2407.13399), and [Liang (2016)](https://web.stanford.edu/class/cs229t/notes.pdf) for additional background.
>
> 3. **(Method Limitations)** Thank you for this comment. We highlight that many of our discrete MBO tasks (i.e., ChEMBL, Molecule) are problems over high-dimensional design spaces as discussed in prior work ([Maus et al. NeurIPS (2022)](https://arxiv.org/abs/2201.11872), [Yao et al. NeurIPS (2024)](https://arxiv.org/abs/2201.11872), [Trabucco et al. ICML (2021)](https://proceedings.mlr.press/v139/trabucco21a.html)). Furthermore, the DKitty task in our work includes a sparse and highly-sensitive oracle reward model as discussed in [Trabucco et al. ICML (2022)](https://proceedings.mlr.press/v162/trabucco22a.html). We have better highlighted these points in our revised manuscript, and have also included an additional paragraph explicitly discussing the challenges with optimization in these settings in alignment with the Reviewer's feedback.
>
> 4. **(Setting Hyperparameters)** Thank you for this comment. We highlight that **all experimental results presented in the main text use the same hyperparameter values (e.g., $\tau=1.0$ and $\beta=1.0$) across *all optimizers and MBO tasks***. Furthermore, **Supp. Figures A4-A6** show that the performance of our method is robust to the choice of hyperparameters. In short, while we domain expertise and tuning of hyperparameters might improve our method further, it is by no means necessary to achieve good experimental results using DynAMO.

---

### Official Review · Reviewer_scBu · 2025-03-13

**Overall Recommendation:** 2

**Summary:**

This paper presents a novel approach to incorporating design diversity as an explicit objective in the offline model-based optimization problem. Specifically, the original optimization objective is modified to enhance the diversity of generated samples using a distribution matching technique inspired by recent advances in imitation learning. Additionally, a source critic constraint is introduced to mitigate out-of-distribution evaluations of the reward surrogate model. Extensive experiments demonstrate the effectiveness of DynAMO in promoting diversity while maintaining the discovery of high-quality candidates.

**Claims And Evidence:**

The claims in this paper are well-supported by clear and compelling evidence.

**Essential References Not Discussed:**

No, there are no essential related works missing that are crucial for understanding the key contributions of this paper.

**Experimental Designs Or Analyses:**

I have verified the soundness and validity of the experimental designs and analyses presented in the main text, as well as the additional results in the appendix.

**Methods And Evaluation Criteria:**

The proposed method and evaluation criteria, including benchmark tasks and baselines, are well-suited to the problem at hand.

**Other Comments Or Suggestions:**

I suggest presenting the experimental results in a clearer and more structured manner to improve readability and analysis.

**Other Strengths And Weaknesses:**

Strengths:
- This paper proposes a novel approach to explicitly balancing the trade-off between reward optimality and design diversity in offline optimization.
- The writing is clear and well-structured.
- The results demonstrate that DynAMO enhances diversity while maintaining high-reward candidates to some extent.

Weaknesses:
- The experimental results in both the main text and appendix are poorly presented, primarily as large tables with multiple bold elements, making it difficult for readers to analyze the data.
- While the paper shows an improvement in diversity, it comes at the cost of reward optimality. Specifically, although the mean rank appears low, DynAMO does not achieve state-of-the-art performance across all settings.

**Questions For Authors:**

1. How does the proposed regularization in Eq. (5) enhance the diversity of candidate designs? According to line 159, $\tau$-weighted probability distribution has nonzero support only at offline data points, meaning $p_D^\tau(x)=0$ for $x \not \in D$. Consequently, the KL regularization does not ensure diversity of samples in out-of-distribution (OOD) regions, regardless of the choice of $\tau$.

2. How does the introduction of the source critic constraint prevent the forward surrogate model from evaluating wildly out-of-distribution inputs? Additionally, could $c^*$ in this constraint be replaced by the forward surrogate $r_\theta$?

3. There may be potential unfairness in the comparisons presented in Table 1. First, the reported baseline results using non-original optimizers may not be optimal. For instance, the COMs baseline was originally designed with the Grad optimizer, meaning its hyperparameters were tuned accordingly. When integrated with other optimizers such as Adam or CMA-ES, did the authors retune its hyperparameters? Second, since each dataset has its own value range, normalizing results to a (0,1) scale is necessary to ensure fair comparisons, especially for metrics like the Optimality Gap (Opt. Gap).

4. Although DynAMO achieves a lower Pairwise Diversity score and competitive Best@128 performance compared to other methods, it exhibits poor Median@128 performance in Table A1. Does the lower Pairwise Diversity score stem from weaker performance at other percentile levels? If so, this suggests that the reward distribution of candidates has a wide range, which contradicts the goal of offline optimization—finding a distribution concentrated on high-value designs. The authors should conduct additional experiments to validate this hypothesis. Furthermore, I am curious about a straightforward approach to improving the diversity of existing baselines. For instance, COMs initializes the K-best designs from the offline dataset and then applies gradient ascent to refine them. Instead, could selecting K random designs as initialization increase the diversity of the final candidates?

**Relation To Broader Scientific Literature:**

This paper presents a promising direction in offline optimization with potential applications in designing more effective drugs, engineering new materials with desirable properties, and solving various scientific challenges. While previous works (COMs, ExPT, BOSS) focus solely on identifying the best designs, this paper introduces a method to balance the trade-off between optimal quality and diversity.

**Theoretical Claims:**

I have verified the correctness of the proofs for Lemma 3.1 and Lemma 3.3.

---

> ### Author Rebuttal · Authors · 2025-03-31
>
> We appreciate Reviewer scBu for their insightful feedback and thorough evaluation of our work. We have provided our responses to your comments below and would be happy to answer any follow-up questions. Thank you!
>
> 1. **(Presentation of Results)** To improve the legibility of our results, we summarize the main findings on the task-averaged Rank and Optimality Gap metrics in the [main table](https://postimg.cc/ygg6fyPt) in our revised manuscript, and move the detailed breakdown of algorithm performance by task to the Appendix for the interested reader.
>
> 2. **(Diversity versus Quality)** To be clear, **we do not claim that DynAMO will propose designs of higher *quality* than other baseline methods across all settings.** (In fact, *no baseline method* is state-of-the-art across all tasks.) The goal of DynAMO is to increase the diversity of designs without significantly reducing the quality of best design proposed. Indeed, across multiple offline MBO tasks evaluated, there is often no statistically significant difference between the quality of designs proposed by DynAMO and other baseline methods according to the Best@128 evaluation metric. **However, DynAMO is consistantly state-of-the-art in producing diverse sets of designs across all settings.** We have better clarified this primary goal and use case of DynAMO in our manuscript.
>
> 3. **(Equation (5) for Diversity)** To clarify, we do not want to sample designs that are truly OOD, as we cannot guarantee the correctness of these designs using an arbitrary surrogate $r_\theta(x)$. Ideally, it would be great to discover designs that are both novel and significantly OOD, but this is not possible while guaranteeing correctness in the offline setting. Consistent with prior work (e.g., [Yao et al. NeurIPS (2024).](https://arxiv.org/abs/2402.06532)), DynAMO seeks to generate **interpolated, in-distribution points** to balance diversity with naive reward maximization. Through the regularization in Eq. (5), we aim to learn a generative policy that proposes designs with coverage of the search space similar to the offline dataset $\mathcal{D}$. If $\mathcal{D}$ is too small or has poor coverage, then our method would likely be less effective - we discuss this in further detail in our Limitations section of our revised manuscript.
>
> 4. **(Source Critic Constraint)** The primary goal of the source critic is to penalize designs that are OOD with regards to the offline dataset $\mathcal{D}$. If $c^*(x)$ is large, then $x$ is likely from the same distribution as $\mathcal{D}$ and so we can trust that the prediction $r_\theta(x)$ is a good estimate of $r(x)$ with reasonable confidence. If $c^*(x)$ is small, then $x$ is likely not from $\mathcal{D}$ and so we penalize the reward associated with $x$ to avoid choosing the design as a proposed candidate. Note that this source critic is different from the forward surrogate $r_\theta$ (approximating oracle $r$), and so we cannot replace $c^*$ with the forward surrogate.
>
> We note that our use of $c^*(x)$ builds on prior literature from [Yao et al. Proc NeurIPS (2024)](https://arxiv.org/abs/2402.06532) and [Arjovsky et al. Proc ICML (2017)](https://arxiv.org/abs/1701.07875) - we refer the Reviewer to these prior works for additional discussion.
>
> 5.  **(Fairness in Comparisons)** To ensure fairness in comparisons, we tuned the hyperparameters of DynAMO according to *only on its performance using the Grad. optimizer*, following the same paradigm as COMs, ROMO, and GAMBO. All hyperparameter values for all methods (including DynAMO) were then held constant for all of the other optimizers. While each of the methods (including DynAMO) might benefit from further optimizer-specific and task-specific hyperparameter fine-tuning, such an approach is not feasible in real-world applications. All of the quality metrics reported have already been max-min normalized to a (0, 1) scale according to the equation $\hat{y}=\frac{y-y_{\text{min}}}{y_{\text{max}}-y_{\text{min}}}$ in alignment with prior work. However, we then multiplied the normalized metrics $\hat{y}$ by 100 to improve the legibility of our results, as highlighted in the table captions where relevant.
>
> 6. **(Optimization Initialization Strategy and Distribution of Oracle Objective Values)** We thank the Reviewer for suggesting an alternative strategy to improve the diversity of designs through random initialization. We compared top-$k$ and random-$k$ initialization strategies for first-order optimization methods in this [table](https://postimg.cc/SndKjsRT). For most methods, neither initialization strategy results in consistently more diverse designs (except for DynAMO, which benefits from random-$k$ initialization). Furthermore, both COMs and RoMA suffer from *lower quality of designs* proposed when a random-$k$ initialization strategy is used. **These results highlight the significance of DynAMO as a novel algorithm to consistently increase the diversity of final candidates.**

---

> > ### Comment · Reviewer_scBu · 2025-04-03
> >
> > Thank you for your responses. However, my main concerns remain unaddressed:
> >
> > ### 1. Potential Trade-off Between Diversity and Performance
> > Your empirical results suggest that the observed increase in diversity might come at the expense of overall performance in the proposed designs. Specifically, this could occur if, among the 128 final selected candidates, only a few (or even just one) achieve a high score, leading to a high **128@best** value. Meanwhile, the remaining candidates may be widely distributed across the search space, increasing **Pairwise Diversity** but lowering their individual performance, which in turn reduces **128@median** and other percentile-based metrics. This raises concerns about whether the method truly enhances diversity while maintaining strong performance or if the reported diversity increase results from a broader but suboptimal search. If this is not the case, the authors should provide explicit evidence demonstrating that the improvement in diversity does not come at the cost of performance degradation.
> >
> > ### 2. Unfair Comparisons
> > Despite tuning hyperparameters for the **DynAMO-Grad** optimizer, its reported performance appears less competitive than other existing methods, as shown in **Table A5**. Additionally, my concern regarding the reported performance values after normalization remains unresolved. If the authors indeed normalize the scores to the range **(0,1)** and then multiply by **100**, all resulting values should theoretically fall within **[0,100]**. However, in the **Molecule** task, some reported performance values exceed **100**, contradicting this normalization process. This discrepancy raises questions about whether the normalization was correctly applied, whether there were inconsistencies in scaling, or if additional transformations were performed.

---

> > > ### Author Response · Authors · 2025-04-03
> > >
> > > Thank you for your additional comments and opportunity for discussion! We would be happy to try to address your remaining concerns:
> > >
> > > 1. **(Diversity)** We acknowledge the concern that the observed increase in diversity may come at the expense of overall performance, particularly if only a few of the 128 selected candidates achieve high scores.  As an example, **Supp. Fig. A2** (top left panel) provides evidence of how DynAMO improves diversity while still identifying high-performing designs. Compared to the baseline, DynAMO produces a distribution of designs with a longer tail over better-than-average oracle scores. This tail of designs is unique to DynAMO and is why our method achieves a strong Best@128 score value.
> > >
> > > However, we acknowledge that the rest of the distribution (i.e., the majority of designs) is indeed suboptimal. This is an inherent tradeoff in our method: by encouraging the exploration of a broader search space, DynAMO is able to uncover a **diverse set of optimal and near-optimal designs that are meaningfully distinct from one another**; however, a significant number sub-optimal designs might also be included as a side effect.
> > >
> > > We agree with the Reviewer that the inclusion of these suboptimal designs could be viewed as a potential limitation. We have explicitly addressed this point in our revised Limitations section of our manuscript to discuss the trade-off between maintaining high diversity and ensuring consistently high performance across all selected candidates. We appreciate the opportunity to clarify this aspect of our work.
> > >
> > > 2. **(DynAMO-Grad Performance and Normalization)** Thank you for this comment. Regarding the performance of DynAMO-Grad, Grad. is a relatively weak backbone optimization method: a clear example is that the optimality gap of baseline CMA-ES, BO-qEI, and BO-qUCB all significantly ouperform that of baseline Grad. At its core, DynAMO is an **objective-modifying algorithm** and still relies on an underlying optimizer to optimize against the DynAMO-modified objective. If the optimizer is relatively weak, such as with Grad., then the best performance achievable with the method may still be subpar even though DynAMO-Grad. outperforms baseline Grad.
> > >
> > > Regarding the normalization schema, the normalization results in a score of 100 if a design achieves an oracle score equal to $y_{\text{max}}$, the best score in the offline dataset. A score greater than 100, such as what we observe in the Molecule task, means that a design was proposed that was *even better* than the best design in the offline dataset. This is indeed the overarching motivation of offline MBO: to discover designs that are better than what we have ever seen previously in the offline dataset. We confirm that normalization was correctly applied (source code available in the Supplementary Material), and there were no inconsistencies in scaling or additional transformations performed.
> > >
> > > We would be happy to answer any additional comments or questions. Thank you!

---

### Official Review · Reviewer_kDhU · 2025-03-14

**Overall Recommendation:** 3

**Summary:**

The paper introduces Diversity in Adversarial Model-based Optimization (DynAMO), a new approach for offline model-based optimization (MBO) that aims to generate diverse and high-quality design candidates. The core idea is to frame diversity as a distribution matching problem by optimizing a KL-divergence-based objective, ensuring that the generated candidates capture the diversity inherent in the offline dataset. The challenge is not only to find high-reward designs but also to ensure diversity, avoiding solutions that cluster around a single optima. Additionally, an adversarial source critic is incorporated to prevent the surrogate model from overestimating rewards on out-of-distribution designs, improving the robustness of offline optimization.

## update after rebuttal
The authors addressed most of my concerns. I update my rating towards weak acceptance.

**Claims And Evidence:**

- DynAMO improves both design diversity and quality compared to baseline methods:

While DynAMO consistently improves pairwise diversity and achieves competitive results against baselines, concerns remain regarding the actual performance improvements. For example, in the D’Kitty evaluation, DynAMO does not surpass the best designs already present in the offline dataset, contradicting the fundamental goal of offline MBO—to improve upon the given data.
Additionally, normalizing the results, as is standard in offline MBO literature, would likely show that DynAMO’s improvements are marginal and possibly indistinguishable from other methods. Even without normalization, results on tasks like ChEMBL are too close across methods, making it difficult to determine whether DynAMO truly provides a performance advantage.
If the goal is to discover multiple high-quality design modes, then a slight sacrifice in top performance might be expected. However, when analyzing Median@128, which should reflect a more consistent quality across diverse designs, DynAMO lags behind other methods. This raises concerns about whether diversity is translating into useful variety or merely scattering designs without maintaining quality. Since the goal of offline MBO is to improve upon the dataset, the results question whether DynAMO's diversity actually contributes to better design candidates or simply increases search space coverage without meaningful gains.

- DynAMO is optimizer-agnostic and improves performance across all optimization strategies:

Extensive results comfirm that dynamo is compatiblewith different optimization strategies.

**Essential References Not Discussed:**

The authors provide a strong coverage of literature across various topics; however, some works in offline MBO are missing, including:
1. Parallel-mentoring for Offline Model-based Optimization (https://arxiv.org/pdf/2309.11592).
2. Importance-aware Co-teaching for Offline Model-based Optimization (https://arxiv.org/pdf/2309.11600).
3. Offline Model-Based Optimization via Policy-Guided Gradient Search (https://arxiv.org/pdf/2405.05349).
4. Learning Surrogates for Offline Black-Box Optimization via Gradient Matching (https://arxiv.org/pdf/2503.01883).
5. Robust Guided Diffusion for Offline Black-Box Optimization (https://arxiv.org/pdf/2410.00983).

(3) is particularly relevant to the reinforcement learning section in related work as it leverages offline RL techniques to guide the search for quality designs..

**Experimental Designs Or Analyses:**

Yes. Please check the previous sections.

**Methods And Evaluation Criteria:**

- Pairwise Diversity (PD) and the other discussed metrics are reasonable measures to evaluate whether DynAMO increases diversity in design generation.

- Best@128 and Median@128 are standard offline MBO performance metrics and are appropriately used.

- ChEMBL results are too close across methods, making it unsuitable for comparing baselines. Similarly, UTR is known to produce similar results across methods, as shown in prior work: Bidirectional Learning for Offline Infinite-width Model-based Optimization (https://arxiv.org/pdf/2209.07507) and Conservative Objective Models for Effective Offline Model-Based Optimization (https://arxiv.org/pdf/2107.06882v1).

- Non-normalized results may exaggerate reported gains—preferably, results should be normalized as done in offline MBO literature.

- In offline RL, KL-divergence prevents OOD actions as a regularization constraint, keeping optimization within the dataset. This setup may enhance diversity but not necessarily high-quality diverse designs, as seen in Median@128 results. The intuition from imitation learning is not valid since the offline dataset contains both good and bad examples, not only expert designs.

- Figure 1 is intended to highlight DynAMO selecting diverse designs, but most appear scattered, raising questions about the trade-off between diversity and maintaining quality. The figure suggests that DynAMO generates varied designs, but they are not concentrated around the different global maxima as intended.

- The paper states that secondary objectives (e.g., manufacturing cost, drug toxicity) can benefit from diversity, yet the chosen tasks do not include any secondary objectives. Would it be more appropriate to validate this claim using offline multi-objective optimization tasks, such as those discussed in Offline Multi-Objective Optimization (https://arxiv.org/pdf/2406.03722)

**Other Comments Or Suggestions:**

- Why frame the offline optimization problem using RL terminology and reward functions instead of the standard terminology used in offline model-based optimization literature?

**Other Strengths And Weaknesses:**

Strengths:
- Diversity is an important factor often ignored in offline MBO literature, and this paper explicitly integrates it into the optimization process.
- Extensive discussion of the method and its components.
- The authors shared the code.

**Questions For Authors:**

- Please check the previous sections.

**Relation To Broader Scientific Literature:**

DynAMO builds on offline MBO methods, incorporating an adversarial critic to mitigate overestimation in surrogate models. Its KL-divergence regularization aligns with offline RL techniques, which constrain policies to remain within the offline dataset distribution to prevent OOD actions. The method also relates to diversity-driven optimization, sharing similarities with quality-diversity algorithms and generative modeling approaches that promote multi-modal exploration. Additionally, DynAMO leverages distribution matching through KL-divergence, guiding optimization to maintain design diversity while staying close to the offline data distribution.

**Theoretical Claims:**

No.

---

> ### Author Rebuttal · Authors · 2025-03-31
>
> We thank Reviewer kDhU for their thoughtful comments and feedback on our proposed work, which we believe have substantially improved the quality of our manuscript. Please find our responses to your comments below.
>
> 1. **(Performance According to the Median@128 Metric)** Thank you for this comment. DynAMO does indeed trade reward (according to the Median@128 metric) for diversity. Diversity enables us to find a greater subset of optimal and sub-optimal designs than baseline methods; naturally, this may result in a lower median score since DynAMO cannot simply return points near the global maximum (e.g., see **Supp. Fig. A2**). In many applications of offline MBO, it is more important to obtain a good maximum score than a good median score, and we therefore prioritize the Best@128 and pairwise diversity metrics in our evaluation of DynAMO. We highlight this potential limitation of DynAMO in our revised Limitations section, included at [this link](https://postimg.cc/YjfWXyf4).
>
> 2. **(ChEMBL and UTR Tasks)** We would be happy to move the results associated with these two tasks to the Appendix in accordance with this feedback. (Notably, DynAMO's average optimality gap consistently *increases* and its average ranking stays the same or improves after excluding the ChEMBL and UTR tasks.)
>
> 3. **(Normalization of Results)** All of the quality metrics reported have already been max-min normalized (0,1) according to the equation $\hat{y}=\frac{y-y_{\text{min}}}{y_{\text{max}}-y_{\text{min}}}$ in alignment with prior work in the offline MBO literature. We multiplied the normalized metrics $\hat{y}$ by 100 to improve the legibility of our results, as highlighted in the table captions.
>
> 4. **(Good and Bad Examples in Offline RL)** We note that KL constraints are also commonly used in offline RL, where the dataset can include non-expert trajectories. Intuitively, the direction of the KL term keeps the generated samples inside the support of the offline dataset, with failing to match the offline dataset outside its support being penalized less. We can further tune the tradeoff by using the temperature hyperparameter $\tau$ defined in Equation (6). Using a $\tau>0$, designs that are good are upweighted in the reference distribution $p^\tau(x)$ compared to bad designs - see **Fig. A1** for examples. As a result, we are able to use the standard intuition and techniques derived from imitation learning, and show in our work that our approach works both theoretically and empirically.
>
> 5. **(Figure 1)** We thank the Reviewer for this comment. Indeed, DynAMO generates a diversity of designs, many of which are not concentrated around the global maxima. However, the point of DynAMO (and **Figure 1**) is that at least *a few* of the proposed designs are located around the different global maxima. This is in contrast with the baseline method, which proposed a batch of designs that might all be near-optimal, but are all clustered around the same optima and region of the input space. Our primary motivation of DynAMO is that obtaining a small number of diverse and high-quality designs is better than obtaining a large number of high-quality but near-identical designs. If a particular application of offline MBO does not suit this use case, then DynAMO may not be well-suited for the application at hand. We have better clarified this point in **Figure 1** of our revised manuscript.
>
> 6. **(Diversity for Secondary Objectives)** To demonstrate how diversity can enable better downstream exploration of secondary objectives, we compare DynAMO-enhanced optimization methods against their baseline counterparts and report the results [here](https://postimg.cc/yDbHbzVP). We find that if an optimization method achieves a higher pairwise diversity score, it also consistently achieves a larger variance in the values of the secondary objectives, supporting the idea that secondary objectives can benefit from diversity.
>
> To be clear, we do not claim that DynAMO will discover designs that maximize multiple objectives while only optimizating against a single objective. The goal is that by increasing diversity, we can better interrogate other design properties, which will hopefully be different from one another.
>
> 7. **(Additional Baselines)** We thank the Reviewer for sharing these additional baselines with us. We have included all 5 referenced methods in our updated design [quality](https://postimg.cc/m17L4ZKS) and [diversity](https://postimg.cc/jwpxBn35) results. Our results do not significantly change with the inclusion of these additional baselines.
>
> 8. **(Choice of Terminology)** Thank you for this comment. We initially chose to leverage terminology from both the offline MBO and RL literature to more explicitly highlight how DynAMO adapts techniques inspired from offline RL to the MBO setting. We have revised our manuscript to remove references to reward functions and other RL-centric terminology to better align with existing offline MBO literature.

---

### Decision · Program_Chairs · 2025-05-01

**Decision:**

Accept (poster)

**Comment:**

**Summary**: The paper considers a new setup for offline optimization problems where the typical goal is to find designs maximizing an objective/property of interest using only a static offline dataset. The paper introduces a criterion for including diversity of generated designs as an explicit criterion in the offline optimization problem setup. The key idea is to formulate adding diversity explicitly as a distribution matching problem, using a KL-divergence penalty to encourage the distribution of generated designs to match the offline dataset. Building on top of Yao et al, the paper  incorporates an adversarial source critic constraint as well. The paper provides a theoretical analysis that leads  to a tractable Lagrangian dual formulation for the overall constrained optimization objective.

The paper received mixed reviews. All reviewers recognized the importance of the problem setup. (Reviewers kDhU, scBu, sKFn) raised concern about the potential trade-off between diversity and quality, particularly highlighted by the Median@128 metric where the proposed method’s (DyNAMO) performance was a bit on the lower side. I found the author's rebuttal very convincing in addressing all the concerns. They acknowledged the diversity-quality trade-off and expanded the limitation section.  Reviewer sKFn even increased their scores after the rebuttal.

Overall, I believe this paper tackles an important aspect relevant for offline optimization: generating a diverse set of high-quality solutions. The proposed DynAMO method is technically sound, combining ideas from distribution matching and adversarial learning. The empirical performance is solid and the authors share well-written code for reproducibility. **Therefore, I recommend accepting the paper.** However, I request the authors to include all the points discussed in the rebuttal period in the final camera-ready version, especially including all relevant baselines mentioned by reviewer kDhU and detailed description of hyperparameter tuning as mentioned by reviewer scBu.